# Understanding and Enhancing Mask-Based Pretraining towards Universal Representations

**Mingze Dong**
Yale University
mingze.dong@yale.edu

**Leda Wang**
Yale University
leda.wang@yale.edu

**Yuval Kluger**
Yale University
yuval.kluger@yale.edu

## Abstract

Mask-based pretraining has become a cornerstone of modern large-scale models across language, vision, and recently biology. Despite its empirical success, its role and limits in learning data representations have been unclear. In this work, we show that the behavior of mask-based pretraining can be directly characterized by test risk in high-dimensional minimum-norm ("ridge-less") linear regression, without relying on further model specifications. Further analysis of linear models uncovers several novel aspects of mask-based pretraining. The theoretical framework and its implications have been validated across diverse neural architectures (including MLPs, CNNs, and Transformers) applied to both vision and language tasks. Guided by our theory, we propose an embarrassingly simple yet overlooked pretraining scheme named *Randomly Random Mask AutoEncoding* ($\mathbf{R}^2\mathbf{MAE}$), which enforces capturing multi-scale features from data and is able to outperform optimal fixed mask ratio settings in our linear model framework. We implement $R^2MAE$ in vision, language, DNA sequence, and single-cell models, where it consistently outperforms standard and more complicated masking schemes, leading to improvements for state-of-the-art models. Our code is available at this URL.

## 1 Introduction

Mask-based pretraining has emerged as a unifying paradigm for self-supervised learning across natural language [1–4], vision [5–13], and biological domains [14–22]. This approach is prevalent particularly for data that cannot be presented sequentially, such as images and tabular data [23]. The representations learned through masked-based pretraining have consistently yielded state-of-the-art zero-shot and fine-tuning performances on diverse downstream tasks [1, 23, 24, 6, 16].

Despite the widespread success of masked autoencoding pretraining schemes, fundamental questions remain about why and how it helps in learning meaningful data representations. Several theoretical works [25–28] investigated its underlying mechanism using different frameworks, yet two critical questions on the qualitative role of masking persist:

- (**Universality across different contexts**) The scheme proves effective across various data domains, masking designs, and neural network architectures beyond transformers. This suggests its underlying mechanism is fundamental and architecture-agnostic.
- (**Diversity across domains and tasks**) The optimal behavior of mask pretraining differs significantly across contexts. In language modeling, BERT employs a moderate masking ratio of 15% [1], while in vision, surprisingly high masking ratios (75%) can produce superior representations despite removing most of the visual content [6]. Moreover, the optimal masking ratio varies even across different downstream tasks for a single model [29].

The performance curves of models with different masking ratios have been explicitly characterized in several works [6, 29], which serve as a foundation for several theoretical explanations [25, 28]. An

39th Conference on Neural Information Processing Systems (NeurIPS 2025).

intriguing phenomenon is the existence of a sweet-spot masking ratio that achieves optimal model performance. Additionally, several interesting quantitative behaviors of the performance curve, such as plateaus in the low-masking and near-optimal-masking regimes, appear in a number of cases [6].

To our knowledge, no previous work has proposed a theoretical framework general enough to address the aforementioned qualitative challenges, nor have they successfully explained these quantitative behaviors of mask pretraining schemes. Moreover, prior works do not explain the effect of model size in determining the optimal mask ratio [29]. In this work, our main contributions are the following:

1. We introduce a novel theoretical framework based on a high-dimensional linear regression setting tailored to mask prediction. We demonstrate that the test risk of this considered model recapitulates both qualitative and quantitative behaviors of diverse pretrained neural networks with respect to masking ratio in large-scale vision and language models.

2. We derive explicit expressions for the test risk under several cases using random matrix theory [30–32] with novel theoretical contributions. Our results suggest that previous observations on mask pretraining behaviors can be explained by solely bias-variance decomposition.

3. We identify and validate several aspects of mask-based pretraining—previously unexplained or overlooked—in various architectures across vision and language tasks: 1) The scheme is only beneficial in the overparametrized regime; 2) The optimal masking ratio is task and model-size-dependent; 3) It enforces feature magnitude disparity.

4. Building on insights from the linear model, we propose $\mathbf{R^2MAE}$, a simple but novel pretraining strategy that replaces fixed mask ratio with uniformly sampled mask ratios from a predefined range. $R^2MAE$ yields consistent improvements in vision, language, DNA, and single-cell pretraining, outperforming standard masking and various existing enhancement strategies on downstream zero-shot, linear probing, and fine-tuning tasks. $R^2MAE$ enforces models to capture different feature scales, and is able to outperform optimal fixed masking ratio performance in real data and linear models under appropriate mask range settings.

## 2 Related works

**Mask-based pretraining in language, vision, and biology.** Mask pretraining has become a dominant self-supervised learning approach in recent years, with significant developments in language modeling, computer vision, and biology. In NLP, BERT introduced the Masked Language Model (MLM) objective where random 15% tokens are corrupted and predicted from context [1]. MLM has been adapted in numerous works with modifications [2–4]. Studies show optimal masking ratios may exceed the 15% default and vary by task [29], while dynamic mask scheduling may improve performance [33, 34]. Other approaches propose learnable masks during pretraining [35–37]. In computer vision, researchers drew inspiration from BERT to devise masked image modeling methods, explored in ViT and BEiT [5, 7]. He et al. [6] propose MAE, showing that images benefit from an extremely high mask ratio of 75% to achieve state-of-the-art results in downstream tasks. This finding sparked numerous empirical studies on evaluating and improving the MAE scheme [8–13, 38].

The mask-based pretraining paradigm has also made inroads into biological data science, in particular for DNA sequences and single-cell gene expressions. Those DNA models are usually trained directly by the BERT pretraining objective [39, 16], whereas variants of mask rates were explored for single-cell self-supervised learning models ranging from 15% to 90% [18, 20, 21, 17, 40]. To the best of our knowledge, there are currently no successful improvements of mask-pretraining schemes in biological models beyond simply tuning masking rates. See Appendix A.1 for additional discussions.

**Understanding neural networks through linear models.** The connection between neural networks and linear models in the proportional regime has been extensively studied in recent years. Here the proportional regime refers to the asymptotic setting where the feature number $d$ and the sample number $n$ both tend to infinity, with their limit ratio $\gamma = d/n \in (0, \infty)$. For instance, the double descent phenomenon, where test error decreases with overparameterization was characterized empirically in deep networks [41] and theoretically shown for high-dimensional ridge(-less) regression [42, 43]. Recent works also addressed generalized polynomial regimes where $\gamma = d/n^\alpha \in (0, \infty)$[44–46]. The equivalence between nonlinear models and linear Gaussian models with matching moment statistics, i.e., universality, have been demonstrated or conjectured in multiple settings [47–49]. Further background of high-dimensional linear models can be seen in Appendix A.2.

**Theoretical endeavors to understand mask pretraining.** Recent theoretical investigations aimed to provide insights into mask-based pretraining objectives. Cao et al. [26] analyzed MAE's attention mechanism through integral kernels and Pan et al. [50] demonstrated autoencoders' capacity to preserve semantic information. Zhang et al. [25] suggests that masking creates implicit positive pairs relevant to contrastive learning. Yue et al. [27] reframed MAE as local contrastive learning where reconstruction loss contrasts different image regions. Kong et al. [28] developed a latent variable framework to explain the existence of optimal masking rate in MAE. To our knowledge, no prior research has precisely characterized the quantitative phenomena observed in mask-based pretraining, nor can these approaches be readily generalized across data domains and masking designs.

# 3 A theoretical framework for mask-based pretraining using high-dimensional linear models

In this work, we formulate the **feature-level mask autoencoding** problem as follows. Let $\boldsymbol{x} = (x_1, \ldots, x_{d+1}) \in \mathbb{R}^{d+1}$ be an input sample, where indices $\{1, \ldots, d+1\}$ denote features (tabular data) or positions (image/language data). A binary mask $\boldsymbol{z} = (z_1, \ldots, z_{d+1}) \in \{0, 1\}^{d+1}$ yields the corrupted input $\boldsymbol{x}' = \boldsymbol{x} \odot \boldsymbol{z}$. The model $f^\theta : \mathbb{R}^{d+1} \to \mathbb{R}^{d+1}$ is trained to reconstruct the original values $x_i$ for features where $z_i = 0$. Denoting the set of masked indices as $S_m = \{i | z_i = 0\}$ and using the Mean Squared Error (MSE) loss $L(a, b) = \|a - b\|^2$, the objective per sample is:

$$\sum_{i \in S_m} L(f^\theta(\boldsymbol{x}')_i, x_i). \tag{1}$$

The purpose of setting the feature dimensionality as $d + 1$ will become clear in the next section. This approach, particularly when $f^\theta$ employs a (transformer-based) encoder-decoder architecture, aligns with prominent masked autoencoding methods like MAE (vision) [6], BERT (language/DNA) [1, 14, 39, 16], and masked autoencoders for single-cell genomics [17, 18, 21].

## 3.1 Reduced linear model

To make exact analysis of this feature-level mask autoencoding problem feasible, we introduce two primary simplifications. First, we assume the model $f^\theta$ is linear in its input $\boldsymbol{x}'$ and has no bias term. Specifically, the reconstruction for the $i$-th original feature $x_i$ is given by:

$$f^\theta(\boldsymbol{x}')_i = \boldsymbol{x}'\boldsymbol{\beta}_i, \quad \text{where } \boldsymbol{\beta}_i \in \mathbb{R}^{d+1} \text{ is a coefficient vector specific to feature } i. \tag{2}$$

Note that if $x_i$ is the feature being reconstructed (i.e., $z_i = 0$), then the $i$-th component of the input $\boldsymbol{x}'$ is $(\boldsymbol{x}')_i = x_i z_i = 0$. Consequently, the $i$-th component of $\boldsymbol{\beta}_i$, $(\boldsymbol{\beta}_i)_i$, does not contribute to the prediction $f^\theta(\boldsymbol{x}')_i = \sum_{j \neq i}(\boldsymbol{x}')_j (\boldsymbol{\beta}_i)_j$. Second, we assume the coefficient vectors $\{\boldsymbol{\beta}_i\}_{i=1}^{d+1}$ are independent sets of parameters across different target features $i$. This allows the problem to be treated as $d + 1$ parallel, though potentially coupled through data, regression-like tasks.

We next consider a stylized version of one such reconstruction task. Let $\boldsymbol{y} = (y_1, ..., y_n) \in \mathbb{R}^n$ represent an arbitrary single feature from the original sample that we aim to reconstruct (e.g., $y = \boldsymbol{x}_k$ for some $k$). We formulate its corresponding regression problem as follows, using $n$ for the total number of training samples. The feature dimension is now $d$ as one feature is removed from $\boldsymbol{x} \in \mathbb{R}^{d+1}$. Let $\boldsymbol{X} \in \mathbb{R}^{n \times d}$ be the matrix containing all $n$ original, unmasked training samples with feature $\boldsymbol{y}$ removed. We henceforth denote the $j$-th row of $\boldsymbol{X}$ as $\boldsymbol{x}_j$. We consider the following teacher model:

$$\boldsymbol{y} = \boldsymbol{X}\boldsymbol{\beta} + \boldsymbol{\epsilon}. \tag{3}$$

Here, $\boldsymbol{\beta} \in \mathbb{R}^d$ is the ground-truth coefficient vector. The noise $\boldsymbol{\epsilon} = (\epsilon_1, ..., \epsilon_n) \in \mathbb{R}^n$ with each entry $\epsilon_j$ i.i.d. and $\mathbb{E}[\epsilon_j] = 0, \mathbb{E}[\epsilon_j^2] = \sigma^2$. Each sample $\boldsymbol{x}_j$ is assumed to have zero expectation $\mathbb{E}[\boldsymbol{x}_j^\top] = \boldsymbol{0}$, and covariance $\boldsymbol{\Sigma} = \mathbb{E}[\boldsymbol{x}_j^\top \boldsymbol{x}_j]$. We denote $\gamma = d/n$, $r = \|\boldsymbol{\beta}\|$, $\tilde{\boldsymbol{\beta}} = \boldsymbol{\beta}/\|\boldsymbol{\beta}\|$, and $\kappa = \sigma^2/r^2$.

In the random-mask autoencoding task, each feature chosen as a target is selected with probability $p$. Thus, for the regression on $\boldsymbol{y}$, the effective number of samples is $\tilde{n} \sim \text{Binomial}(n, p)$. We let $\tilde{\boldsymbol{y}} \in \mathbb{R}^{\tilde{n}}$ be the vector of these target values, and $\boldsymbol{X}_{\text{sub}} \in \mathbb{R}^{\tilde{n} \times d}$ be the rows of $\boldsymbol{X}$ corresponding to these $\tilde{n}$ instances. By Hoeffding's concentration inequality, we have $\tilde{n}/n = p + o(1)$ with high probability. Since we only deal with the proportional regime, it suffices to assume the case of $\tilde{n}/n = p$ for establishing asymptotic risk quantities in our framework.

The observed covariates are a randomly masked version of $\boldsymbol{X}_{\text{sub}}$. Let $\boldsymbol{Z} \in \{0,1\}^{\tilde{n} \times d}$ be a random matrix where each entry $z_{ij}$ is i.i.d. Bernoulli$(1-p)$, with $p$ the masking probability defined before. Then the covariate matrix is $\tilde{\boldsymbol{X}} = \boldsymbol{X}_{\text{sub}} \odot \boldsymbol{Z} \in \mathbb{R}^{\tilde{n} \times d}$. We consider the solution of the following **ridge-less** regression in the proportional regime ($d, \tilde{n} \to \infty$, with $d/\tilde{n} \to \tilde{\gamma} \in (0, \infty)$ constant):

$$\hat{\boldsymbol{\beta}} = \lim_{\lambda \to 0^+} \arg\min_{\boldsymbol{\beta}'} \left( \|\tilde{\boldsymbol{y}} - \tilde{\boldsymbol{X}} \boldsymbol{\beta}'\|_2^2 + \lambda \|\boldsymbol{\beta}'\|_2^2 \right) = \lim_{\lambda \to 0^+} (\tilde{\boldsymbol{X}}^\top \tilde{\boldsymbol{X}} + \lambda \boldsymbol{I}_d)^{-1} \tilde{\boldsymbol{X}}^\top \tilde{\boldsymbol{y}}. \quad (4)$$

We are interested in the test risk of the model. For a new, unmasked sample $\boldsymbol{x}_0 \in \mathbb{R}^d$, it is of form:

$$R_{\tilde{\boldsymbol{X}}}(\hat{\boldsymbol{\beta}}; \boldsymbol{\beta}) = \mathbb{E}\left[ (\boldsymbol{x}_0 \hat{\boldsymbol{\beta}} - \boldsymbol{x}_0 \boldsymbol{\beta})^2 \mid \tilde{\boldsymbol{X}} \right] = \mathbb{E}\left[ \|\hat{\boldsymbol{\beta}} - \boldsymbol{\beta}\|_{\boldsymbol{\Sigma}}^2 \mid \tilde{\boldsymbol{X}} \right]. \quad (5)$$

**Relation with standard ridge-less regression framework.** Our setup diverges from the standard ridge-less regression framework in two key aspects. First, the effective number of training samples, $\tilde{n} = np$, is directly modulated by $p$. Second, the design matrix $\tilde{\boldsymbol{X}}$ exhibits a level of induced sparsity (or feature corruption) determined by $p$. As we will demonstrate, these two $p$-dependent factors lead to complex and distinct behaviors in the bias and variance of the estimator, compared to classical ridge-less regression. Further background and preliminaries are provided in Appendix A.2.

**Test risk and model performance.** The test risk for reconstructing a feature $\boldsymbol{y}$ can be viewed as a feature-wise generalization error, analogous to validation loss. Here, $\boldsymbol{y}$ can represent latent features, whose reconstruction error is connected to the validation loss in the original space through the model's decoding transformation. Therefore, this risk reflects the model's feature learning ability, which indicates its utility for downstream tasks like probing and fine-tuning. The correlation between MAE validation loss and fine-tuning performance, as noted in [8], supports this interpretation.

**Relation with real network optimization.** Beyond the key linear simplification, complexities such as mini-batch processing and multi-epoch training are not incorporated into our current setup. Our goal in this study is to develop a minimal model that captures essential aspects of mask pretraining behaviors. A more detailed characterization of these additional factors remains future research.

**Next token prediction.** Our linear model addresses an independent sample-wise prediction setting. While it aligns well with the mask-based pretraining task, it cannot adequately model the other prevalent pretraining procedure, i.e., autoregression, which is a token-wise prediction task with strong contextual dependencies. We anticipate the latter task to exhibit distinct statistical behaviors, which may be revealed through the analysis of a more complex high-dimensional linear model.

### 3.2 Isotropic model

Here we present our main theoretical results regarding the test risk of the considered high-dimensional linear model. We first consider the simplest case where the covariance matrix $\boldsymbol{\Sigma} = \boldsymbol{I}$.

**Theorem 1** (Isotropic model). *When $\boldsymbol{\Sigma} = \boldsymbol{I}$, the test risk (5) can be asymptotically expressed as:*

$$\lim_{n,d \to \infty} R_{\tilde{\boldsymbol{X}}}(\hat{\boldsymbol{\beta}}; \boldsymbol{\beta})/r^2 = \begin{cases} \frac{(p+\kappa)\gamma}{(1-p)(p-\gamma)}, & \tilde{\gamma} < 1 \ (\gamma < p); \\ 1 - \frac{p}{\gamma} + \frac{p(p+\kappa)}{(1-p)(\gamma-p)}, & \tilde{\gamma} > 1 \ (\gamma > p). \end{cases} \quad (6)$$

The proof of the theorem, provided in Appendix B.2, extends well-known results from standard high-dimensional linear regression [43] by employing an isotropic local law for sample covariance matrices [30]. According to the formula, in the underparameterized regime ($\tilde{\gamma} < 1$), the test risk is a monotonically increasing function of $p$. In the overparameterized regime ($\tilde{\gamma} > 1$), the test risk exhibits non-monotonic behavior with respect to $p$, achieving its minimum at some $p^* \in (0, 1)$. Depending on the value of $\gamma$, the test risk curve will either monotonically increase regarding $p$ ($\gamma > 1$), or exhibit a transition at the threshold at $\gamma = p$ ($\gamma < 1$). These predictions, including the phase transition phenomenon, are validated by simulations as shown in Fig. 1A.

Nevertheless, this outcome is largely unconstructive, as the minimal risk achieved does not offer a substantial reduction compared to that of a null prediction (i.e., $\hat{\boldsymbol{\beta}} = \boldsymbol{0}$, for which $R_x(\boldsymbol{0}; \boldsymbol{\beta}) = r^2$). In the following sections, we will demonstrate that the benefit of masking is due to the conditional dependence between unmasked and masked features, a key component missing in this setting.

### 3.3 Spiked covariance model

Identity covariance represents a special case without feature dependency, whose characterization effectively reduces to the standard ridge-less case. If the covariance involves interaction terms, a non-trivial standalone treatment would be required. Below, we consider a spiked covariance model, $\boldsymbol{\Sigma} = \boldsymbol{I} + \delta \boldsymbol{v}\boldsymbol{v}^\top$, where $\boldsymbol{v} \in \mathbb{R}^d$ is a vector and $\delta > 0$ is a scalar. Below we denote the masked data covariance as $\tilde{\boldsymbol{\Sigma}} = (1-p)^2\boldsymbol{\Sigma} + p(1-p)\operatorname{diag}(\boldsymbol{\Sigma})$. We characterize the limiting test risk of this rank-1 spiked covariance model in the overparametrized regime:

**Corollary 1** (Limiting test risk of spiked covariance model). *The test risk* (5) *has the following limit:*

$$\lim_{n \to \infty} \frac{R_{\tilde{\boldsymbol{X}}}(\hat{\boldsymbol{\beta}}; \boldsymbol{\beta})}{r^2} \to \lim_{n \to \infty} \left( \phi_\beta + c^2(1 - \phi_v) + \delta(c(1 - \phi_v) - \psi)^2 + u\left(\frac{\sigma^2}{r^2} + p + cp\tilde{\boldsymbol{\beta}}^\top \boldsymbol{v}\right) \right),$$
(7)

$$\text{where} \quad c = \frac{p\delta \cdot \boldsymbol{v}^\top \tilde{\boldsymbol{\beta}}}{1 + \delta(1-p)}, \quad \phi_\beta = \lambda_\star \tilde{\boldsymbol{\beta}}^\top (\lambda_\star \boldsymbol{I} + \tilde{\boldsymbol{\Sigma}})^{-1}\tilde{\boldsymbol{\beta}}, \quad \phi_v = \lambda_\star \boldsymbol{v}^\top (\lambda_\star \boldsymbol{I} + \tilde{\boldsymbol{\Sigma}})^{-1}\boldsymbol{v},$$
(8)
$$\psi = \lambda_\star \tilde{\boldsymbol{\beta}}^\top (\lambda_\star \boldsymbol{I} + \tilde{\boldsymbol{\Sigma}})^{-1}\boldsymbol{v}, \quad u = \frac{\operatorname{Tr}(\boldsymbol{\Sigma}\tilde{\boldsymbol{\Sigma}}(\lambda_\star \boldsymbol{I} + \tilde{\boldsymbol{\Sigma}})^{-2})}{\tilde{n} - \operatorname{Tr}(\tilde{\boldsymbol{\Sigma}}^2(\lambda_\star \boldsymbol{I} + \tilde{\boldsymbol{\Sigma}})^{-2})},$$

*and $\lambda_\star$ is the unique non-negative solution of the fixed point equation $\tilde{n} = \operatorname{Tr}\left(\tilde{\boldsymbol{\Sigma}}(\tilde{\boldsymbol{\Sigma}} + \lambda_\star \boldsymbol{I})^{-1}\right)$.*

The result is a corollary of Theorem 2, which characterizes the asymptotics of the test risk in this setup. Theorem 2 and its proof are presented in Appendix B.3, along with a moderate delocalization assumption required for the proof. The validity of our derived test risk expression is confirmed by numerical experiments (Fig. 1B). Intuitively, when the spike level $\delta$ is small, the setting effectively reduces to the identity covariance case, where the test risk does not significantly descend. When $\delta$ is large, the behavior of the bias term is mostly characterized by the quadratic term $\delta(c(1 - \phi_v) - \psi)^2$. In this case, there can exist a "sweet-spot" masking ratio that minimizes the quadratic term achieving near-zero bias and near-optimal risk, especially when $\delta$ is large. This yields the desired descent behavior in real-world mask pretraining curves and is validated via simulations (Figs. 1C, 3).

According to the quadratic term, the test risk and the optimal masking ratio both depend on the feature strength, defined as the alignment between $\boldsymbol{\beta}$ and $\boldsymbol{\Sigma}$ (reducing to $\boldsymbol{\beta}^\top \boldsymbol{v}$ in this case). A stronger feature strength results in a steeper test risk descent and a higher optimal masking ratio (Figs. 1B-C, 3). Finally, we empirically observed that higher masking leads to a greater disparity in prediction magnitude, $\mathbb{E}\left[\|\boldsymbol{X}\hat{\boldsymbol{\beta}}\|^2 | \tilde{\boldsymbol{X}}\right]$, between $\boldsymbol{\beta}$s aligned with $\boldsymbol{\Sigma}$ and those that are not (Figs. 1D, 3).

### 3.4 General covariance models recapitulate real-world mask pretraining curves

For general covariance matrices, an analytic expression of test risk remains infeasible. Nevertheless, when $\boldsymbol{\beta}$ is an eigenvector of $\boldsymbol{\Sigma}$, the behavior of bias and variance terms can be revealed through a simplified form of the test risk, presented as Theorem 3 in Appendix B.4. Similar to the spiked covariance case, the test risk displays a descent with respect to the masking ratio $p$ due to cancellation in the bias term. To verify our results, we simulated covariance models constructed by orthonormal projections of various spectrum distributions (Fig. 1E, see Appendix for details). The non-monotonic pattern of the test risk emerges in all models, with stronger effects and higher optimal masking ratios for stronger signal $\boldsymbol{\beta}$s (those corresponding to higher eigenvalues in $\boldsymbol{\Sigma}$, Fig. 1E). We also observed a comparable transition threshold where the minimum test risk gains an advantage over null prediction (Fig. 1E), which may suggest a form of universality that warrants further theoretical investigation.

We further compared our results with real language models (BERT) pretrained by MLM with different masking ratios [29]. Even for the same set of models, the behavior of mask pretraining curves varies with respect to the evaluation dataset (Fig. 1F). The resulting family of curves aligns well with our observations in linear models (Fig. 1E). In vision MAE models [6], the mask pretraining curve can exhibit unusual behavior with two plateaus: 1) Before the performance improves with respect to masking ratio, the model performance remains relatively stable; 2) A range of masking ratios where the model achieves similar near-optimal. Interestingly, with another latent space model (see Appendix for details), we faithfully reproduced the observed two plateaus in real mask pretraining curves (Fig. 1G). Notably, another sample from the model results in a different curve aligning with MAE linear probing performance (Fig. 4). Together, these comparisons suggest a connection between real-world mask pretraining and our linear model framework, which we will further validate in the next section.

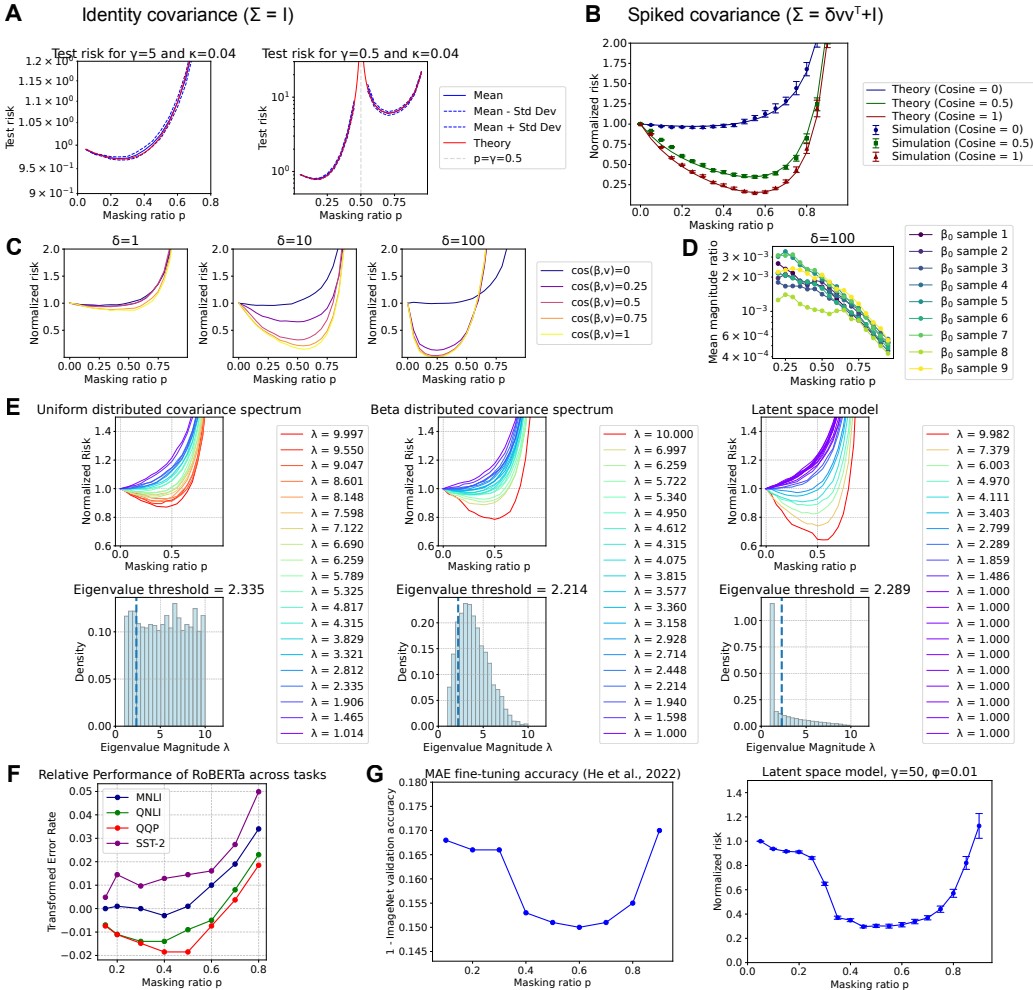

Figure 1: **A-B.** Plots of theoretical test risk and simulations (showing mean and standard deviation from 50 samples) against the masking ratio $p$ for the identity covariance model $\boldsymbol{\Sigma} = \boldsymbol{I}$ (**A**) and the spiked covariance model $\boldsymbol{\Sigma} = \delta \boldsymbol{v}\boldsymbol{v}^\top + \boldsymbol{I}$ (**B**). For the former model, $n = 2000$ (left), $4000$ (right). For the latter model, each entry in $\boldsymbol{v}$ is i.i.d. sampled from $\mathcal{U}(0, 1)$ and then scaled to ensure $\|\boldsymbol{v}\| = 1$. $n = 200, \gamma = 5, \delta = 10$. **C-D.** Plots of mean simulation test risk (**C**) and mean magnitude ratio (**D**, defined as $\mathbb{E}\big[\|\boldsymbol{X}\hat{\boldsymbol{\beta}}_0\|^2 | \tilde{\boldsymbol{X}}\big] / \mathbb{E}\big[\|\boldsymbol{X}\hat{\boldsymbol{\beta}}_1\|^2 | \tilde{\boldsymbol{X}}\big]$ between $\cos(\boldsymbol{\beta}_1, \boldsymbol{v}) = 1$ and $\cos(\boldsymbol{\beta}_0, \boldsymbol{v}) = 0$) over 50 samples in the spiked covariance model. $n = 200, \gamma = 5$. **E.** Normalized test risk of different covariance models plotted against masking ratio $p$, where $\boldsymbol{\beta}$ was selected as different eigenvectors of the covariance matrix $\boldsymbol{\Sigma}$ (Upper). Histogram of covariance spectrum densities for each model above (Lower). The threshold where the minimal risk becomes smaller than the null risk is highlighted by a dashed blue line. **F.** Transformed error rates of fine-tuned BERT models evaluated on different benchmark tasks [29] (See Appendix C for details). **G.** Plots of MAE fine-tuning accuracy on ImageNet-1K [6] and the normalized test risk of a latent space model against masking ratio.

## 3.5  Validating insights from linear models in real neural networks

Apart from reproducing existing observations, a successful theory should also provide hypotheses that can be empirically validated. Here we summarize main predictions from our theoretical framework:

1. Mask-based pretraining is only beneficial in the overparametrized regime. This is because it reduces risk through the bias term, which only appears in the overparametrized case. Moreover, for these overparametrized models, the optimal masking ratio should be dependent on the model parameter size, which determines the limit ratio $\gamma$ thus also the test risk.

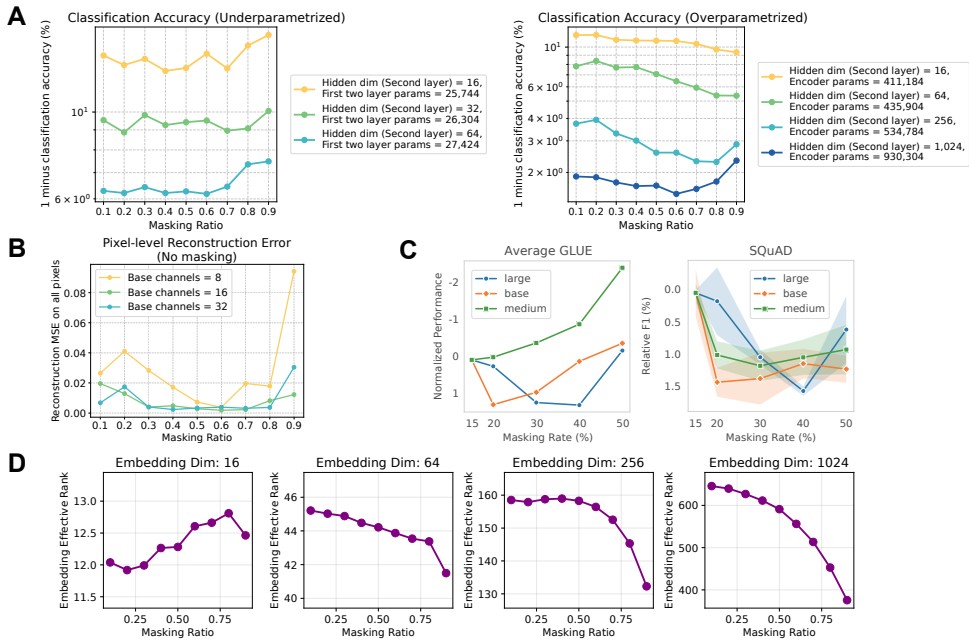

Figure 2: **A.** Linear probing classification accuracy of MLPs in parameter-insufficient (left) and sufficient (right) settings on MNIST. **B.** Pixel-level reconstruction error without masking for CNN models trained on CelebA. **C.** Impact of masking ratio on different RoBERTa model sizes (large > base > medium). Adapted from [29] licensed CC-BY 4.0. y axes were flipped for consistency. **D.** Effective rank of MNIST embedding in overparametrized MLP models of different settings.

    2. The performance curve regarding the mask ratio can differ by evaluation tasks even for the same set of pretrained models, due to different features required for the downstream task.

The most decisive support of our theory would be on the first point that cannot be explained via previous arguments centered on training data [25, 27, 28]. We validate these points on MultiLayer Perceptrons (MLPs) trained on MNIST, convolutional neural networks (CNNs) trained on CelebA, and large-scale RoBERTa transformer models [29]. For the former two setups, we pretrained encoder-decoder architectures by pixel-level mask reconstruction tasks. We refer to extensive comparisons performed in [29] for effects of mask ratio and RoBERTa model size on pretraining performance. We implemented both parameter-insufficient and sufficient settings for MNIST, whereas the latter was used for evaluating CNNs and transformers. In MLPs, the linear probing error rate exhibits a descent for all parameter-sufficient models, while the error rate first fluctuates then monotonically increases for parameter-insufficient models (Fig. 2A). The transition observed in parameter-insufficient models can be explained by the test risk of underparametrized linear models ($\gamma < 1$, Fig. 1A).

For CNNs, the optimal reconstruction of original images was observed for different sets of intermediate masking ratios across model sizes (Fig. 2B). As for linear probing, all models suddenly improve after the masking ratio increases to a model-size-specific threshold (Figs. 5-7). Together, different CNN model sizes exhibit distinct optimal masking ratios (0.6, 0.7, 0.8 for 8, 16, 32 base channels respectively). For RoBERTa, larger models correspond to higher optimal masking ratio, which is further altered by the evaluation task (Fig. 2C). These evaluations provide strong support for our first prediction, which would not be addressed by existing explanations. Differences of optimal masking ratio across evaluation tasks for CNNs and RoBERTa models further support the second point.

We then explored whether the increased feature magnitude disparity in spiked covariance models (Fig. 1D) appears in real neural networks. Specifically, we evaluated the effective rank (ER) of MNIST image embeddings in MLP models. ER is defined as the entropy of sum-normalized matrix singular values and measures spectrum uniformity [51]. Except for the extremely small embedding case (dim=16), all parameter-sufficient models indeed exhibit a decrease of ER with respect to the masking ratio (Fig. 2D), with curve patterns resembling those in Fig. 1D, confirming our hypothesis. This also aligns with the previously observed decrease of ER during training in vision MAEs [25].

# 4 R²MAE for universal representation learning

As a final contribution of our work, we aim to employ our gained understanding to improve current mask pretraining schemes. Our theoretical framework highlights that different masking ratios selectively emphasize features of varying strength. Therefore, we conclude that it is essential to expose the model to a range of masking ratios during pretraining. We propose the simplest pretraining method that serves the purpose, which can be described and implemented in one line:

*Expose the model to data corrupted with a uniformly sampled masking ratio $p \sim \mathcal{U}(p_{\min}, p_{\max})$.*

We term this scheme as *Randomly Random Mask AutoEncoding* (**R²MAE**). Despite its simplicity, it has not been implemented in prior works to our knowledge. Existing works focused on improving the mask-based pretraining objective mostly aim to learn adaptive masks during pretraining [38, 37, 52] or perform (deterministic) mask rate scheduling during training [33, 34]. Technically, the closest variant of R²MAE may be the training phase of a mask diffusion language model (MDLM) [53, 54], which reconstructs tokens in unmasked to completely masked samples, constituting a special case of $(p_{\min}, p_{\max}) = (0, 1)$. Nevertheless, MDLMs are used for generation instead of fine-tuning related tasks, and fine-tuning standard BERT models with MDLM does not affect/improve downstream task performance [54]. The issue of setting $(p_{\min}, p_{\max}) = (0, 1)$ for feature learning is apparent with our theoretical framework, as the test risk either degenerates or explodes when $p \approx 0/1$.

## 4.1 Evaluation of R²MAE on vision and language modeling

We first evaluated R²MAE on well-studied image and language pretraining tasks. Our implementations closely follow established practices [6, 29]. For vision pretraining, we implemented different mask ratio settings on the same ViT-base MAE model as in [6]. The considered settings include: 1. Default MAE with constant masking ratio 0.75; 2. R²MAE with masking rate $p \sim \mathcal{U}(0.6, 0.9)$; 3. the training phase of MDLM [54] with masking rate $p \sim \mathcal{U}(0, 1)$; 4. Dynamic MR [34] that linearly decreases masking ratio from 0.9 to 0.6; 4. High (0.9) and low (0.5) mask ratios. We trained all models for 150 epochs. While the training is shorter than default 800-epoch experiments in [6], their evaluation shows predictable improvements from 100 to 1600 epochs in ViT-Large models. Therefore, we anticipate our results to be comparable across different settings despite suboptimal absolute accuracy. All models were then fine-tuned for classification for 100 epochs following [6].

As shown in Table 1, R²MAE marginally outperforms the best alternatives (default MAE and dynamic MR) and does not suffer from suboptimal MR as observed in high and low masking baselines. Across our experiments, R²MAE yields its smallest improvement for ViT-MAE, potentially for two reasons: 1) Its training involves significantly longer epochs with augmentation, which deviates from other experimental settings and our theoretical framework; 2) R²MAE's pre-training loss in ViT-MAE fluctuates, likely due to variable-length of unmasked token sequences, warranting future improvement.

Table 1: Fine-tuning accuracies of ViT-base MAE models [6] on ImageNet classification. In our benchmarks, masking scheme metrics outperforming optimal fixed MR settings are labeled red.

| Metric | MR 0.75 (default) | MR 0.9 | MR 0.5 | MDLM | Dynamic MR | R²MAE (Ours) |
|---|---|---|---|---|---|---|
| Top1 Acc. | 81.97 | 81.20 | 81.80 | 81.02 | 81.97 | 82.00 |
| Top5 Acc. | 96.02 | 95.68 | 95.93 | 95.60 | 96.04 | 96.05 |

For language modeling, we trained RoBERTa-base and RoBERTa-medium (named following [29]) models on the FineWeb dataset for 10B tokens, and fine-tuned them on GLUE benchmarks. The reported accuracy for each task is the average of three runs with different random seeds, consistent with [29]. Similar to vision experiments, we evaluated: 1. Default MLM (MR 0.15); 2. R²MAE ($p \sim \mathcal{U}(0.15, 0.4)$); 3. Dynamic MR [34] (0.4 to 0.15); 4. MLM with a fixed 0.4 MR. Our fine-tuning accuracies are comparable to those in [29]. In both models, R²MAE achieves best performance in three tasks (MNLI, QQP, SST-2), achieving best overall rank, followed by dynamic MR (Table 2).

## 4.2 Evaluation of R²MAE on DNA sequence and gene expression modeling

One focus of R²MAE is on biological data including DNA sequences and single-cell gene expression data, where the standard mask-based pretraining scheme remains the prevalent choice [16–18, 21],

Table 2: GLUE fine-tuning accuracies of RoBERTa models with different pretraining settings.

| Method | RoBERTa-Medium (52M) | | | | | RoBERTa-Base (125M) | | | | |
|---|---|---|---|---|---|---|---|---|---|---|
| | MNLI | QQP | SST-2 | QNLI | Rank | MNLI | QQP | SST-2 | QNLI | Rank |
| MLM default | 80.8 | 89.8 | 89.9 | 86.3 | 3.25 | 81.5 | 90.7 | 91.7 | 87.8 | 3.00 |
| Fixed MR 0.4 | 80.3 | 89.7 | 90.1 | 86.6 | 3.75 | 81.7 | 90.7 | 91.2 | 88.5 | 3.25 |
| MDLM | 79.4 | 89.8 | 89.3 | 85.4 | 4.50 | 80.3 | 90.3 | 91.4 | 87.7 | 4.50 |
| Dynamic MR | 80.8 | 90.1 | 90.5 | 87.1 | 1.50 | 81.8 | 90.7 | 91.4 | 89.1 | 2.00 |
| $R^2$MAE (Ours) | 80.9 | 90.1 | 90.6 | 86.7 | 1.25 | 81.9 | 90.8 | 91.9 | 88.6 | 1.25 |

and improving the scheme is a pressing need. We evaluated a 12-layer BERT style model for DNA sequence (GPN-MSA [16]), and a 5-layer MLP encoder-decoder model for single-cell gene expression respectively. Apart from $R^2$MAE, we implemented standard MLM/MAE with different masking ratios, MDLM [54], dynamic MR [34], and learnable mask (named as CL-MAE following [38], which effectively covers AutoMAE [52]). We also compared alternative DNA sequence and single-cell models [55–59, 39, 17, 21]. To evaluate if other masking strategies synergize with $R^2$MAE, we further implemented the combination of $R^2$MAE with Dynamic MR or CL-MAE. After training, DNA models are evaluated through zero-shot missense/regulatory (Clinvar/OMIM) variant prediction tasks [60–62]. Gene expression models are evaluated using linear probing performances in predicting cell type, disease, and age across donors in lung and brain atlas datasets [63, 64].

As shown in Tables 3–4 and 6, $R^2$MAE achieves the best overall performance in both DNA and single-cell tasks. The only tasks without clear advantage are DNA missense variant prediction (where all best models achieved near-optimal performance) and cell type classification (where the target label is artificially curated). Together, among all tested model domains (vision, language, DNA, single-cell), $R^2$MAE is **the only scheme** that consistently outperforms standard MLM/MAE with best mask ratios, among the default value and min/max ratios used in $R^2$MAE. The consistent improvement in our well-controlled comparisons highlights robustness and generalizability of $R^2$MAE.

Interestingly, for the cases where Dynamic MR and CL-MAE outperform the baseline setting, combining them with $R^2$MAE results in a disadvantage compared to $R^2$MAE alone. For better understanding, we inspected specific DNA sequence classes with different prediction performances. The combination of $R^2$MAE with CL improves classification of harder variants including 3'UTR and ncRNA, but not the easier 5'UTR variants (Table 3). These results demonstrate that combining $R^2$MAE with additional designs may bring advantages in certain cases but not overall improvement.

Table 3: Comparison on DNA variant effect prediction. pAUROC, partial AUROC.

| Methods | Clinvar (Missense) | | OMIM (Regulatory) | | OMIM subset class AUPRC | | |
|---|---|---|---|---|---|---|---|
| | AUROC | AUPRC | pAUROC | AUPRC | 5'UTR | 3'UTR | ncRNA |
| NT | 0.601 | 0.652 | 0.500 | 0.001 | 0.010 | 0.001 | 0.000 |
| phastCons-100v | 0.883 | 0.848 | 0.514 | 0.006 | 0.081 | 0.005 | 0.005 |
| phyloP-241m | 0.912 | 0.913 | 0.590 | 0.028 | 0.175 | 0.015 | 0.028 |
| phyloP-100v | 0.927 | 0.937 | 0.574 | 0.038 | 0.251 | 0.029 | 0.039 |
| CADD | 0.966 | 0.967 | 0.595 | 0.048 | 0.279 | 0.010 | 0.090 |
| GPN-MSA (MLM) | 0.970 | 0.974 | 0.644 | 0.127 | 0.331 | 0.044 | 0.102 |
| – MR 5% | 0.967 | 0.970 | 0.647 | 0.134 | 0.330 | 0.048 | 0.171 |
| – MR 30% | 0.970 | 0.974 | 0.645 | 0.131 | 0.335 | 0.047 | 0.081 |
| MDLM | 0.970 | 0.974 | 0.647 | 0.131 | **0.341** | 0.048 | 0.110 |
| Dynamic MR | 0.970 | 0.973 | 0.645 | 0.132 | 0.332 | 0.054 | 0.082 |
| CL-MAE | 0.968 | 0.972 | 0.644 | 0.117 | 0.328 | 0.056 | 0.128 |
| $R^2$MAE (Ours) | 0.969 | 0.973 | 0.649 | 0.148 | 0.339 | 0.050 | 0.136 |
| + Dynamic MR | 0.970 | 0.974 | 0.649 | 0.138 | 0.324 | 0.045 | 0.106 |
| + CL | 0.967 | 0.971 | 0.643 | 0.139 | 0.330 | **0.058** | 0.192 |
| + CL ($k = 0$) | 0.965 | 0.969 | 0.649 | 0.140 | 0.325 | 0.051 | **0.208** |

Table 4: Comparison for different single-cell models trained on brain SEA-AD dataset.

| Methods | Cell state | | Alzheimers AUROC | | Age Spearman $r$ | | Avg performance | |
|---|---|---|---|---|---|---|---|---|
| | BAcc. | $F1_{macro}$ | Cell | Donor | Cell | Donor | Score | Rank |
| Normalized exp. | 0.798 | 0.738 | 0.571 | 0.611 | 0.129 | 0.511 | 0.560 | 9.00 |
| scGPT | 0.784 | 0.693 | 0.549 | 0.556 | 0.065 | 0.272 | 0.486 | 12.67 |
| scVI | 0.826 | **0.740** | 0.631 | **0.731** | 0.201 | 0.502 | 0.605 | 7.00 |
| MAE (MR 25%) | **0.841** | 0.737 | 0.667 | 0.699 | 0.483 | 0.575 | 0.667 | 4.33 |
| $-$ MR 10% | 0.837 | 0.726 | 0.543 | 0.536 | 0.449 | 0.399 | 0.582 | 10.67 |
| $-$ MR 50% | 0.839 | 0.738 | 0.574 | 0.591 | 0.516 | 0.536 | 0.632 | 6.17 |
| MDLM | 0.831 | 0.719 | 0.667 | 0.686 | 0.543 | 0.622 | 0.678 | 6.83 |
| Dynamic MR | 0.838 | 0.735 | 0.662 | 0.694 | 0.444 | 0.446 | 0.636 | 7.50 |
| CL-MAE | 0.838 | 0.729 | 0.687 | 0.722 | 0.462 | 0.484 | 0.654 | 5.83 |
| **$R^2$MAE** (Ours) | 0.840 | 0.737 | 0.687 | 0.716 | 0.572 | 0.628 | **0.697** | **2.17** |
| + Dynamic MR | 0.834 | 0.735 | 0.642 | 0.682 | 0.551 | 0.545 | 0.665 | 6.67 |
| + CL | 0.836 | 0.730 | 0.684 | 0.719 | 0.570 | 0.559 | 0.683 | 4.83 |
| + CL ($k = 0$) | 0.837 | 0.734 | 0.676 | 0.707 | 0.511 | 0.506 | 0.662 | 6.17 |

### 4.3 $R^2$MAE enforces learning multi-scale features and can outperform optimal MR

We further investigated the mechanism underlying the improvement of $R^2$MAE. On real single-cell data, $R^2$MAE achieves near-optimal reconstruction performance across its entire masking range, whereas models trained with a single, fixed MR are effective only within a narrower range (Tables 7–8). This observation aligns with the intuition that $R^2$MAE enforces learning multi-scale features, thereby enhancing downstream task performance. Intriguingly, at low masking ratios (e.g., 10%), $R^2$MAE can even outperform a model trained specifically at that fixed MR on the reconstruction task. In our linear model framework, we found that with appropriate $(p_{min}, p_{max})$ settings, $R^2$MAE can surpass optimal fixed MR in terms of test risk across different covariance settings in most cases, even when the optimal MR is mildly misaligned with $R^2$MAE masking range (Tables 5,9). These findings suggest additional beneficial properties of $R^2$MAE that warrant future theoretical research.

Table 5: Normalized test risk of $R^2$MAE (MR range 0.5-0.6) against optimal fixed MR and mean MR settings across different random seeds for Beta covariance and latent space models. The ground truth signal $\beta$ is set to be the first eigenvector of covariance $\Sigma$ in all cases. $n = 200$, $\gamma = 5$.

| | Beta Covariance Model | | | | Latent Space Model | | | |
|---|---|---|---|---|---|---|---|---|
| Seed | Best MR | Min Risk | MR 55% | **$R^2$MAE** | Best MR | Min Risk | MR 55% | **$R^2$MAE** |
| 2 | 0.55 | 0.520 | 0.520 | **0.504** | 0.55 | 0.611 | 0.611 | **0.599** |
| 12 | 0.53 | 0.606 | 0.612 | **0.599** | 0.65 | **0.641** | 0.673 | 0.662 |
| 22 | 0.58 | 0.626 | 0.631 | **0.615** | 0.36 | 0.662 | 0.667 | **0.653** |
| 32 | 0.51 | **0.543** | 0.563 | 0.549 | 0.59 | 0.640 | 0.659 | **0.634** |
| 42 | 0.53 | 0.649 | 0.658 | **0.645** | 0.38 | 0.640 | 0.643 | **0.629** |

## 5 Conclusions

In this work, we introduced and analyzed a theoretical framework to elucidate mask-based pretraining in large-scale deep learning models. Motivated by this framework, we propose an extremely simple approach **$R^2$MAE**, which is shown to improve upon state-of-the-art self-supervised image, vision, DNA sequence, and single-cell models by solely modifying the pretraining objective.

**Limitations.** Explicit characterization of the test risk in more complex model settings (e.g., $R^2$MAE) requires new analysis tools and remains a direction for future research. Potential improvements of $R^2$MAE with dedicated domain-specific designs also remains to be explored.

**Broader impact.** We envision that our theoretical framework will serve as a basis for better understanding self-supervised pretraining, one of the most important components in modern deep learning and foundation models. Furthermore, our work addresses a pressing need for building better models towards universal representations, with immediate impact for the (biological) AI community.

## Acknowledgements

The authors thank Theodor Misiakiewicz, Boris Landa and the anonymous reviewers for helpful discussions and feedback. Y.K. acknowledges support by NIH grants U54AG076043, U54AG079759, P50CA121974, R01GM131642, UM1DA051410, and U01DA053628.

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

# Appendix Contents

## A  Additional text

### A.1  Further discussions on alternative mask pretraining schemes

Approaches to improving mask-based pretraining can be broadly divided into two categories. The first category focuses on refining the masking scheme itself, i.e., optimizing the selection of pixels/tokens to be masked to maximize pretraining efficacy or downstream performance. Numerous efforts have explored this direction [35, 3, 36, 65]. A number of these schemes assume specific data structures, such as sequential information in text, and thus may not readily generalize across all data domains. A prominent recent direction in this category involves learning the masks themselves during training, for instance, by optimizing them to enhance the pretraining objective or, conversely, to adversarially challenge it [52, 38]. Apart from these general enhancement strategies, several works specifically design masking procedures to emphasize specific downstream tasks [66–68].

Wettig et al. [29] conducted an extensive evaluation of different masking strategies for BERT masked language models, including a number of those cited above. Their findings highlight that while the optimal masking ratio might vary across strategies, simple uniform random masking often suffices to achieve peak performance. In the single-cell genomics context, Richter et al. [24] evaluated various structured masking schemes (e.g., gene program and transcriptional factor-based masking) against uniform masking, all at a fixed masking ratio. They observed no consistent overall advantage for the more complex, structured masking schemes over uniform random masking. These results from both language and genomics domains align, suggesting that highly sophisticated, domain-specific masking strategies may not always be necessary for effective pretraining. The comparable performance achieved by different masking schemes may serve as a support for the general applicability of our theoretical framework, which is established based on uniform masking.

The second category of approaches focuses on altering the masking rate. Wettig et al. [29] observed that higher masking ratios generally boost masked language modeling performance, particularly for larger models. Ankner et al. [34] further demonstrated that a dynamic masking schedule, gradually reducing the masking rate from 40% to 15%, improves performance, while the reverse schedule does not. A key insight from the influential Masked Autoencoders (MAE) work [6] is that an extremely high masking rate for images can force the model to learn robust and generalizable representations through the reconstruction task, leading to improved downstream performance. Notably, our theoretical

framework highlights a potential limitation of existing mask pretraining schemes: employing a single, static masking strategy—whether it involves carefully designed masking pattern or masking ratio—may not be sufficient to optimally capture the diverse spectrum of features present in data.

## A.2 Background on high-dimensional linear regression

The major theoretical focus in this work is the linear model

$$\boldsymbol{y} = \boldsymbol{X}\boldsymbol{\beta} + \boldsymbol{\varepsilon}, \qquad \boldsymbol{\varepsilon} \sim N(0, \sigma^2 \boldsymbol{I}_n),$$

where $\boldsymbol{X} \in \mathbb{R}^{n \times d}$ is the design matrix and $\boldsymbol{\beta} \in \mathbb{R}^d$ the true parameter. For an estimator $\hat{\boldsymbol{\beta}}$, the out-of-sample prediction risk at a fresh covariate $\boldsymbol{x}_0$ admits the bias–variance decomposition

$$R_{\boldsymbol{X}}(\hat{\boldsymbol{\beta}}; \boldsymbol{\beta}) = \underbrace{(\mathbb{E}[\hat{\boldsymbol{\beta}}|\boldsymbol{X}] - \boldsymbol{\beta})^T \boldsymbol{\Sigma} (\mathbb{E}[\hat{\boldsymbol{\beta}}|\boldsymbol{X}] - \boldsymbol{\beta})}_{\text{Bias}^2} + \underbrace{\text{Tr}\big[\text{Cov}(\hat{\boldsymbol{\beta}}|\boldsymbol{X})\boldsymbol{\Sigma}\big]}_{\text{Variance}},$$

where $\boldsymbol{\Sigma} = \mathbb{E}[\boldsymbol{x}_0 \boldsymbol{x}_0^T]$ is the population covariance.

In the proportional regime $d/n \to \gamma \in (0, \infty)$, the ridge regression estimator is of form

$$\hat{\boldsymbol{\beta}}_\lambda = (\boldsymbol{X}^T \boldsymbol{X} + \lambda \boldsymbol{I})^{-1} \boldsymbol{X}^T \boldsymbol{y}.$$

For fixed $\boldsymbol{X}$, its bias and variance are

$$\text{Bias}^2 = \lambda^2 \boldsymbol{\beta}^T (\boldsymbol{X}^T \boldsymbol{X} + \lambda \boldsymbol{I})^{-1} \boldsymbol{\Sigma} (\boldsymbol{X}^T \boldsymbol{X} + \lambda \boldsymbol{I})^{-1} \boldsymbol{\beta},$$

$$\text{Variance} = \frac{\sigma^2}{n} \text{Tr}\big[\boldsymbol{\Sigma}(\boldsymbol{X}^T \boldsymbol{X} + \lambda \boldsymbol{I})^{-2} \boldsymbol{X}^T \boldsymbol{X}\big].$$

A streamlined way to capture asymptotics in proportional models is via the theory of deterministic equivalents [69, 32]. Two sequences of (possibly random) matrices $\boldsymbol{A}_n, \boldsymbol{B}_n \in \mathbb{R}^{n \times n}$ are declared asymptotically equivalent (denoted $\boldsymbol{A}_n \asymp \boldsymbol{B}_n$) if for every sequence $\boldsymbol{\Theta}_n$ bounded in trace norm,

$$\text{Tr}\big[\boldsymbol{\Theta}_n(\boldsymbol{A}_n - \boldsymbol{B}_n)\big] \longrightarrow 0, \quad n \to \infty.$$

Within this framework, Rubio and Mestre [70] showed that the resolvent of the sample covariance $(\hat{\boldsymbol{\Sigma}} - zI)^{-1}$ is equivalent to a deterministic matrix $(a_n \boldsymbol{\Sigma} - zI)^{-1}$, where $a_n$ solves an explicit fixed-point equation. Such equivalences yield precise control over traces of analytic functions of random matrices and underpin modern high-dimensional risk calculations. The limiting risk of high-dimensional ridge regression has been extensively studied in the past. As a representative result, the exact asymptotic risk has been established in [71]. We refer to [30, 31] for more results on the local law, which asserts the convergence of the resolvent entrywise under high probability bounds, at scales finer than the global limit.

In [43], the authors study the behavior of ridgeless least squares interpolation in high-dimensional settings, where the model interpolates the training data perfectly analogous to overparametrized neural networks. Surprisingly, they show that such interpolating solutions can theoretically generalize well under certain conditions. Their work derives exact asymptotic expressions for the bias and variance of minimum-norm interpolators in the overparameterized regime, using tools from random matrix theory and the theory of deterministic equivalents. The results serve as a basis for understanding neural network behavior from the lens of high-dimensional linear regression theory.

## A.3 Implementations of CL and R$^2$MAE + CL

Inspired by recent curriculum learning (CL) approaches in MAE [38, 52], which often involve an adversarial mask generator and an easy-to-hard progression (e.g., by scheduling a gradient coefficient $k$ for the mask generator [38]), we evaluated whether these approach improves self-supervised learning for our biological data settings. Given that our DNA sequence and single-cell gene expression models process entire input sequences rather than patches, we implemented CL by introducing learnable, positive coefficients that modulate the element-wise reconstruction loss for each feature. These coefficients are constrained to have a mean of one. We then applied a gradient scheduling mechanism to these loss coefficients: one setting used a constant gradient multiplier of 1 (termed the $k = 0$ setting, which learns an easy mask with the smallest loss value throughout pretraining), while another employed a dynamic multiplier decreasing from 1 to $-1$ to simulate an easy-to-hard progression.

Our initial evaluations showed that the $k = 0$ fixed curriculum led to severe learning degeneration and hampered performance (Tables 3, 5), which would be an anticipated outcome.

To address this and integrate these CL principles with our R$^2$MAE framework, we propose to randomly sample masking ratios from a predefined discrete set of $l$ values, e.g., [min_ratio, . . . , max_ratio]. For each of the $l$ discrete masking ratios, we learn an adaptive, positive weight vector $\boldsymbol{w}_j \in \mathbb{R}^d_+$ (where $d$ is the feature dimension), which form the columns of a weight matrix $\boldsymbol{W} \in \mathbb{R}^{d \times l}_+$. These weights dynamically adjust the importance of reconstructing each feature under that specific masking ratio $j$. To ensure balanced learning across features and masking ratios, we impose mass-conservation constraints on $\boldsymbol{W}$ such that:

$$\forall i \in \{1, \ldots, d\}, \sum_{j=1}^{l} \boldsymbol{W}_{ij} = l; \quad \forall j \in \{1, \ldots, l\}, \sum_{i=1}^{d} \boldsymbol{W}_{ij} = d, \tag{9}$$

This approach aims to provide, on average, the same learning signal magnitude per feature, while still allowing each masking ratio to prioritize different feature subsets. In practice, these constraints are efficiently enforced using a few iterations of the differentiable Sinkhorn algorithm on an initial positive matrix $\boldsymbol{W}_0$ [72]. $\boldsymbol{W}_0$ itself is generated by a small MLP applied to the full uncorrupted data features (for single-cell models) or the transformer's masked token representations (for DNA sequence models). Notably, this R$^2$MAE-CL approach effectively resolved the learning degeneration observed in the simpler $k = 0$ setting without requiring additional regularizations [38] (Tables 1–4,6).

## B  Additional theoretical results and proofs

In this section, with a slight abuse of notation, we denote $\boldsymbol{X}_{\text{sub}}$ as $\boldsymbol{X}$ for brevity.

### B.1  Statement and proof of technical lemmas

**Lemma 1** (Bias-variance decomposition for general covariance model). *The test risk* $R_{\tilde{\boldsymbol{X}}}(\hat{\boldsymbol{\beta}}, \boldsymbol{\beta}) :=$ $\mathbb{E}\left[\|\hat{\boldsymbol{\beta}} - \boldsymbol{\beta}\|_{\boldsymbol{\Sigma}}^2 | \tilde{\boldsymbol{X}}\right]$ *has the following decomposition* $R_{\tilde{\boldsymbol{X}}}(\hat{\boldsymbol{\beta}}, \boldsymbol{\beta}) = B_{\tilde{\boldsymbol{X}}}(\hat{\boldsymbol{\beta}}, \boldsymbol{\beta}) + V_{\tilde{\boldsymbol{X}}}(\hat{\boldsymbol{\beta}}, \boldsymbol{\beta})$ *outside a negligible set, which can be expressed as:*

$$B_{\tilde{\boldsymbol{X}}}(\hat{\boldsymbol{\beta}}, \boldsymbol{\beta}) = \|\tilde{\boldsymbol{\Pi}}\boldsymbol{\beta} + \tilde{\boldsymbol{X}}^+\boldsymbol{u}\|_{\boldsymbol{\Sigma}}^2, \tag{10}$$

$$V_{\tilde{\boldsymbol{X}}}(\hat{\boldsymbol{\beta}}, \boldsymbol{\beta}) = \sum_{i,j=1}^{d} \beta_i\beta_j \sum_{a=1}^{\tilde{n}}((\tilde{\boldsymbol{X}}^\top)^+\boldsymbol{\Sigma}\tilde{\boldsymbol{X}}^+)_{aa}w_a^{ij} + \sigma^2 \text{Tr}\left((\tilde{\boldsymbol{X}}^\top\tilde{\boldsymbol{X}})^+\boldsymbol{\Sigma}\right), \tag{11}$$

*where we denote the projection matrix* $\tilde{\boldsymbol{\Pi}} = \hat{\boldsymbol{\Sigma}}^+\hat{\boldsymbol{\Sigma}} - \boldsymbol{I}$, $\hat{\boldsymbol{\Sigma}} = \tilde{\boldsymbol{X}}^\top\tilde{\boldsymbol{X}}$, *as well as*

$$\mathcal{Z}_a = \{j \in [d]|\boldsymbol{Z}_{aj} = 1\}, \quad \mathcal{Z}_a^c = \{j \in [d]|\boldsymbol{Z}_{aj} = 0\}, \quad \forall a \in [\tilde{n}]; \tag{12}$$

$$\boldsymbol{u} \in \mathbb{R}^{\tilde{n}}, \quad u_i = \tilde{\boldsymbol{X}}_{i,\mathcal{Z}_i}\boldsymbol{\Sigma}_{\mathcal{Z}_i\mathcal{Z}_i}^{-1}\boldsymbol{\Sigma}_{\mathcal{Z}_i\mathcal{Z}_i^c}\boldsymbol{\beta}_{\mathcal{Z}_i^c}, \quad \forall i \in [\tilde{n}]; \tag{13}$$

$$w_a^{ij} = \mathbb{1}_{\tilde{\boldsymbol{X}}_{ai=0}, \tilde{\boldsymbol{X}}_{aj=0}}(\boldsymbol{\Sigma}_{ij} - \boldsymbol{\Sigma}_{i\mathcal{Z}_a}\boldsymbol{\Sigma}_{\mathcal{Z}_a\mathcal{Z}_a}^{-1}\boldsymbol{\Sigma}_{\mathcal{Z}_aj}), \quad \forall a \in [\tilde{n}], i, j \in [d]. \tag{14}$$

*Proof.* From the definition, our test error is given by

$$R_{\tilde{\boldsymbol{X}}}(\hat{\boldsymbol{\beta}}; \boldsymbol{\beta}) = \mathbb{E}\left[(\boldsymbol{x}_0^\top\hat{\boldsymbol{\beta}} - \boldsymbol{x}_0^\top\boldsymbol{\beta})^2 | \tilde{\boldsymbol{X}}\right] = \mathbb{E}\left[\|\hat{\boldsymbol{\beta}} - \boldsymbol{\beta}\|_{\boldsymbol{\Sigma}}^2 | \tilde{\boldsymbol{X}}\right],$$

where $\|x\|_{\boldsymbol{\Sigma}}^2 = x^\top\boldsymbol{\Sigma}x$. Note that we have the bias-variance decomposition

$$R_{\tilde{\boldsymbol{X}}}(\hat{\boldsymbol{\beta}}; \boldsymbol{\beta}) = \underbrace{\|\mathbb{E}(\hat{\boldsymbol{\beta}}|\tilde{\boldsymbol{X}}) - \boldsymbol{\beta}\|_{\boldsymbol{\Sigma}}^2}_{B_{\tilde{\boldsymbol{X}}}(\hat{\boldsymbol{\beta}};\boldsymbol{\beta})} + \underbrace{\text{Tr}[\text{Cov}(\hat{\boldsymbol{\beta}}|\tilde{\boldsymbol{X}})\boldsymbol{\Sigma}]}_{V_{\tilde{\boldsymbol{X}}}(\hat{\boldsymbol{\beta}};\boldsymbol{\beta})}. \tag{15}$$

Since $\hat{\boldsymbol{\beta}} = (\tilde{\boldsymbol{X}}^\top\tilde{\boldsymbol{X}})^+\tilde{\boldsymbol{X}}^\top\boldsymbol{y} = (\tilde{\boldsymbol{X}}^\top\tilde{\boldsymbol{X}})^+\tilde{\boldsymbol{X}}^\top(\boldsymbol{X}\boldsymbol{\beta} + \boldsymbol{\epsilon})$, We have

$$\mathbb{E}[\hat{\boldsymbol{\beta}}|\tilde{\boldsymbol{X}}] - \boldsymbol{\beta} = \mathbb{E}[((\tilde{\boldsymbol{X}}^\top\tilde{\boldsymbol{X}})^+\tilde{\boldsymbol{X}}^\top\boldsymbol{X} - \boldsymbol{I})\boldsymbol{\beta}|\tilde{\boldsymbol{X}}] = \tilde{\boldsymbol{\Pi}}\boldsymbol{\beta} + (\tilde{\boldsymbol{X}}^\top\tilde{\boldsymbol{X}})^+\tilde{\boldsymbol{X}}^\top\mathbb{E}[(\boldsymbol{X} - \tilde{\boldsymbol{X}})|\tilde{\boldsymbol{X}}]\boldsymbol{\beta}, \tag{16}$$

$$\text{Tr}\,\text{Cov}(\hat{\boldsymbol{\beta}}|\tilde{\boldsymbol{X}}) = \text{Tr}(\mathbb{E}[(\hat{\boldsymbol{\beta}} - \mathbb{E}[\hat{\boldsymbol{\beta}}|\tilde{\boldsymbol{X}}])(\hat{\boldsymbol{\beta}} - \mathbb{E}[\hat{\boldsymbol{\beta}}|\tilde{\boldsymbol{X}}])^\top|\tilde{\boldsymbol{X}}]). \tag{17}$$

Here, $\boldsymbol{X} - \tilde{\boldsymbol{X}} = \boldsymbol{X} \odot (1 - \boldsymbol{Z})$. We notice that the event $\{\tilde{\boldsymbol{X}}_{ij} = 0\}$ is the same as $\{\boldsymbol{Z}_{ij} = 0\}$ except for a negligible set, so in the following proof, we take them as two identical events. In this case, we have the following two relations: for any sample $i$,

$$(\boldsymbol{X} - \tilde{\boldsymbol{X}})_{i,\mathcal{Z}_i} | \tilde{\boldsymbol{X}}_{i\cdot} = 0 \,; \; (\boldsymbol{X} - \tilde{\boldsymbol{X}})_{i,\mathcal{Z}_i^c} | \tilde{\boldsymbol{X}}_{i\cdot} \sim \mathcal{N}(\boldsymbol{\Sigma}_{\mathcal{Z}_i^c \mathcal{Z}_i} \boldsymbol{\Sigma}_{\mathcal{Z}_i \mathcal{Z}_i}^{-1} \tilde{\boldsymbol{X}}_{i,\mathcal{Z}_i}, \boldsymbol{\Sigma}_{\mathcal{Z}_i^c \mathcal{Z}_i^c} - \boldsymbol{\Sigma}_{\mathcal{Z}_i^c \mathcal{Z}_i} \boldsymbol{\Sigma}_{\mathcal{Z}_i \mathcal{Z}_i}^{-1} \boldsymbol{\Sigma}_{\mathcal{Z}_i \mathcal{Z}_i^c}). \tag{18}$$

Therefore

$$\tilde{\boldsymbol{X}}^\top \mathbb{E}[(\boldsymbol{X} - \tilde{\boldsymbol{X}}) | \tilde{\boldsymbol{X}}] = \sum_i \tilde{\boldsymbol{X}}_{i\cdot}^\top \mathbb{E}[(\boldsymbol{X} - \tilde{\boldsymbol{X}})_{i\cdot} | \tilde{\boldsymbol{X}}_{i\cdot}] = \sum_i \boldsymbol{U}^i, \; \boldsymbol{U}_{\mathcal{Z}_i, \mathcal{Z}_i^c}^i = \tilde{\boldsymbol{X}}_{i,\mathcal{Z}_i}^\top \tilde{\boldsymbol{X}}_{i,\mathcal{Z}_i} \boldsymbol{\Sigma}_{\mathcal{Z}_i \mathcal{Z}_i}^{-1} \boldsymbol{\Sigma}_{\mathcal{Z}_i \mathcal{Z}_i^c}. \tag{19}$$

The remaining entries of $\boldsymbol{U}^i$ are equal to zero. For the bias term, we have that

$$B_{\tilde{\boldsymbol{X}}}(\hat{\boldsymbol{\beta}}, \boldsymbol{\beta}) = \|\mathbb{E}(\hat{\boldsymbol{\beta}} | \tilde{\boldsymbol{X}}) - \boldsymbol{\beta}\|_{\tilde{\boldsymbol{\Sigma}}}^2 = \|\tilde{\boldsymbol{\Pi}} \boldsymbol{\beta} + (\tilde{\boldsymbol{X}}^\top \tilde{\boldsymbol{X}})^+ \sum_i \boldsymbol{U}^i \boldsymbol{\beta}\|_{\tilde{\boldsymbol{\Sigma}}}^2. \tag{20}$$

The latter term can further be simplified as

$$(\tilde{\boldsymbol{X}}^\top \tilde{\boldsymbol{X}})^+ \sum_i \boldsymbol{U}^i \boldsymbol{\beta} = \tilde{\boldsymbol{X}}^+ \boldsymbol{u}, \; \boldsymbol{u} \in \mathbb{R}^{\tilde{n}}, u_i = \tilde{\boldsymbol{X}}_{i,\mathcal{Z}_i} \boldsymbol{\Sigma}_{\mathcal{Z}_i \mathcal{Z}_i}^{-1} \boldsymbol{\Sigma}_{\mathcal{Z}_i \mathcal{Z}_i^c} \boldsymbol{\beta}_{\mathcal{Z}_i^c}. \tag{21}$$

For the variance term, we have

$$\mathbb{E}[(\boldsymbol{X} - \tilde{\boldsymbol{X}}) \boldsymbol{\beta} \boldsymbol{\beta}^\top (\boldsymbol{X} - \tilde{\boldsymbol{X}})^\top | \tilde{\boldsymbol{X}}] = \sum_{i,j} \beta_i \beta_j \mathbb{E}[(\boldsymbol{X} - \tilde{\boldsymbol{X}})_{\cdot i} (\boldsymbol{X} - \tilde{\boldsymbol{X}})_{\cdot j} | \tilde{\boldsymbol{X}}] = \sum_{i,j=1}^d \beta_i \beta_j \boldsymbol{W}^{ij},$$
$$\boldsymbol{W}^{ij} = \mathrm{diag}_a(w_a^{ij}) := \mathrm{diag}_a(\mathbb{1}_{\tilde{\boldsymbol{X}}_{ai}=0, \tilde{\boldsymbol{X}}_{aj}=0}(\boldsymbol{\Sigma}_{ij} - \boldsymbol{\Sigma}_{i\mathcal{Z}_a} \boldsymbol{\Sigma}_{\mathcal{Z}_a \mathcal{Z}_a}^{-1} \boldsymbol{\Sigma}_{\mathcal{Z}_a j})). \tag{22}$$

With this relation, we have

$$\begin{aligned}
V_{\tilde{\boldsymbol{X}}}(\hat{\boldsymbol{\beta}}, \boldsymbol{\beta}) &= \mathrm{Tr}[\mathrm{Cov}(\hat{\boldsymbol{\beta}} | \tilde{\boldsymbol{X}}) \boldsymbol{\Sigma}] \\
&= \mathrm{Tr}(\mathbb{E}[(\hat{\boldsymbol{\beta}} - \mathbb{E}[\hat{\boldsymbol{\beta}} | \tilde{\boldsymbol{X}}])(\hat{\boldsymbol{\beta}} - \mathbb{E}[\hat{\boldsymbol{\beta}} | \tilde{\boldsymbol{X}}])^\top \boldsymbol{\Sigma} | \tilde{\boldsymbol{X}}]) \\
&= \mathrm{Tr}\left(\tilde{\boldsymbol{X}}^+ \left(\mathbb{E}[(\boldsymbol{X} - \tilde{\boldsymbol{X}}) \boldsymbol{\beta} \boldsymbol{\beta}^\top (\boldsymbol{X} - \tilde{\boldsymbol{X}})^\top | \tilde{\boldsymbol{X}}] + \sigma^2 \boldsymbol{I}\right)(\tilde{\boldsymbol{X}}^\top)^+ \boldsymbol{\Sigma}\right) \\
&= \mathrm{Tr}\left(\sigma^2 (\tilde{\boldsymbol{X}}^\top \tilde{\boldsymbol{X}})^+ \boldsymbol{\Sigma} + \left(\sum_{i,j=1}^d \beta_i \beta_j \boldsymbol{W}^{ij}\right)(\tilde{\boldsymbol{X}}^\top)^+ \boldsymbol{\Sigma} \tilde{\boldsymbol{X}}^+\right) \\
&= \sum_{i,j=1}^d \beta_i \beta_j \sum_{a=1}^{\tilde{n}} ((\tilde{\boldsymbol{X}}^\top)^+ \boldsymbol{\Sigma} \tilde{\boldsymbol{X}}^+)_{aa} w_a^{ij} + \sigma^2 \mathrm{Tr}\left((\tilde{\boldsymbol{X}}^\top \tilde{\boldsymbol{X}})^+ \boldsymbol{\Sigma}\right).
\end{aligned} \tag{23}$$

Combining all results above, we get the final expressions for $B_{\tilde{\boldsymbol{X}}}(\hat{\boldsymbol{\beta}}, \boldsymbol{\beta})$ and $V_{\tilde{\boldsymbol{X}}}(\hat{\boldsymbol{\beta}}, \boldsymbol{\beta})$. $\qquad \square$

**Lemma 2** (Deterministic Equivalence For Trace-class Statistics). *For the model* $\boldsymbol{\Sigma} = \delta \boldsymbol{v} \boldsymbol{v}^\top + \boldsymbol{I}$, *with* $\|\boldsymbol{v}\|^2 = 1$, *we have the following holds with high probability: for* $a, b \in \{\tilde{\boldsymbol{\beta}}, \boldsymbol{v}\}$, *and any* $\epsilon > 0$, *there exists some constant* $C$ *independent of* $n, d$,

$$\left|a^\top \hat{\boldsymbol{\Sigma}}^+ \hat{\boldsymbol{\Sigma}} b - \int \frac{s}{s + \mu_\star} \mathrm{d}G_d^{ab}(s)\right| \leq C n^{-1/2 + \epsilon}, \tag{24}$$

$$\left|\mathrm{Tr}(\hat{\boldsymbol{\Sigma}}^+ \boldsymbol{\Sigma}) - \frac{\int \frac{s}{(s+\mu_\star)^2} \mathrm{d}\tilde{H}_d(s) + \delta \int \frac{s}{(s+\mu_\star)^2} \mathrm{d}G_d^{vv}(s)}{\frac{\tilde{n}}{d} - \int \frac{s^2}{(s+\mu_\star)^2} \mathrm{d}\tilde{H}_d(s)}\right| \leq C n^{-1/2 + \epsilon}. \tag{25}$$

*Proof.* Since $\tilde{\boldsymbol{X}}$ has i.i.d. rows and each row is $\|\tilde{\boldsymbol{\Sigma}}\|$-subgaussian, by the deterministic equivalent of resolvent for random sub-gaussian sample covariance matrices, see e.g. [32, Theorem 4], for any two given $D, K > 0$, we have that the following holds with probability at least $1 - C\tilde{n}^{-D}$ for $\lambda = \Omega(\tilde{n}^{-K})$:

$$\left|\lambda a^\top \left(\tilde{\boldsymbol{X}}^\top \tilde{\boldsymbol{X}} / \tilde{n} + \lambda \boldsymbol{I}\right)^{-1} b - \lambda_\star a^\top (\lambda_\star \boldsymbol{I} + \tilde{\boldsymbol{\Sigma}})^{-1} b\right| \lesssim \frac{\mathrm{polylog}(\tilde{n})}{\sqrt{\tilde{n}}(\lambda \tilde{n})^{5/2}} \cdot \lambda_\star |a^\top (\tilde{\boldsymbol{\Sigma}} + \lambda_\star)^{-1} b|, \tag{26}$$

where $\lambda_\star$ is the unique solution of the fixed point equation

$$\tilde{n} - \frac{\lambda\tilde{n}}{\lambda_\star} = \mathrm{Tr}\big(\tilde{\boldsymbol{\Sigma}}(\tilde{\boldsymbol{\Sigma}} + \lambda_\star \boldsymbol{I})^{-1}\big).$$

We also notice that for the eigenvalue decomposition $\tilde{\boldsymbol{X}}^\top \tilde{\boldsymbol{X}}/\tilde{n} = \boldsymbol{U} D_{\tilde{\boldsymbol{X}}} \boldsymbol{U}^\top$,

$$\left| \lambda a^\top (\tilde{\boldsymbol{X}}^\top \tilde{\boldsymbol{X}}/\tilde{n} + \lambda \boldsymbol{I})^{-1} b - a^\top (\boldsymbol{I} - \hat{\boldsymbol{\Sigma}}^+ \hat{\boldsymbol{\Sigma}}) b \right| \leq \lambda |a^\top \boldsymbol{U} (\lambda \boldsymbol{I} + D_{\tilde{\boldsymbol{X}}})^{-1} \mathbb{1}_{D_{\tilde{\boldsymbol{x}}} > 0} \boldsymbol{U}^\top b| \leq \frac{\lambda}{\sigma_{\min}(\tilde{\boldsymbol{X}})^2/\tilde{n}},$$
(27)

where $\sigma_{\min}$ represents the smallest non-zero singular value. It is standard using concentration on random subgaussian matrices, see e.g. [73], to get $\sigma_{\min}(\tilde{\boldsymbol{X}})/\sqrt{\tilde{n}} \geq C(\sigma_{\min}(\tilde{\boldsymbol{\Sigma}}), \gamma/p)$ with overwhelming probability for $d/\tilde{n} = \gamma_n/p \to \gamma/p$ strictly different with $1$. Therefore, combining the above results, we get

$$\left| \lambda_\star a^\top (\lambda_\star \boldsymbol{I} + \tilde{\boldsymbol{\Sigma}})^{-1} b - a^\top (\boldsymbol{I} - \hat{\boldsymbol{\Sigma}}^+ \hat{\boldsymbol{\Sigma}}) b \right| \lesssim \frac{\mathrm{polylog}(\tilde{n})}{\lambda^{5/2} \tilde{n}^{3/2}} + \lambda.$$
(28)

On the other hand, while $\lambda\tilde{n} \to 0^+$, the above fixed point equation still makes sense, and $\lambda_\star \to \mu_\star$ such that

$$\mathrm{Tr}\big(\tilde{\boldsymbol{\Sigma}}(\tilde{\boldsymbol{\Sigma}} + \mu_\star)^{-1}\big) = \tilde{n}.$$

We next claim that

$$|\lambda_\star a^\top (\lambda_\star \boldsymbol{I} + \tilde{\boldsymbol{\Sigma}})^{-1} b - \mu_\star a^\top (\mu_\star \boldsymbol{I} + \tilde{\boldsymbol{\Sigma}})^{-1} b| \leq C\lambda,$$
(29)

while we notice that the the fixed point equation can be equivalently be written as

$$1 - \frac{p}{\gamma} = -\frac{p\lambda}{\gamma\lambda_\star} + \int \frac{\lambda_\star}{s + \lambda_\star} \mathrm{d}\tilde{H}_d(s) := f(\lambda_\star; \lambda),$$
(30)

where $\tilde{H}_d$ represents the empirical spectral distribution of $\tilde{\boldsymbol{\Sigma}}$. As a direct consequence, it is not hard to see $\mathrm{supp}(\tilde{H}_d) \subseteq [(1-p)^2, 1+\delta]$. From [31, Lemma 2.2], $\lambda_\star$ is nonnegative and monotone increasing in $\lambda$. Therefore, $f(x; \lambda)$ is non-decreasing on $(0, \infty)$ for any $\lambda \geq 0$ with $\lim_{x\to\infty} f(x; \lambda) = 1$, $\lim_{x\to 0} f(x; 0) = 0$, $\lim_{x\to 0} f(x; \lambda) = -\infty$ for any $\lambda > 0$. We also have the natural upper and lower bound as $\underline{f}(x; \lambda) \leq f(x; \lambda) \leq \bar{f}(x; \lambda)$, where $\underline{f}(x; \lambda)$ is the same as $f(x; \lambda)$ while changing $\tilde{H}_d$ to the dirac measure at $(1-p)^2$, and $\bar{f}(x; \lambda)$ is the same as $f(x; \lambda)$ while changing $\tilde{H}_d$ to the dirac measure at $1 + \delta$. So there exists some constant $C > 0$, such that $C^{-1} \leq \mu_\star \leq \lambda_\star \leq C$. This further implies that $\partial_\lambda f(x; \lambda)$ is uniformly bounded. Finally

$$\partial_x f(x; \lambda) = \frac{p\lambda}{\gamma x^2} + \int \frac{s}{(x+s)^2} \mathrm{d}\tilde{H}_d(s),$$
(31)

for $s \in [(1-p)^2, 1+\delta]$ and $x \in [C^{-1}, C]$, the above is bounded away from zero and above. Therefore, there exists another constant $\tilde{C}$ such that $\tilde{C}^{-1} \leq \partial_x f(x; \lambda) \leq \tilde{C}$. To be concise, we replace $C$ by $\max\{C, \tilde{C}\}$. Utilizing the implicit function theorem, we can get $|\partial_\lambda f(x; \lambda)| \leq C$ for $\lambda \in [0, 1]$, and therefore $|\lambda_\star - \mu_\star| \leq C\lambda$. Combining all the above arguments, we would have

$$|\lambda_\star a^\top (\lambda_\star \boldsymbol{I} + \tilde{\boldsymbol{\Sigma}})^{-1} b - \mu_\star a^\top (\mu_\star \boldsymbol{I} + \tilde{\boldsymbol{\Sigma}})^{-1} b| \leq C\lambda.$$
(32)

Taking $\lambda = \tilde{n}^{-1-\epsilon/3}$, the right-hand side turns to $0$ with the speed at least $\tilde{n}^{-1/2+\epsilon}$, so we can get the final control

$$\left| a^\top \hat{\boldsymbol{\Sigma}}^+ \hat{\boldsymbol{\Sigma}} b - a^\top \tilde{\boldsymbol{\Sigma}}(\mu_\star \boldsymbol{I} + \tilde{\boldsymbol{\Sigma}})^{-1} b \right| \leq Cn^{-1/2+\epsilon}.$$
(33)

The remaining part of the proof for $\mathrm{Tr}(\hat{\boldsymbol{\Sigma}}^+ \boldsymbol{\Sigma})$ is analogous to the result above using [32, Theorem 4, (46)], so we omit the full proof here. □

## B.2 Proof of Theorem 1

*Proof.* One important observation here is that the bias term and the variance term are homogeneous for $\|\boldsymbol{\beta}\|^2$, so below we assume $\|\boldsymbol{\beta}\| = 1$ without loss of generality. Since we are under the isotropic setting, $\boldsymbol{\Sigma}_{\mathcal{Z}\mathcal{Z}^c} = 0$ for any indices set $\mathcal{Z}$. Also, $\tilde{\boldsymbol{X}}$ has i.i.d. elements with variance $1 - p$. By Lemma 1, the bias term is simply given by

$$B_{\tilde{\boldsymbol{X}}}(\hat{\boldsymbol{\beta}}, \boldsymbol{\beta}) = \|\tilde{\boldsymbol{\Pi}}\boldsymbol{\beta}\|_2^2 = \boldsymbol{\beta}^\top(\hat{\boldsymbol{\Sigma}}^+\hat{\boldsymbol{\Sigma}} - \boldsymbol{I})\boldsymbol{\beta}, \tag{34}$$

while for the variance term, direct calculation shows $w_a^{ij} = \mathbb{1}_{\tilde{\boldsymbol{X}}_{ai=0}, \tilde{\boldsymbol{X}}_{aj=0}}\delta_{ij}$. Therefore,

$$V_{\tilde{\boldsymbol{X}}}(\hat{\boldsymbol{\beta}}, \boldsymbol{\beta}) = \sum_{i,j=1}^d \beta_i\beta_j \sum_{a=1}^{\tilde{n}} ((\tilde{\boldsymbol{X}}^\top)^+\tilde{\boldsymbol{X}}^+)_{aa}\mathbb{1}_{\tilde{\boldsymbol{X}}_{ai=0}, \tilde{\boldsymbol{X}}_{aj=0}}\delta_{ij} + \sigma^2\mathrm{Tr}\left((\tilde{\boldsymbol{X}}^\top\tilde{\boldsymbol{X}})^+\right) \tag{35}$$

$$= \underbrace{\sum_{i=1}^d \beta_i^2 \sum_{a=1}^{\tilde{n}} (\tilde{\boldsymbol{X}}\tilde{\boldsymbol{X}}^\top)_{aa}^+\mathbb{1}_{\tilde{\boldsymbol{X}}_{ai}=0}}_{(I)} + \sigma^2\mathrm{Tr}\left((\tilde{\boldsymbol{X}}\tilde{\boldsymbol{X}}^\top)^+\right). \tag{36}$$

To deal with the first term $(I)$ in variance, we use the isotropic local law [30, Theorem 2.5]. Define $R_{\tilde{n}}(\lambda) = \lambda(\tilde{\boldsymbol{X}}\tilde{\boldsymbol{X}}^\top/\tilde{n} + \lambda\boldsymbol{I})^{-1}$, $Q_{\tilde{n}}(\lambda) = \tilde{\boldsymbol{X}}\tilde{\boldsymbol{X}}^\top/n(\tilde{\boldsymbol{X}}\tilde{\boldsymbol{X}}^\top/\tilde{n} + \lambda\boldsymbol{I})^{-2}/n$, then $Q_{\tilde{n}}(\lambda) = \partial_\lambda R_{\tilde{n}}(\lambda)/n$. and we consider $\lambda$ such that $0 < \mathrm{Im}(-\lambda) < 1$, $\mathrm{Re}(\lambda) > \tilde{n}^{-2/3+\epsilon'}$ for some $\epsilon' > 0$. we have the following with high probability that

$$|R_{\tilde{n}}(\lambda)_{aa} - m(\lambda)| \le \sqrt{\frac{\mathrm{Im}(m(\lambda))}{\mathrm{Im}(-\lambda)} \cdot \tilde{n}^{-1+\epsilon}}, \tag{37}$$

uniformly for all $a \in [\tilde{n}]$. Using the similar argument as in [43, A.3], for all real $\lambda \ge \tilde{n}^{-2/3+\epsilon'}$, we get

$$|Q_{\tilde{n}}(\lambda)_{aa} - \partial_\lambda m(\lambda)/n| \le \lambda^{-2}\tilde{n}^{-(3-\epsilon)/2}, \tag{38}$$

the following holds with high probability

$$\left|\sum_{i=1}^d \beta_i^2 \cdot \sum_{a=1}^{\tilde{n}} (Q_{\tilde{n}}(\lambda)_{aa} - \partial_\lambda m(\lambda)/n)\mathbb{1}_{\tilde{\boldsymbol{X}}_{ai}=0}\right| \le \sum_{a=1}^{\tilde{n}} |Q_{\tilde{n}}(\lambda)_{aa} - \partial_\lambda m(\lambda)/n| \le \tilde{n}^{-(1-\epsilon)/2}\lambda^{-2}. \tag{39}$$

Hoeffding's inequality shows that for arbitrary small $\epsilon > 0$, with probability at least $1 - 2d\exp(-\tilde{n}^\epsilon)$,

$$\left|\sum_{a=1}^{\tilde{n}} \mathbb{1}_{\tilde{\boldsymbol{X}}_{ai}=0} - p\tilde{n}\right| \le \tilde{n}^{(1+\epsilon)/2},$$

and as a direct consequence,

$$\left|\sum_{i=1}^d \beta_i^2 \cdot \sum_{a=1}^{\tilde{n}} \mathbb{1}_{\tilde{\boldsymbol{X}}_{ai}=0} - p\tilde{n}\sum_{i=1}^d \beta_i^2\right| \le \sum_{i=1}^d \beta_i^2 \cdot \left|\sum_{a=1}^{\tilde{n}} \mathbb{1}_{\tilde{\boldsymbol{X}}_{ai}=0} - p\tilde{n}\right| \le \tilde{n}^{(1+\epsilon)/2}.$$

Combined with these, we have the following with some absolute constant $C$:

$$\left|\sum_{i=1}^d \beta_i^2 \cdot \sum_{a=1}^{\tilde{n}} Q_{\tilde{n}}(\lambda)_{aa}\mathbb{1}_{\tilde{\boldsymbol{X}}_{ai}=0} - p\mathrm{Tr}(Q_{\tilde{n}}(\lambda))\right| \le C\tilde{n}^{-(1-\epsilon)/2}\lambda^{-2}. \tag{40}$$

similarly as (27), we have that there exists a constant $C$ only depend on $\gamma_n, p$, such that with high probability, $\left|\mathrm{Tr}(Q_{\tilde{n}}(\lambda)) - \mathrm{Tr}\left((\tilde{\boldsymbol{X}}\tilde{\boldsymbol{X}}^\top)^+\right)\right| \le C\lambda$. Finally, up to some constant, we can bound the difference between $(I)$ and $p\mathrm{Tr}\left((\tilde{\boldsymbol{X}}\tilde{\boldsymbol{X}}^\top)^+\right)$ by $\lambda + C\tilde{n}^{-(1-\epsilon)/2}\lambda^{-2}$, which converges to 0 for $\lambda = \tilde{n}^{-(1-\epsilon)/6}$. To sum up, based on Lemma 2 with $\delta = 0$ and $\lambda = \tilde{n}^{-2/3+\epsilon'}$, we get that

$$\left|\frac{B_{\tilde{\boldsymbol{X}}}(\hat{\boldsymbol{\beta}}, \boldsymbol{\beta})}{r^2} - \int \frac{\mu_\star}{s + \mu_\star}\mathrm{d}G_d^{\beta\beta}(s)\right| \le Cn^{-2/3+\epsilon'}, \tag{41}$$

where $\mu_\star$ is the solution of $\tilde{n} = \mathrm{Tr}((1 + \mu_\star)^{-1}) = d/(1 + \mu_\star)$, that is, $\mu_\star = (1-p)(\gamma_n/p - 1)_+$. Also, $G_d^{\beta\beta}(s)$ is the dirac measure at $1 - p$. This suggests $B_{\tilde{X}}(\hat{\beta}, \beta)/r^2 \to (\gamma - p)/\gamma$ almost surely when $\gamma > p$, and $B_{\tilde{X}}(\hat{\beta}, \beta)/r^2 \to 0$ when $\gamma < p$. It is almost the same for us to use Lemma 2 to get that with high probability,

$$\left| \frac{V_{\tilde{X}}(\hat{\beta}, \beta)}{r^2} - \frac{\gamma(p + \kappa)(1-p)}{p(1 - p + \mu_\star)^2 - \gamma(1-p)^2} \right| \to 0,$$

which leads to our final result. $\qquad\square$

## B.3 Statement and proof of Theorem 2

In this section, we first provide the delocalized signal assumption and several definitions, then present the statement and proof of Theorem 2.

**Assumption 1** (Delocalized signal). *$\exists \alpha > 0$, such that $\|v\|_4^4 = O(d^{-\alpha})$ and $\|\tilde{\beta}\|_4^4 = O(d^{-\alpha})$.*

Here we assume that $v$ and the direction of $\beta$ should not be too sparse in order to establish concentration properties of the masking process on our signals. This is purely technical, and we can select $\alpha$ sufficiently small to accommodate specific scenarios in the application.

**Definition 1** (Spiked Covariance Structure). For $\Sigma = I + \delta v v^\top$, where $\delta > 0$ and $\|v\|^2 = 1$, denote the **masked covariance** $\tilde{\Sigma} = (1-p)^2 \Sigma + p(1-p)\,\mathrm{diag}(\Sigma)$. In other words, $\tilde{\Sigma} = (1-p)I + (1-p)^2 \delta v v^\top + p(1-p)\delta\,\mathrm{diag}(v \odot v)$. Suppose $\tilde{\Sigma} = \sum_{i=1}^d \tilde{\delta}_i \chi_i \chi_i^\top$ is the spectral decomposition of $\tilde{\Sigma}$ with $1 + \delta \geq \tilde{\delta}_1 \geq \tilde{\delta}_2 \geq \ldots \geq \tilde{\delta}_d \geq (1-p)^2$. We use $\tilde{H}_d(s) := \frac{1}{d} \cdot \sum_{i=1}^d \mathbb{1}_{s \geq \tilde{\delta}_i}$ to represent the empirical spectral distribution of $\tilde{\Sigma}$. We also denote the following (signed) empirical measures as

$$G_d^{\beta\beta}(s) = \sum_{i=1}^d \langle \tilde{\beta}, \chi_i \rangle^2 \mathbb{1}_{s \geq \tilde{\delta}_i}, \quad G_d^{\beta v}(s) = \sum_{i=1}^d \langle \tilde{\beta}, \chi_i \rangle \langle v, \chi_i \rangle \mathbb{1}_{s \geq \tilde{\delta}_i}, \quad G_d^{vv}(s) = \sum_{i=1}^d \langle v, \chi_i \rangle^2 \mathbb{1}_{s \geq \tilde{\delta}_i}.$$

Denote $\mu_\star$ to be the unique non-negative solution of

$$1 - \frac{p}{\gamma} = \int \frac{\mu_\star}{s + \mu_\star} \mathrm{d}\tilde{H}_d(s), \tag{42}$$

We then define the predicted bias and variance by

$$\mathcal{B}(\tilde{H}_d, G_d^{\beta\beta}, G_d^{\beta v}, G_d^{vv}) := \int \frac{\mu_\star}{s + \mu_\star} \mathrm{d}G_d^{\beta\beta}(s) + \left( \frac{p\delta \cdot v^\top \tilde{\beta}}{1 + \delta(1-p)} \right)^2 \cdot \int \frac{s}{s + \mu_\star} \mathrm{d}G_d^{vv}(s)$$

$$+ \delta \cdot \left( -\int \frac{\mu_\star}{s + \mu_\star} \mathrm{d}G_d^{\beta v}(s) + \frac{p\delta \cdot v^\top \tilde{\beta}}{1 + \delta(1-p)} \int \frac{s}{s + \mu_\star} \mathrm{d}G_d^{vv}(s) \right)^2,$$

$$\tag{43}$$

$$\mathcal{V}(\tilde{H}_d, G_d^{\beta v}, G_d^{vv}) := \left( \kappa + p + \frac{p^2 \delta \cdot (v^\top \tilde{\beta})^2}{1 + \delta(1-p)} \right) \cdot \frac{\int \frac{s}{(s + \mu_\star)^2} \mathrm{d}\tilde{H}_d(s) + \delta \int \frac{s}{(s + \mu_\star)^2} \mathrm{d}G_d^{vv}(s)}{\frac{\tilde{n}}{d} - \int \frac{s^2}{(s + \mu_\star)^2} \mathrm{d}\tilde{H}_d(s)}. \tag{44}$$

**Theorem 2** (Spiked covariance model). *The test risk (5) can be decomposed as $R_{\tilde{X}}(\hat{\beta}; \beta) = B_{\tilde{X}}(\hat{\beta}; \beta) + V_{\tilde{X}}(\hat{\beta}; \beta)$ with forms of the two terms available in Lemma 1. Suppose Assumption 1 holds, and for some arbitrary $\epsilon > 0$ that sufficiently small, assume $\delta = O(d^{(\alpha - 3\epsilon)/2})$ for $\alpha$ given in Assumption 1. then with overwhelming probability,*

$$\left| \frac{B_{\tilde{X}}(\hat{\beta}, \beta)}{r^2} - \mathcal{B}(\tilde{H}_d, G_d^{\beta\beta}, G_d^{\beta v}, G_d^{vv}) \right| \leq C d^{-\epsilon}; \tag{45}$$

$$\left| \frac{V_{\tilde{X}}(\hat{\beta}, \beta)}{r^2} - \mathcal{V}(\tilde{H}_d, G_d^{\beta v}, G_d^{vv}) \right| \leq C d^{-\epsilon}. \tag{46}$$

*Furthermore, if we assume $\tilde{H}_d \Rightarrow H$, $G_d^{\beta\beta} \Rightarrow G^{\beta\beta}$, $G_d^{\beta v} \Rightarrow G^{\beta v}$, $G_d^{vv} \Rightarrow G^{vv}$, then almost surely*

$$\frac{R_{\tilde{X}}(\hat{\beta}; \beta)}{r^2} \to \mathcal{B}(\tilde{H}, G^{\beta\beta}, G^{\beta v}, G^{vv}) + \mathcal{V}(\tilde{H}, G^{\beta v}, G^{vv}). \tag{47}$$

*Proof.* One important observation here is that the bias term and the variance term are homogeneous to $\|\boldsymbol{\beta}\|^2$, so below we assume $\|\boldsymbol{\beta}\| = 1$ without loss of generality.

Here, we have $\boldsymbol{\Sigma} = \delta \boldsymbol{v}\boldsymbol{v}^\top + \boldsymbol{I}$. In this case, we have

$$\boldsymbol{\Sigma}_{\mathcal{Z}_i \mathcal{Z}_i} = \delta \boldsymbol{v}_{\mathcal{Z}_i} \boldsymbol{v}_{\mathcal{Z}_i}^\top + \boldsymbol{I}_{\mathcal{Z}_i}; \quad \boldsymbol{\Sigma}_{\mathcal{Z}_i \mathcal{Z}_i}^{-1} = \boldsymbol{I}_{\mathcal{Z}_i} - \frac{\delta}{1 + \delta \|\boldsymbol{v}_{\mathcal{Z}_i}\|^2} \boldsymbol{v}_{\mathcal{Z}_i} \boldsymbol{v}_{\mathcal{Z}_i}^\top. \tag{48}$$

$$u_i = \tilde{\boldsymbol{X}}_{i,\mathcal{Z}_i} \boldsymbol{\Sigma}_{\mathcal{Z}_i \mathcal{Z}_i}^{-1} \boldsymbol{\Sigma}_{\mathcal{Z}_i \mathcal{Z}_i^c} \boldsymbol{\beta}_{\mathcal{Z}_i^c} = \tilde{\boldsymbol{X}}_{i,\mathcal{Z}_i} (\boldsymbol{I}_{\mathcal{Z}_i} - \frac{\delta}{1 + \delta \|\boldsymbol{v}_{\mathcal{Z}_i}\|^2} \boldsymbol{v}_{\mathcal{Z}_i} \boldsymbol{v}_{\mathcal{Z}_i}^\top) \delta \boldsymbol{v}_{\mathcal{Z}_i} \boldsymbol{v}_{\mathcal{Z}_i^c}^\top \boldsymbol{\beta}_{\mathcal{Z}_i^c}$$

$$= \tilde{\boldsymbol{X}}_{i,\mathcal{Z}_i} \frac{\delta}{1 + \delta \|\boldsymbol{v}_{\mathcal{Z}_i}\|^2} \boldsymbol{v}_{\mathcal{Z}_i} \boldsymbol{v}_{\mathcal{Z}_i^c}^\top \boldsymbol{\beta}_{\mathcal{Z}_i^c} = \frac{\delta}{1 + \delta \|\boldsymbol{v}_{\mathcal{Z}_i}\|^2} \left( (1 - \boldsymbol{Z}_i) \odot \boldsymbol{v} \right)^\top \boldsymbol{\beta} \cdot \tilde{\boldsymbol{X}}_i \boldsymbol{v}, \tag{49}$$

So

$$\tilde{\boldsymbol{X}}^+ \boldsymbol{u} = \tilde{\boldsymbol{X}}^+ \operatorname{diag}_i \left( \frac{\delta \left( (1 - \boldsymbol{Z}_i) \odot \boldsymbol{v} \right)^\top \boldsymbol{\beta}}{1 + \delta \|\boldsymbol{v}_{\mathcal{Z}_i}\|^2} \right) \tilde{\boldsymbol{X}} \boldsymbol{v}. \tag{50}$$

For $\boldsymbol{\beta}$ and $\boldsymbol{v}$ satisfies Assumption 1, utilizing Hoeffding's inequality, we know that with probability $1 - \exp(-d^\epsilon)$,

$$\left| \frac{\delta \left( (1 - \boldsymbol{Z}_i) \odot \boldsymbol{v} \right)^\top \boldsymbol{\beta}}{1 + \delta \|\boldsymbol{v}_{\mathcal{Z}_i}\|^2} - \frac{p \delta \cdot \boldsymbol{v}^\top \boldsymbol{\beta}}{1 + \delta (1 - p) \|\boldsymbol{v}\|^2} \right| \le C \cdot d^{-(\alpha - \epsilon)/2}.$$

Therefore, denote $c := p\delta \cdot \boldsymbol{v}^\top \boldsymbol{\beta} / (1 + \delta(1-p))$ and take a union bound, we have the following holds with high probability that

$$\left\| \tilde{\boldsymbol{X}}^+ \boldsymbol{u} - \frac{p \delta \cdot \boldsymbol{v}^\top \boldsymbol{\beta}}{1 + \delta(1-p)} \hat{\boldsymbol{\Sigma}}^+ \hat{\boldsymbol{\Sigma}} \boldsymbol{v} \right\| \le C d^{-(\alpha-\epsilon)/2} \| \hat{\boldsymbol{\Sigma}}^+ \hat{\boldsymbol{\Sigma}} \boldsymbol{v} \| \le C d^{-(\alpha-\epsilon)/2}. \tag{51}$$

Therefore we have

$$B_{\tilde{\boldsymbol{X}}}(\hat{\boldsymbol{\beta}}, \boldsymbol{\beta}) = \left\| \tilde{\boldsymbol{\Pi}} \boldsymbol{\beta} + (\tilde{\boldsymbol{X}}^\top \tilde{\boldsymbol{X}})^+ \sum_i \boldsymbol{U}^i \boldsymbol{\beta} \right\|_{\delta \boldsymbol{v}\boldsymbol{v}^\top + \boldsymbol{I}}^2 \tag{52}$$

$$= \left\| \tilde{\boldsymbol{\Pi}} \boldsymbol{\beta} \right\|^2 + \left\| \tilde{\boldsymbol{X}}^+ \boldsymbol{u} \right\|^2 + \delta \left( \boldsymbol{v}^\top (\tilde{\boldsymbol{\Pi}} \boldsymbol{\beta} + \tilde{\boldsymbol{X}}^+ \boldsymbol{u}) \right)^2,$$

and by Lemma 2, we have that for $a, b \in \{\boldsymbol{\beta}, \boldsymbol{v}\}$,

$$\left| a^\top \hat{\boldsymbol{\Sigma}}^+ \hat{\boldsymbol{\Sigma}} b - \int \frac{s}{s + \mu_\star} \mathrm{d}G_d^{ab}(s) \right| \le C n^{-(1-\epsilon')/2}, \tag{53}$$

This leads to the fact that

$$\left| \left\| \tilde{\boldsymbol{\Pi}} \boldsymbol{\beta} \right\|^2 - \int \frac{\mu_\star}{s + \mu_\star} \mathrm{d}G_d^{\beta\beta}(s) \right| \le C n^{-(1-\epsilon')/2}, \tag{54}$$

$$\left| \left\| \tilde{\boldsymbol{X}}^+ \boldsymbol{u} \right\|^2 - \left( \frac{p\delta \cdot \boldsymbol{v}^\top \tilde{\boldsymbol{\beta}}}{1 + \delta(1-p)} \right)^2 \cdot \int \frac{s}{s + \mu_\star} \mathrm{d}G_d^{vv}(s) \right| \le C (n^{-(1-\epsilon')/2} + d^{-(\alpha-\epsilon)/2}), \tag{55}$$

$$\delta \left| \left( \boldsymbol{v}^\top (\tilde{\boldsymbol{\Pi}} \boldsymbol{\beta} + \tilde{\boldsymbol{X}}^+ \boldsymbol{u}) \right)^2 - \left( - \int \frac{\mu_\star}{s + \mu_\star} \mathrm{d}G_d^{\beta v}(s) + \frac{p\delta \cdot \boldsymbol{v}^\top \tilde{\boldsymbol{\beta}}}{1 + \delta(1-p)} \int \frac{s}{s + \mu_\star} \mathrm{d}G_d^{vv}(s) \right)^2 \right| \tag{56}$$

$$\le C\delta (n^{-(1-\epsilon')/2} + d^{-(\alpha-\epsilon)/2}). \tag{57}$$

Combined with the result above, for $\delta = O(d^{(\alpha-3\epsilon)/2})$, we finally get

$$\left| \frac{B_{\tilde{\boldsymbol{X}}}(\hat{\boldsymbol{\beta}}, \boldsymbol{\beta})}{r^2} - \mathcal{B}(\tilde{H}_d, G_d^{\beta\beta}, G_d^{\beta v}, G_d^{vv}) \right| \le C d^{-\epsilon}. \tag{58}$$

We next consider the variance term. For this $\boldsymbol{\Sigma}$ model, we have that

$$\mathbb{1}_{\tilde{\boldsymbol{X}}_{ai=0}, \tilde{\boldsymbol{X}}_{aj=0}} (\boldsymbol{\Sigma}_{ij} - \boldsymbol{\Sigma}_{i\mathcal{Z}_a} \boldsymbol{\Sigma}_{\mathcal{Z}_a \mathcal{Z}_a}^{-1} \boldsymbol{\Sigma}_{\mathcal{Z}_a j}) = \mathbb{1}_{\tilde{\boldsymbol{X}}_{ai=0}, \tilde{\boldsymbol{X}}_{aj=0}} \left( \delta_{ij} + \frac{\delta v_i v_j}{1 + \delta \|\boldsymbol{v}_{\mathcal{Z}_a}\|^2} \right). \tag{59}$$

Therefore, using Lemma 1, we have

$$
\begin{aligned}
V_{\tilde{\boldsymbol{X}}}(\hat{\boldsymbol{\beta}}, \boldsymbol{\beta}) &= \sum_{i,j=1}^{d} \beta_i \beta_j \sum_{a=1}^{\tilde{n}} ((\tilde{\boldsymbol{X}}^{\top})^+ \boldsymbol{\Sigma} \tilde{\boldsymbol{X}}^+)_{aa} w_a^{ij} + \kappa \operatorname{Tr}\left((\tilde{\boldsymbol{X}}^{\top}\tilde{\boldsymbol{X}})^+ \boldsymbol{\Sigma}\right) \\
&= \sum_{i,j=1}^{d} \beta_i \beta_j \sum_{a=1}^{\tilde{n}} ((\tilde{\boldsymbol{X}}^{\top})^+ \boldsymbol{\Sigma} \tilde{\boldsymbol{X}}^+)_{aa} \mathbb{1}_{\tilde{\boldsymbol{X}}_{ai=0}, \tilde{\boldsymbol{X}}_{aj=0}}\left(\delta_{ij} + \frac{\delta v_i v_j}{1 + \delta \|\boldsymbol{v}_{\mathcal{Z}_a}\|^2}\right) + \kappa \operatorname{Tr}\left((\tilde{\boldsymbol{X}}^{\top}\tilde{\boldsymbol{X}})^+ \boldsymbol{\Sigma}\right) \\
&= \sum_{a=1}^{\tilde{n}} ((\tilde{\boldsymbol{X}}^{\top})^+ \boldsymbol{\Sigma} \tilde{\boldsymbol{X}}^+)_{aa} \sum_{i=1}^{d} \beta_i^2 \mathbb{1}_{\tilde{\boldsymbol{X}}_{ai=0}} + \delta \sum_{a=1}^{\tilde{n}} ((\tilde{\boldsymbol{X}}^{\top})^+ \boldsymbol{\Sigma} \tilde{\boldsymbol{X}}^+)_{aa} \frac{\sum_{i,j=1}^{d} \beta_i \beta_j \mathbb{1}_{\tilde{\boldsymbol{X}}_{ai=0}, \tilde{\boldsymbol{X}}_{aj=0}} v_i v_j}{1 + \delta \|\boldsymbol{v}_{\mathcal{Z}_a}\|^2} \\
&\quad + \kappa \operatorname{Tr}\left((\tilde{\boldsymbol{X}}^{\top}\tilde{\boldsymbol{X}})^+ \boldsymbol{\Sigma}\right) \\
&= \sum_{a=1}^{\tilde{n}} ((\tilde{\boldsymbol{X}}^{\top})^+ \boldsymbol{\Sigma} \tilde{\boldsymbol{X}}^+)_{aa} \left(\frac{\delta\left((1 - \boldsymbol{Z}_a)^{\top}(\boldsymbol{\beta} \odot \boldsymbol{v})\right)^2}{1 + \delta \boldsymbol{Z}_a^{\top}(\boldsymbol{v} \odot \boldsymbol{v})} + (1 - \boldsymbol{Z}_a)^{\top}(\boldsymbol{\beta} \odot \boldsymbol{\beta})\right) + \kappa \operatorname{Tr}\left((\tilde{\boldsymbol{X}}^{\top}\tilde{\boldsymbol{X}})^+ \boldsymbol{\Sigma}\right).
\end{aligned}
\tag{60}
$$

Hoeffding's inequality gives

$$
\left| \frac{\delta\left((1 - \boldsymbol{Z}_a)^{\top}(\boldsymbol{\beta} \odot \boldsymbol{v})\right)^2}{1 + \delta \boldsymbol{Z}_a^{\top}(\boldsymbol{v} \odot \boldsymbol{v})} - \frac{\delta p^2 (\boldsymbol{\beta}^{\top}\boldsymbol{v})^2}{1 + \delta(1 - p)} \right| \leq C d^{-(\alpha-\epsilon)/2},
$$

as well as $|(1 - \boldsymbol{Z}_a)^{\top}(\boldsymbol{\beta} \odot \boldsymbol{\beta}) - p| \leq d^{-(\alpha-\epsilon)/2}$, which holds uniformly for every $a \leq \tilde{n}$ with probability $1 - \exp(-d^{\epsilon})$. Therefore, with high probability,

$$
\left| \operatorname{Tr}[\operatorname{Cov}(\hat{\boldsymbol{\beta}}|\tilde{\boldsymbol{X}})\boldsymbol{\Sigma}] - \operatorname{Tr}\left(\hat{\boldsymbol{\Sigma}}^+\boldsymbol{\Sigma}\right)\left(\kappa + p + \frac{\delta p^2 (\boldsymbol{\beta}^{\top}\boldsymbol{v})^2}{1 + \delta(1 - p)}\right) \right| \leq \operatorname{Tr}\left(\hat{\boldsymbol{\Sigma}}^+\boldsymbol{\Sigma}\right) \cdot d^{-(\alpha-\epsilon)/2}, \tag{61}
$$

Note that by Lemma 2, we have the following with overwhelming probability that

$$
\left| \operatorname{Tr}(\hat{\boldsymbol{\Sigma}}^+\boldsymbol{\Sigma}) - \frac{\int \frac{s}{(s+\mu_\star)^2}\mathrm{d}\tilde{H}_d(s) + \delta \int \frac{s}{(s+\mu_\star)^2}\mathrm{d}G_d^{vv}(s)}{\frac{\tilde{n}}{d} - \int \frac{s^2}{(s+\mu_\star)^2}\mathrm{d}\tilde{H}_d(s)} \right| \leq C\tilde{n}^{-1/2+\epsilon}, \tag{62}
$$

To sum up, we have

$$
\left| \operatorname{Tr}[\operatorname{Cov}(\hat{\boldsymbol{\beta}}|\tilde{\boldsymbol{X}})\boldsymbol{\Sigma}] - \mathcal{V}(\tilde{H}_d, G_d^{\beta v}, G_d^{vv}) \right| \leq C d^{-(\alpha-\epsilon)/2}, \tag{63}
$$

which is equivalent to

$$
\left| \frac{V_{\tilde{\boldsymbol{X}}}(\hat{\boldsymbol{\beta}}, \boldsymbol{\beta})}{r^2} - \mathcal{V}(\tilde{H}_d, G_d^{\beta v}, G_d^{vv}) \right| \leq C d^{-\epsilon}. \tag{64}
$$

As a direct consequence, if we have $\tilde{H}_d \Rightarrow H$, $G_d^{\beta\beta} \Rightarrow G^{\beta\beta}$, $G_d^{\beta v} \Rightarrow G^{\beta v}$, $G_d^{vv} \Rightarrow G^{vv}$, then by the definition of weak convergence, almost surely we have

$$
\frac{R(\hat{\boldsymbol{\beta}}, \boldsymbol{\beta})}{r^2} \to \mathcal{B}(\tilde{H}, G^{\beta\beta}, G^{\beta v}, G^{vv}) + \mathcal{V}(\tilde{H}, G^{\beta v}, G^{vv}). \tag{65}
$$

Here we are able to tell the weak convergence of the (signed) measure $G_d^{\beta v}$ because the total variation of this measure is given by $|G_d^{\beta v}| = \sum_{i=1}^{d} |\langle \boldsymbol{\beta}, \boldsymbol{\chi}_i \rangle \langle \boldsymbol{v}, \boldsymbol{\chi}_i \rangle| \leq \sqrt{\sum_{i=1}^{d} \langle \boldsymbol{\beta}, \boldsymbol{\chi}_i \rangle^2 \cdot \sum_{i=1}^{d} \langle \boldsymbol{v}, \boldsymbol{\chi}_i \rangle^2} = 1$, so the sequence of $G_d^{\beta v}$ is also tight. See e.g. [74] for detailed argument. $\qquad \square$

### B.4  Statement and proof of Theorem 3

**Theorem 3** (General covariance model). *For general $\boldsymbol{\Sigma}$, assume $\boldsymbol{\beta}$ is an eigenvector of $\boldsymbol{\Sigma}$ with eigenvalue $\eta$. The test risk $R(\hat{\boldsymbol{\beta}}, \boldsymbol{\beta}) := \mathbb{E}\left[\|\hat{\boldsymbol{\beta}} - \boldsymbol{\beta}\|_{\boldsymbol{\Sigma}}^2 | \tilde{\boldsymbol{X}}\right]$ can be expressed as:*

$$
R_{\tilde{\boldsymbol{X}}}(\hat{\boldsymbol{\beta}}; \boldsymbol{\beta}) = \underbrace{\|(\tilde{\boldsymbol{X}}^+ \tilde{\boldsymbol{X}}' - \boldsymbol{I})\boldsymbol{\beta}\|_{\boldsymbol{\Sigma}}^2}_{B_x(\hat{\boldsymbol{\beta}}; \boldsymbol{\beta})} + \underbrace{\sum_{i,j} \beta_i \beta_j \sum_{a=1}^{\tilde{n}} ((\tilde{\boldsymbol{X}}^{\top})^+ \boldsymbol{\Sigma} \tilde{\boldsymbol{X}}^+)_{aa} w_a^{ij} + \sigma^2 \operatorname{Tr}\left((\tilde{\boldsymbol{X}}^{\top}\tilde{\boldsymbol{X}})^+ \boldsymbol{\Sigma}\right)}_{V_x(\hat{\boldsymbol{\beta}}; \boldsymbol{\beta})}.
$$

$$\tag{66}$$

*Here we denote*

$$\mathcal{Z}_a = \{j|\boldsymbol{Z}_{aj}=1\}, \quad \mathcal{Z}_a^c = \{j|\boldsymbol{Z}_{aj}=0\}; \tag{67}$$

$$\tilde{\boldsymbol{X}}' \in \mathbb{R}^{\tilde{n}\times d}, \quad \tilde{\boldsymbol{X}}'_{a,\mathcal{Z}_a} = \eta \tilde{\boldsymbol{X}}_{a,\mathcal{Z}_a}\boldsymbol{\Sigma}^{-1}_{\mathcal{Z}_a\mathcal{Z}_a}, \quad \tilde{\boldsymbol{X}}'_{a,\mathcal{Z}_a^c}=0; \tag{68}$$

$$w^{ij} \in \mathbb{R}^n, \quad w^{ij} = \mathrm{diag}_a(\mathbb{1}_{\tilde{\boldsymbol{X}}_{ai=0},\tilde{\boldsymbol{X}}_{aj=0}}(\boldsymbol{\Sigma}_{ij} - \boldsymbol{\Sigma}_{i\mathcal{Z}_a}\boldsymbol{\Sigma}^{-1}_{\mathcal{Z}_a\mathcal{Z}_a}\boldsymbol{\Sigma}_{\mathcal{Z}_a j})). \tag{69}$$

*Proof.* For the bias term, we have that

$$\|\mathbb{E}(\hat{\boldsymbol{\beta}}|\tilde{\boldsymbol{X}}) - \boldsymbol{\beta}\|^2_{\boldsymbol{\Sigma}} = \|\tilde{\boldsymbol{\Pi}}\boldsymbol{\beta} + \tilde{\boldsymbol{X}}^+\boldsymbol{u}\|^2_{\boldsymbol{\Sigma}}, \ \boldsymbol{u} \in \mathbb{R}^{\tilde{n}}, u_i = \tilde{\boldsymbol{X}}_{i,\mathcal{Z}_i}\boldsymbol{\Sigma}^{-1}_{\mathcal{Z}_i\mathcal{Z}_i}\boldsymbol{\Sigma}_{\mathcal{Z}_i\mathcal{Z}_i^c}\boldsymbol{\beta}_{\mathcal{Z}_i^c}. \tag{70}$$

In our case, we consider $\boldsymbol{\beta}$ to be an eigenvector of $\boldsymbol{\Sigma}$ with eigenvalue $\eta$. That is, we have

$$\forall \mathcal{Z}_i, \quad \boldsymbol{\Sigma}_{\mathcal{Z}_i\mathcal{Z}_i}\boldsymbol{\beta}_{\mathcal{Z}_i} + \boldsymbol{\Sigma}_{\mathcal{Z}_i\mathcal{Z}_i^c}\boldsymbol{\beta}_{\mathcal{Z}_i^c} = \eta\boldsymbol{\beta}_{\mathcal{Z}_i}. \tag{71}$$

Substituting $\boldsymbol{\Sigma}_{\mathcal{Z}_i\mathcal{Z}_i^c}\boldsymbol{\beta}_{\mathcal{Z}_i^c}$ with $\eta\boldsymbol{\beta}_{\mathcal{Z}_i} - \boldsymbol{\Sigma}_{\mathcal{Z}_i\mathcal{Z}_i}\boldsymbol{\beta}_{\mathcal{Z}_i}$ in $\boldsymbol{u}$ leads to

$$u_i = \eta\tilde{\boldsymbol{X}}_{i,\mathcal{Z}_i}\boldsymbol{\Sigma}^{-1}_{\mathcal{Z}_i\mathcal{Z}_i}\boldsymbol{\beta}_{\mathcal{Z}_i} - \tilde{\boldsymbol{X}}_{i,\mathcal{Z}_i}\boldsymbol{\beta}_{\mathcal{Z}_i}. \tag{72}$$

Thus we can define $\boldsymbol{u}' \in \mathbb{R}^{\tilde{n}}$, with $\boldsymbol{u}'_i = \eta\tilde{\boldsymbol{X}}_{i,\mathcal{Z}_i}\boldsymbol{\Sigma}^{-1}_{\mathcal{Z}_i\mathcal{Z}_i}\boldsymbol{\beta}_{\mathcal{Z}_i}$. Then we have

$$\boldsymbol{u} = \boldsymbol{u}' - \tilde{\boldsymbol{X}}\boldsymbol{\beta}. \tag{73}$$

Plugging $\boldsymbol{u}'$ in the bias term, we have that

$$\|\mathbb{E}(\hat{\boldsymbol{\beta}}|\tilde{\boldsymbol{X}}) - \boldsymbol{\beta}\|^2_{\boldsymbol{\Sigma}} = \|\tilde{\boldsymbol{X}}^+\boldsymbol{u}' - \boldsymbol{\beta}\|^2_{\boldsymbol{\Sigma}}. \tag{74}$$

That is, the term $\tilde{\boldsymbol{X}}^+\tilde{\boldsymbol{X}}\boldsymbol{\beta}$ is canceled in the bias term. Furthermore, we define $\tilde{\boldsymbol{X}}' \in \mathbb{R}^{\tilde{n}\times d}$, such that $\tilde{\boldsymbol{X}}'_{i,\mathcal{Z}_i} = \eta\tilde{\boldsymbol{X}}_{i,\mathcal{Z}_i}\boldsymbol{\Sigma}^{-1}_{\mathcal{Z}_i\mathcal{Z}_i}, \tilde{\boldsymbol{X}}'_{i,\mathcal{Z}_i^c} = 0$. Then the bias term can be written as:

$$\|\mathbb{E}(\hat{\boldsymbol{\beta}}|\tilde{\boldsymbol{X}}) - \boldsymbol{\beta}\|^2_{\boldsymbol{\Sigma}} = \|(\tilde{\boldsymbol{X}}^+\tilde{\boldsymbol{X}}' - \boldsymbol{I})\boldsymbol{\beta}\|^2_{\boldsymbol{\Sigma}}. \tag{75}$$

The variance term can be directly obtained due to Lemma 1. $\square$

*Remark* 1. Comparing the risk terms with standard ridge-less regression, we see that the masking introduces additional variance through the term:

$$\sum_{a=1}^{\tilde{n}}((\tilde{\boldsymbol{X}}^\top)^+\boldsymbol{\Sigma}\tilde{\boldsymbol{X}}^+)_{aa}w_a^{ij} \tag{76}$$

Here $w_a^{ij}$ can be interpreted as the sum of covariances for pair-wise masked features in $\tilde{\boldsymbol{X}}$ conditioning on remaining unmasked features. As for the bias term, for eigenvectors $\boldsymbol{\beta}$ with large eigenvalues $\eta$, replacing $\tilde{\boldsymbol{X}}$ with $\tilde{\boldsymbol{X}}'$ cancels the shrinkage effect in the projection matrix $\tilde{\boldsymbol{X}}^+\tilde{\boldsymbol{X}}$, which effectively reduces the bias term. In summary, if there are significant dependency between features in $\boldsymbol{X}$ resulting in small $w^{ij}$, then a reduced risk of the masked regression compared to ordinary ridgeless regression is anticipated due to the bias term.

## C   Experimental details

### C.1   Simulations

Across all simulations, input data matrices $\boldsymbol{X} \in \mathbb{R}^{n\times d}$ had rows sampled i.i.d. from $\mathcal{N}(\boldsymbol{0},\boldsymbol{\Sigma})$, where the specific covariance $\boldsymbol{\Sigma}$ and dimensions $(n,d)$ varied by experiment. Target values $\boldsymbol{y} \in \mathbb{R}^n$ were generated as $\boldsymbol{y} = \boldsymbol{X}\boldsymbol{\beta} + \boldsymbol{\epsilon}$, with noise $\epsilon_i \sim \mathcal{N}(0,0.04)$. The ground-truth coefficient vector $\boldsymbol{\beta} \in \mathbb{R}^d$ was also experiment-specific. For each masking probability $p \in \{0.05,0.10,\ldots,0.95\}$, masked data $\tilde{\boldsymbol{X}}$ and targets $\tilde{\boldsymbol{y}}$ were constructed as per our problem formulation. Regression estimates $\hat{\boldsymbol{\beta}}$ were obtained either via the pseudo-inverse $\tilde{\boldsymbol{X}}^+\tilde{\boldsymbol{y}}$ (Figs. 1B–D, 3) or by solving $(\tilde{\boldsymbol{X}}^\top\tilde{\boldsymbol{X}} + \lambda\boldsymbol{I}_d)^{-1}\tilde{\boldsymbol{X}}^\top\tilde{\boldsymbol{y}}$ with $\lambda = 10^{-6}$ (other figures), with negligible empirical difference between methods. Test risk was computed using $10n$ new test samples, and $\hat{\boldsymbol{\beta}}$ was calculated with 50 repetitions per $p$ through sampling different $\tilde{\boldsymbol{X}}$ from $\boldsymbol{X}$. All error bars shown in this work indicate standard deviations.

**Fig. 1A** Here, $\mathbf{\Sigma} = \mathbf{I}_d$. The vector $\boldsymbol{\beta}$ was sampled from a uniform distribution and normalized to $\|\boldsymbol{\beta}\|_2^2 = 1$. We evaluated two settings: 1) Overparametrized: $n = 2000, d = 5n = 10000$ ($\gamma = 5$); 2) Underparametrized: $n = 4000, d = 0.5n = 2000$ ($\gamma = 0.5$). The theoretical risk was calculated using the formula in Theorem 1.

**Figs. 1B–D and 3** The covariance was $\mathbf{\Sigma} = \mathbf{I}_d + \delta \boldsymbol{v}\boldsymbol{v}^\top$, where $\boldsymbol{v} \in \mathbb{R}^d$ was a uniformly sampled vector (Fig. 1B-D) or a all-ones vector, both scaled to have norm 1 (Fig. 3), and $\delta \in \{1, 10, 100\}$ controlled spike strength. Coefficients $\boldsymbol{\beta}$ with norm 1 were generated with $\boldsymbol{\beta} = \cos\theta\boldsymbol{v} + \sin\theta\boldsymbol{u}$, where $\boldsymbol{u} \propto \boldsymbol{b} - (\boldsymbol{b}^\top\boldsymbol{v})\boldsymbol{v}$ is the normalized component of a uniformly sampled vector $\boldsymbol{b}$ after removing its projection onto $\boldsymbol{v}$. Parameters were $n = 200, d = 5n = 1000$. For theoretical risk calculation in these figures, $\lambda_\star$ was first obtained from $\tilde{n} - \lambda_{\text{reg}}/\lambda_\star = \text{Tr}(\tilde{\mathbf{\Sigma}}(\tilde{\mathbf{\Sigma}} + \lambda_\star \mathbf{I}_d)^{-1})$, where $\tilde{\mathbf{\Sigma}} = (1 - p)^2\mathbf{\Sigma} + p(1 - p)\,\text{diag}(\mathbf{\Sigma})$ and $\lambda_{\text{reg}} = 10^{-8}$. Then, parameters $(\phi_\beta, \phi_v, \psi, u, c)$ were calculated using Eq. (8), and the final risk via Corollary 1 .

**Fig. 1E** The covariance $\mathbf{\Sigma} \in \mathbb{R}^{d \times d}$ was constructed by one of three methods. In all cases, $n = 500, d = 5n = 2500$. For each $\mathbf{\Sigma}$, $\boldsymbol{\beta}$ was an eigenvector of $\mathbf{\Sigma}$ corresponding to an eigenvalue at a specific quantile of its spectrum.

- **Uniform:** $\mathbf{\Sigma} = \boldsymbol{Q}\,\text{diag}(\lambda_1, \ldots, \lambda_d)\boldsymbol{Q}^\top$, with $\lambda_i \sim \mathcal{U}(1, 10)$ and $\boldsymbol{Q}$ a random orthogonal matrix generated via QR decomposition of a randomly sampled Gaussian matrix.

- **Beta distributed:** Similar to 'uniform', except that the eigenvalues were sampled from a $\text{Beta}(2, 6)$ distribution then scaled to have min 1 and max 10.

- **Latent space model:** $\mathbf{\Sigma} = \mathbf{I}_d + \boldsymbol{W}\boldsymbol{W}^\top$, with $\boldsymbol{W} \in \mathbb{R}^{d \times q}$ ($q = 0.5d$) having i.i.d. Gaussian entries $\sim N(0, (10 - 1)/(\sqrt{d} + \sqrt{q})^2)$. This construction ensures that the eigenvalues of $\mathbf{\Sigma}$ approximately range from 1 to 10.

**Figs. 1G and 4** The covariance was $\mathbf{\Sigma} = \mathbf{I}_d + \boldsymbol{W}\boldsymbol{W}^\top$. $\boldsymbol{W} \in \mathbb{R}^{d \times q} = \boldsymbol{Q}\boldsymbol{D}\boldsymbol{R}^\top$, where $\boldsymbol{Q} \in \mathbb{R}^{d \times d}$ and $\boldsymbol{R} \in \mathbb{R}^{q \times q}$ were random orthogonal matrices (Haar distribution via `scipy.stats.ortho_group.rvs`), and $\boldsymbol{D} \in \mathbb{R}^{d \times q}$ was a diagonal matrix with its $q$ non-zero entries set to a specified eigenvalue (e.g., 100). Coefficients were $\boldsymbol{\beta} = \boldsymbol{W}(\mathbf{I}_q + \boldsymbol{W}^\top\boldsymbol{W})^{-1}\boldsymbol{\theta}$ for a uniformly sampled vector $\boldsymbol{\theta} \in \mathbb{R}^q$. Parameters were $n = 100, d = 5000, q = 50$.

**Tables 5 and 9** The covariance matrices were the same as those constructed for the Beta-distributed and latent space models in **Fig. 1E**. In the linear model, the implementation of $R^2$MAE is as follows. We first sample a row-wise masking ratio $p_i \sim \mathcal{U}(p_{\min}, p_{\max})$. This ratio $p_i$ is then used to sample the mask for each row of the data matrix. The resulting masked matrix $\boldsymbol{X}_{\text{sub}}$ is further used to construct $\tilde{\boldsymbol{X}}$ and calculate $\tilde{\boldsymbol{\beta}}$. Note that the removal probability for each row depends on $1 - p_i$, so the simplification of sample size as $np$ from fixed MR settings is no longer applicable. Therefore, to ensure a fair comparison, in the corresponding fixed MR settings, we sample each row with a constant probability equal to the fixed MR. Finally, we test five random seeds for generating the model and masking matrices. Similar to previous experiments, the normalized test risks shown are the average values over 50 runs. For fixed MR settings, we tested MR values $\{0, 0.01, 0.02, \cdots, 0.99\}$. We confirmed that the $R^2$MAE results exactly match those of the fixed MR setting when $p_{\min} = p_{\max}$ and the same random seed is used.

## C.2 Evaluations on trained BERT and MAE models

For Fig. 1F, BERT fine-tuning accuracies at different masking ratios were obtained from [29] (data sourced from the GitHub repository). These reported accuracies were transformed as follows. First, error rates were calculated as (1 - accuracy). To normalize the y-intercepts, each curve was linearly extrapolated to a masking ratio of zero using its values at masking ratios 0.15 and 0.3; each curve was then vertically shifted so that its extrapolated value at 0% masking ratio became zero. Subsequently, each curve was multiplied by a unique scaling factor to ensure all curves shared a common slope for the line segment connecting their points at masking ratios 0.4 and 0.8. For Figs. 1G and 4, MAE fine-tuning and linear probing accuracies were directly obtained from [6].

## C.3 MNIST

**Dataset and model architecture.** We used the standard MNIST dataset that consists of 60,000 training and 10,000 test grayscale images of handwritten digits at $28 \times 28$ pixels. We implemented a

three-layer MLP with 784-dimensional input (flattened images), a first hidden layer with variable size (16 for underparameterized setting, 512 for overparametrized setting), a second hidden layer of variable size (16, 32, 64, 256, or 1024 units), and a 784-dimensional output layer. Each hidden layer uses ReLU activation with batch normalization, and the output layer uses sigmoid activation. The training objective is mean squared error (MSE) calculated on the masked pixels.

**Training procedure.** For each model setting, we tested masking ratios {0.1, 0.2, 0.3, 0.4, 0.5, 0.6, 0.7, 0.8, 0.9}. Models were trained for 15 epochs using Adam optimizer (learning rate 0.003, batch size 128). All experiments used PyTorch on a NVIDIA A6000 GPU with fixed random seeds. Each experiment was finished in several minutes.

**Evaluation.** We performed digit classification for each pretrained model through linear probing. Specifically, we froze the first two layers of each pretrained model and trained only a new classification layer (mapping from the second hidden layer to 10 output classes) using cross-entropy loss. The linear probing classifier was trained for 15 epochs using Adam optimizer (learning rate 0.003).

## C.4 CelebA

**Dataset and model architecture.** The CelebA dataset contains over 200,000 celebrity face images with 40 attribute annotations. Images were resized to $128 \times 128$ pixels using the official training/validation/test split. We used a U-Net with four downsampling blocks in the encoder, a bottleneck, and four upsampling blocks in the decoder with skip connections. Each convolutional block contains two 3×3 convolutional layers with batch normalization and ReLU activation. The training objective is mean squared error (MSE) calculated on the masked pixels.

**Training procedure.** We tested base channel counts of {8, 16, 32} with masking ratios {0.1, 0.2, 0.3, 0.4, 0.5, 0.6, 0.7, 0.8, 0.9}. Each model was trained for 10 epochs using Adam optimizer (learning rate 0.001, batch size 256). All experiments used a NVIDIA 6000 GPU with fixed random seeds. Each experiment was finished in one hour.

**Evaluation.** We tested representation performance through inputting uncorrupted images and evaluating the reconstruction MSE of the output. For linear probing, we extracted the U-Net encoder and bottleneck, froze their weights, and trained a classifier for the 40 CelebA attributes. The classifier included global average pooling, a shared feature extraction layer (512 units with dropout), and 40 independent linear output heads. For each setting, the linear probing classifier was trained for 10 epochs using Adam optimizer (learning rate 0.001) and binary cross-entropy loss.

## C.5 ViT MAE models

**Dataset and model architecture.** We used the ViT-base MAE model and the ImageNet-1K training split as pretraining data, following the MAE codebase [6].

**Mask pretraining schemes.** The patch tokens in the MAE input sequence are masked by one of the following strategies:

- **Fixed MR**. A constant fraction $\rho$ of patch tokens in the sequence is masked. We tested MR values of $\{0.5, 0.75, 0.9\}$. MR 0.75 is the MAE default.
- **Dynamic MR**. The masking ratio follows a linear decay: $\rho_t = \max\{\rho_{\min}, \rho_{\max} - \rho_{\max} t \lambda_{\text{decay}}\}$. Here, $t$ represents the number of training epochs. We set $\rho_{\max} = 0.9, \rho_{\min} = 0.6$, and $\lambda_{\text{decay}}$ is chosen such that $\rho_t$ linearly decays throughout the training. This mimics the scheme proposed in [34].
- **MDLM**. For every mini-batch, a masking ratio $\rho \sim \mathcal{U}(0, 1)$ is sampled. We use mean token mask loss for each MR (for all experiments), which is equivalent to $w_t = 1/k$ in the ELBO of [54]. The implementation is very similar to the standard log-linear schedule of $\alpha(t)$ [54].
- **R²MAE**. For every mini-batch, a masking ratio $\rho \sim \mathcal{U}(0.6, 0.9)$ is sampled.

**Training procedure and evaluation.** We trained all models for 150 epochs with 10 warmup epochs. All models were later fine-tuned on the ImageNet-1K training split and evaluated on the validation split. Other pretraining and fine-tuning configurations exactly follow the instructions in the MAE codebase. Each experiment was performed on one NVIDIA H100 GPU with the same fixed random seed, and each epoch took approximately 0.3 hours.

## C.6 RoBERTa models

**Dataset and model architecture.** We used the HuggingFace RoBERTa-medium and RoBERTa-base models, and the 10B token subset of FineWeb (sample-10BT, downloaded from HuggingFace) [75] as the training set. Although our implementation differs from [29] in its training set and layer-norm design, we found the fine-tuning accuracies to be overall comparable.

**Mask pretraining schemes.** The tokens in the input sequence are masked by one of the following strategies:

- **Fixed MR**. A constant fraction $\rho$ of tokens in the sequence is masked. We tested MR values of $\{0.15, 0.4\}$. MR 0.15 is the MLM default, and an MR of 0.4 is recommended in [29].
- **Dynamic MR**. The masking ratio follows a linear decay: $\rho_t = \max\{\rho_{\min}, \rho_{\max} - \rho_{\max} t \lambda_{\text{decay}}\}$. Here, $t$ represents the number of training steps. We set $\rho_{\max} = 0.4, \rho_{\min} = 0.15$, and $\lambda_{\text{decay}}$ is chosen such that $\rho_t$ linearly decays throughout the training.
- **MDLM**. For every mini-batch, a masking ratio $\rho \sim \mathcal{U}(0, 1)$ is sampled.
- **$R^2$MAE**. For every mini-batch, a masking ratio $\rho \sim \mathcal{U}(0.15, 0.4)$ is sampled.

**Training procedure and evaluation.** RoBERTa-base follows the HuggingFace default setting, while RoBERTa-medium overrides the following parameters: `vocab_size=50265`, `hidden_size=512`, `num_hidden_layers=8`, `num_attention_heads=8`, `intermediate_size=2048`, `max_position_embeddings=514`. We used AdamW optimizer, a max sequence length of 128, an effective batch size of 2048, a weight decay of 0.01, a warmup ratio of 0.03, a learning rate of 7e-4/3e-4 for the RoBERTa-medium/base models, and default linear learning rate decay. Fine-tuning was performed on the GLUE datasets (MNLI, QQP, SST-2, QNLI) for 5 epochs with a learning rate of 2e-5 and a batch size of 32. The average accuracy of three fine-tuning runs was reported as the final accuracy, following [29]. Each experiment was performed on one NVIDIA H100 GPU with fixed random seeds, and finished in one day.

## C.7 DNA sequence models

**Dataset and model architecture.** We adopted the GPN-MSA [16] framework, which is a 12-layer transformer model with 12 attention heads per transformer layer. Benegas et al. [16] curated a training set comprising multiple-sequence alignment (MSA) from human DNA and 89 other species, with careful filtering and biological considerations; please refer to [16] for more details on the model and the training set. Each training sample consists of a 128-base pair (bp) window of human genome and its corresponding MSA, and the pretraining task is to predict the token (A/C/G/T) in the masked locations of human DNA, based on the input of other unmasked locations and the auxiliary MSA information.

**Mask pretraining schemes.** Before encoding, the raw input $X$ is corrupted as $X_{\text{mask}}$ by one of the following strategies:

- **Fixed MR**. A constant fraction $\rho$ of base pairs in the sequence is masked. $\rho = 15\%$ corresponds to GPN-MSA default [16].
- **Dynamic MR**. Same as the fixed MR case, except that the masking ratio follows a linear decay $\rho_t = \max\{\rho_{\min}, \rho_{\max} - \rho_{\max} t \lambda_{\text{decay}}\}$. Here $t$ represents the number of training steps. We set $\rho_{\max} = 0.30, \rho_{\min} = 0.15$ and choose $\lambda$ such that $\rho_t$ linearly decays throughout training.
- **MDLM**. For every mini-batch, a masking ratio $\rho \sim \mathcal{U}(0, 1)$ is sampled.
- **$R^2$MAE**. For every mini-batch, a masking ratio $\rho \sim \mathcal{U}(0.05, 0.3)$ is sampled.
- **$R^2$MAE + Dynamic MR**. For every mini-batch, we sample the masking ratio $\rho \sim \mathcal{U}(\rho_t, 0.3)$, where $\rho_t = \max\{\rho_{\min}, \rho_{\max} - \rho_{\max} t \lambda_{\text{decay}}\}$. To match masking ratio expectation with the Dynamic MR setting, we set $\rho_{\max} = 0.30, \rho_{\min} = 0.00$ and choose $\lambda$ such that $\rho_t$ linearly decays throughout training. Although we applied early stopping criteria based on validation loss to all models, only this model triggered an early stop, occurring at a masking ratio of $\rho_t = 0.10$.

For CL approaches, we first describe their combinations with $R^2$MAE, as their standalone implementation results from straightforward simplification of $R^2$MAE + CL schemes.

- **$R^2$MAE + CL (k=0)**. We implement a token-wise MLP layer with hidden space [128,128] and ReLU activation that projects the transformer-learned masked location representations to output $\tilde{P} \in \mathbb{R}^{n_{\text{mask}} \times l}$. Here $l = 10$ represents the length of the pre-defined mask ratio vector. We further implement a row-wise projection layer with sigmoid activation plus 0.5 to obtain strictly positive entries (and to improve optimization stability), which we term as $P \in \mathbb{R}^{R \times l}$. Finally, we apply $K = 10$ iterations of the non-square Sinkhorn operator described in Appendix A.3: $M(y) = \text{Sinkhorn}^{(K)}(\sigma(P))$. For every mini-batch, we now sample the mask ratio in $[0.05, ..., 0.30]$ with length 10, and select the column of $M(y)$ based on the selected mask ratio index. Then we multiply this column of $M(y)$ to the element-wise loss for the masked locations. The newly implemented layers are optimized together with the main reconstruction model, after being fixed for 5000 initial training steps.

- **$R^2$MAE + CL**. Apart from the additional components described above, we employed a dynamic multiplier to the gradient received by these additionally implemented layers, decreasing from 1 to $-1$ throughout its training with a linear decay.

- **CL.** The setting is effectively implemented by always selecting one fixed masking ratio (15%) in the $R^2$MAE + CL setting.

**Training procedure.** Models were trained for 30000 steps using the defaults in [16] with AdamW optimizer, learning rate 1e-4, and effective batch size 2048. All experiments used PyTorch on 4 NVIDIA 6000 Ada GPUs with fixed random seeds. Each experiment takes 6.5 hours to complete.

**Evaluation.** We evaluated different models' performance in zero-shot predictions of pathological missense (Clinvar pathologic versus GnomAD common) and regulatory (OMIM pathologic versus GnomAD common) variants [60–62]. The inference was performed by `vep.inference` implemented in [16]. The evaluation sets as well as the scores of alternative models shown in Table 1 [55–59, 39] are provided by the original GPN-MSA work on Huggingface. Partial AUROC (max FPR 0.001) was used in the OMIM evaluation to account for high imbalance of positive and negative classes.

## C.8 Single-cell gene expression models

**Dataset.** We employed the Human Lung Cell Atlas dataset [63] and human brain MTG SEA-AD dataset [64]. Both datasets were downloaded from the CellXGene portal and were subsetted to 5000 highly variable genes (HVGs) using the default procedure in Scanpy [76]. For the HLCA dataset, we further filtered out cells that have fewer than 20 of these HVGs. After preprocessing, these datasets have 2161082 and 1378211 cells respectively, along with metadata of fine-grained cell types, disease/Alzheimer status (Alzheimer's Disease Neuropathologic Change, ADNC), and age labels.

**Model architecture.** For all settings that require pretraining from scratch, we implemented a 5-layer MLP encoder-decoder based architecture described as follows. The model receives corrupted count matrix $\boldsymbol{X}_{\text{mask}} \in \mathbb{R}^{n_{\text{cells}} \times n_{\text{input}}}$ as input, where $n_{\text{input}} = 5000$:

- **Latent encoder** $E_z$ (3-layer-MLP with $n_{\text{hidden}}$ units and ReLU activations, the final layer being a linear projection from $n_{\text{hidden}}$ to $n_{\text{latent}}$) maps logarithmically transformed (corrupted) counts together with the dataset batch covariate to the latent space.

- **Decoder** $D$ (2-layer-MLP with $n_{\text{hidden}}$ units and ReLU activations, the final layer being a projection from $n_{\text{hidden}}$ to $n_{\text{input}}$ with softmax activation) receives the embedding, observed library size, and batch covariates and produces negative-binomial parameters $(\mu_g, \theta_g)$ for every gene $g$ [77]. Batch normalizations are used in both encoders and decoders.

- **Objective.** The objective is defined as the average negative reconstruction likelihood for the masked genes in the input data: $L = -\frac{1}{|\text{mask}|} \sum_{g \in \text{mask}} \log \text{NB}(x_g \mid \mu_g, \theta_g)$.

In all implemented models, we set $n_{\text{hidden}} = 2000$ and $n_{\text{latent}} = 1000$. These are much higher values than those of typical scVI models [78, 77] and are comparable to latent space sizes in recent single-cell foundation models [17, 21].

**Mask pretraining schemes.** Before encoding, the raw count matrix $X$ is converted to the masked matrix $X_{\text{mask}}$ by one of following strategies:

- **Fixed MR**. A constant fraction $\rho$ of gene columns in the count matrix sampled once per mini-batch is replaced by zero.

- **Dynamic MR**. Same as the fixed MR case, except that the masking ratio follows a linear decay $\rho_t = \max\{\rho_{\min}, \rho_{\max} - \rho_{\max} t\,\lambda_{\text{decay}}\}$. Here $t$ represents the number of training steps. Here $\rho_{\max} = 0.5$ and $\rho_{\min} = 0.1$, with $\lambda$ an dataset-specific parameter to enforce linear decay throughout training (1/300000 for HLCA, 1/150000 for SEA-AD).

- **MDLM**. For every mini-batch, a masking ratio $\rho \sim \mathcal{U}(0,1)$ is sampled.

- **R$^2$MAE**. For every mini-batch, a masking ratio $\rho \sim \mathcal{U}(0.1, 0.5)$ is sampled.

- **R$^2$MAE + Dynamic MR**. For every mini-batch, we sample the masking ratio $\rho \sim \mathcal{U}(\rho_t, \rho_{\max})$, where $\rho_t = \max\{\rho_{\min}, \rho_{\max} - \rho_{\max} t\,\lambda_{\text{decay}}\}$, with the same parameter selections as the Dynamic MR setting.

For CL approaches, we first describe their combinations with R$^2$MAE, as their standalone implementation results from straightforward simplification of R$^2$MAE + CL schemes.

- **R$^2$MAE + CL (k=0)**. We implement an MLP layer that projects the original count matrix (we pass a transformed version, $\log((X/20)+1)$ in practice) to a two-layer MLP with hidden dims [128, 256] and ReLU activations, reshaped to output $\tilde{P} \in \mathbb{R}^{n_{\text{input}} \times 64}$. We further implement a row-wise projection layer with sigmoid activation plus 1e-9 to obtain strictly positive entries, which we term as $P \in \mathbb{R}^{n_{\text{input}} \times l}$. Finally, we apply $K = 4$ iterations of the non-square Sinkhorn operator so that each row sums to $l$ and each column to $n_{\text{input}}$: $M(y) = \text{Sinkhorn}^{(K)}(\sigma(P))$. For every mini-batch, we now sample the mask ratio in the length-$l$ vector $[0.10, 0.15, ..., 0.50]$, and select the column of $M(y)$ based on the sampled mask ratio index. Then the element-wise negative log-likelihood is multiplied by this column of $M(y)$ as the training objective. The newly implemented layers are optimized together after being fixed for 5000 training steps.

- **R$^2$MAE + CL**. Apart from the settings in R$^2$MAE + CL (k=0), we employed a dynamic multiplier to the gradient received by these additionally implemented layers, decreasing from 1 to $-1$ throughout its training with a linear decay from 30000 to 120000 training steps.

- **CL (k=0), CL.** These settings are effectively implemented by always selecting one fixed masking ratio in the above R$^2$MAE + CL (k=0) and R$^2$MAE + CL settings respectively.

- **scVI.** In this setting, we no longer mask the data, and instead formulate variational posteriors and train the model using the evidence lower bound (ELBO) objective [78, 77].

**Training procedure.** Models were trained for 50 epochs using Adam optimizer (learning rate 1e-3, weight decay 1e-4, batch size 400). 90% of data were selected as the training set and the remaining 10% was set as the validation set. All experiments used PyTorch on an NVIDIA 6000 GPU with fixed random seeds. Each experiment takes 1-2 hours to complete.

**Evaluation.** We evaluated different model embeddings' performance in identifying key metadata across donors through linear probing. Apart from previously described models, we used pretrained scGPT and UCE models to output zero-shot embeddings of the preprocessed datasets [17, 21]. For scGPT, the dataset is further log-normalized according to the instructions [17]. We also evaluated the linear probing performance of the log-normalized expression itself. We selected all cells from randomly sampled $\lfloor 0.6 \times \text{Total donors} \rfloor$ donors in the dataset as the training set, and the remaining cells as the test set. Reference control donors from the SEA-AD datasets were removed. We further removed data without corresponding metadata for the respective regression/classification tasks. Specifically, we removed cells whose cell type was labeled as Unknown for the fine-grained cell state classification task (HLCA), and removed cell types that do not contain Alzheimer-specific subtypes in the SEA-AD dataset; we removed cells without age labels (HLCA) or containing broad categories instead of exact ages (SEA-AD) for the age regression task. After each removal, we subset the training and test sets so that each donor comprises a maximum of $\frac{200000}{n_{\text{donor}}}$ in the set ($\frac{100000}{n_{\text{donor}}}$ numbers for cell type classification task in SEA-AD), to further balance cell numbers across donors. The training and test sets for cell type classification in HLCA is obtained by removal after subsetting.

We performed ridge regression for regression tasks and logistic regression for classification tasks. GridSearchCV was utilized for selecting the best regularization parameters (`np.logspace(0,8,20)` to minimize regression MSE, `np.logspace(-6,2,10)` to maximize classification balanced accuracy), and the training samples were separated by donors for five-fold cross-validation.

Finally, for classification tasks, we evaluated balanced accuracy and macro F1 score on the test set. For regression tasks, we evaluated both cell and donor level (obtained through averaging cell-level score per donor) Spearman $r$. The only exception is for the ADNC classification, where all methods perform poorly in terms of balanced accuracy and macro F1 scores. Therefore, we instead evaluated macro AUROC on both cell and donor levels.

For those models trained from scratch, we additionally evaluated their performance in reconstructing randomly masked genes in the pretraining validation set. The masking ratios evaluated are [0.1, 0.2, 0.3, 0.5, 0.7]. We calculated Pearson $r$ between the model output $\mu_g$ and log-normalized gene expression per cell, and then averaged the Pearson $r$ over all cells in the validation set (which stays the same across all methods tested).

# D  Appendix Figures

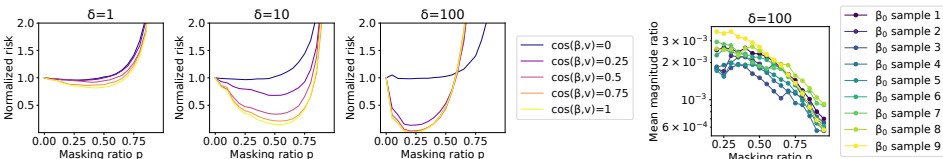

Figure 3: The spiked covariance model ($\boldsymbol{\Sigma} = \delta \boldsymbol{v}\boldsymbol{v}^\top + \boldsymbol{I}$), where $\boldsymbol{v} = \mathbb{1}/\sqrt{d}$. Plots of mean simulation test risk and prediction magnitude ratio ($\mathbb{E}\big[\|\boldsymbol{X}\hat{\boldsymbol{\beta}}_0\|^2|\tilde{\boldsymbol{X}}\big]/\mathbb{E}\big[\|\boldsymbol{X}\hat{\boldsymbol{\beta}}_1\|^2|\tilde{\boldsymbol{X}}\big]$ between $\cos(\boldsymbol{\beta}_1, \boldsymbol{v}) = 1$ and $\cos(\boldsymbol{\beta}_0, \boldsymbol{v}) = 0$) over 50 samples in the spiked covariance model against the masking ratio $p$. $n = 200, \gamma = 5$.

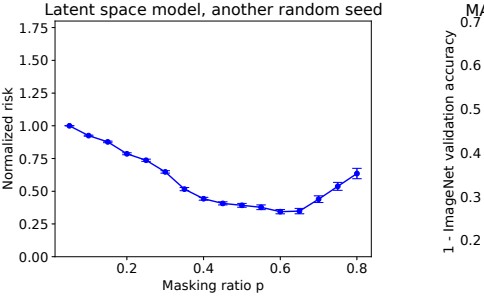
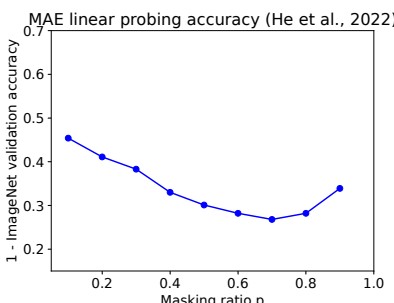

Figure 4: Plots of the normalized test risk of a latent space model and MAE linear probing accuracy on ImageNet1k [6] against masking ratio. The covariance $\boldsymbol{\Sigma}$ for the latent space model is another sample from the same generative process as in Fig. 1G. Note that there is a slight horizontal shift between the two curves.

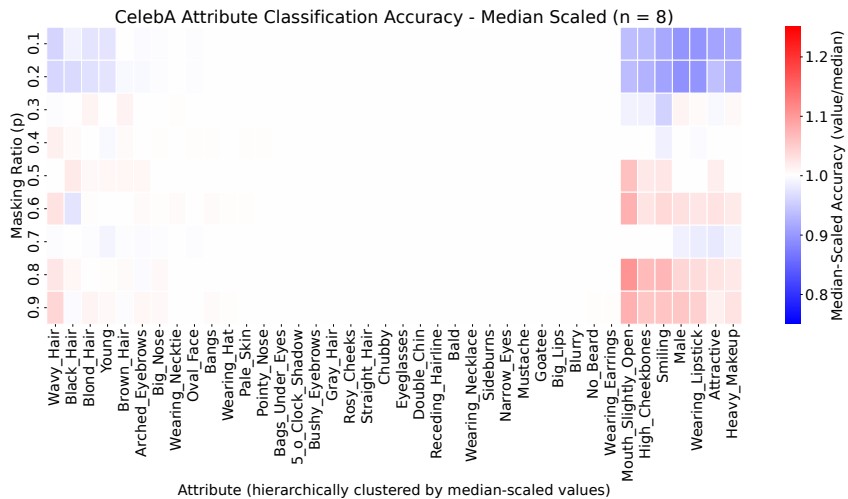

Figure 5: Median scaled accuracy of U-Net models (base channel = 8) on CeleBA classification tasks.

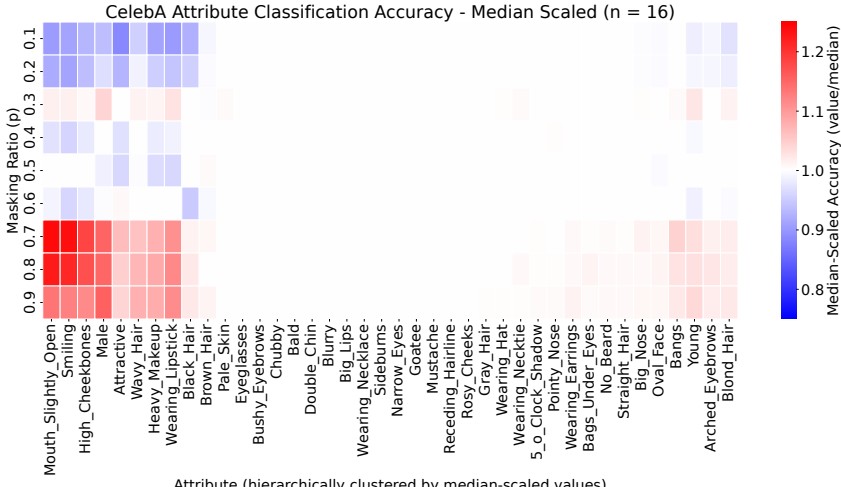

Figure 6: Median scaled accuracy of U-Net models (base channel = 16) on CeleBA classification tasks.

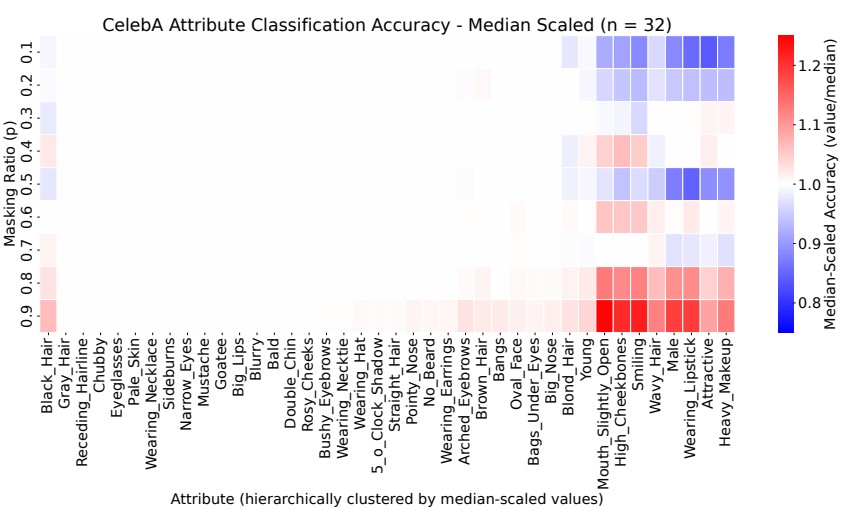

Figure 7: Median scaled accuracy of U-Net models (base channel = 32) on CeleBA classification tasks.

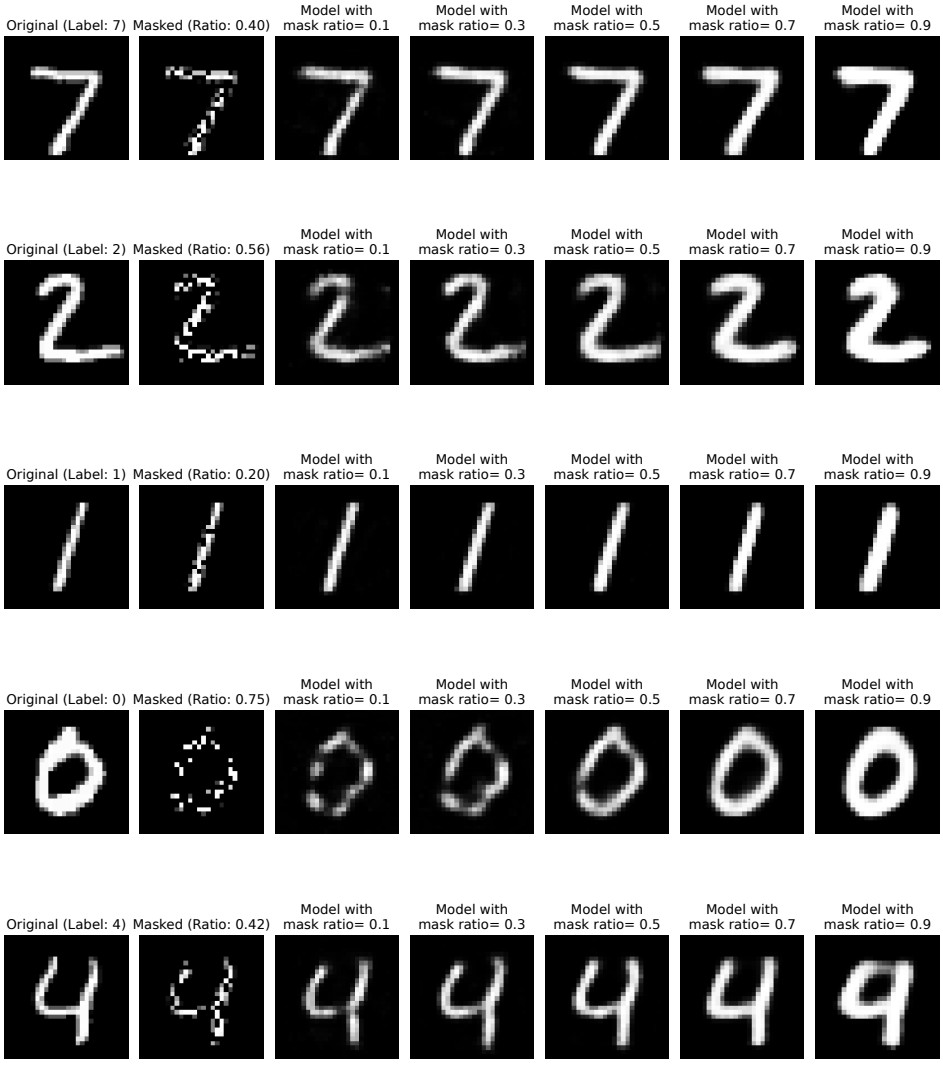

Figure 8: Comparison of overparametrized MLP reconstructions on MNIST data across different training mask ratios. Original digits (first column) and their masked versions (second column) are followed by reconstructions from models with identical architecture but varying mask ratios during training. The second layer hidden dim = 256 for all models.

Comparing Different Model Configurations on the Same Random Mask Test Set

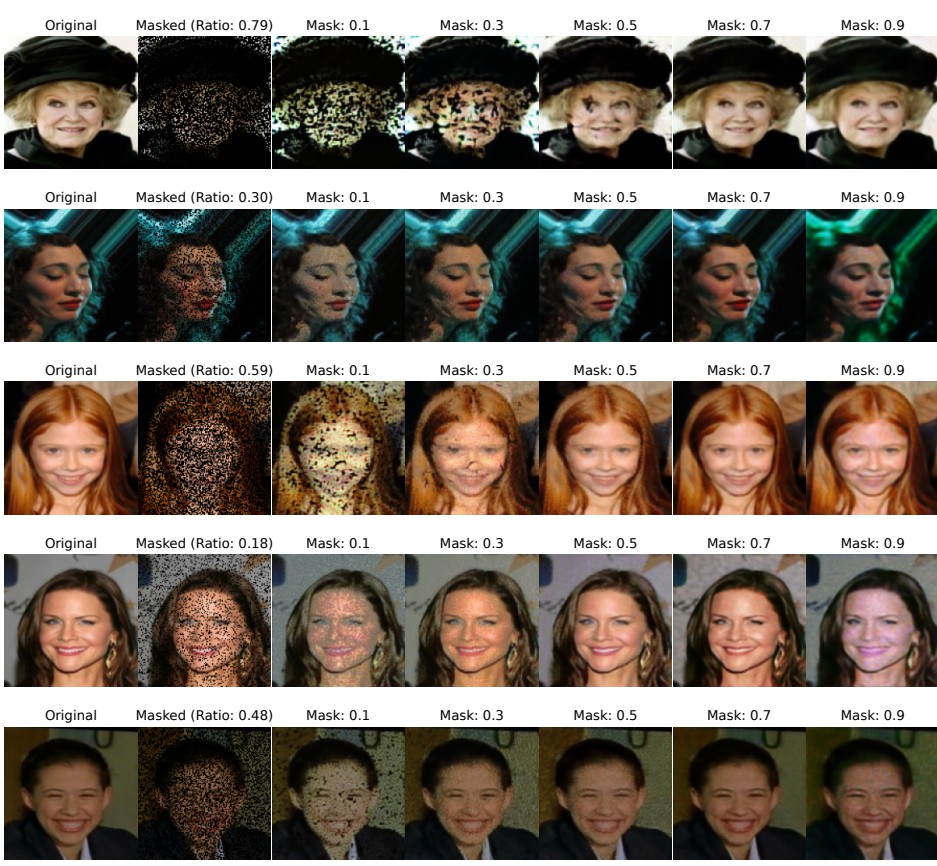

Model Base Channels: [16, 16, 16, 16, 16]

Figure 9: Comparison of U-net reconstructions on CeleBA across different training mask ratios. Original images (first column) and their masked versions (second column) are followed by reconstructions from models with identical architecture but varying mask ratios during training. All models have base channel = 16.

# E   Appendix Tables

Table 6: Comparison for different single-cell gene expression models trained on Human Lung Cell Atlas (HLCA). BAcc, Balanced Accuracy. For each specific task, pretraining scheme metrics outperforming optimal fixed masking ratio settings are labeled red.

| Methods | Cell state BAcc. | Cell state $F1_{macro}$ | Disease BAcc. | Disease $F1_{macro}$ | Age Spearman $r$ Cell | Age Spearman $r$ Donor | Avg performance Score | Avg performance Rank |
|---|---|---|---|---|---|---|---|---|
| Normalized exp. | 0.834 | 0.774 | 0.675 | 0.489 | 0.470 | 0.574 | 0.636 | 12.50 |
| scGPT (Lung) | 0.813 | 0.717 | 0.624 | 0.401 | 0.429 | 0.523 | 0.584 | 15.67 |
| scGPT (All) | 0.834 | 0.711 | 0.629 | 0.403 | 0.438 | 0.500 | 0.586 | 15.00 |
| scGPT (CP) | 0.816 | 0.696 | 0.613 | 0.389 | 0.431 | 0.521 | 0.578 | 16.67 |
| UCE (4L) | 0.808 | 0.702 | 0.631 | 0.417 | 0.436 | 0.518 | 0.585 | 15.50 |
| UCE (33L) | 0.800 | 0.699 | 0.619 | 0.419 | 0.447 | 0.540 | 0.587 | 15.50 |
| scVI | 0.897 | 0.804 | 0.767 | 0.626 | 0.556 | 0.618 | 0.711 | 9.67 |
| MAE (MR 25%) | 0.908 | 0.830 | 0.834 | 0.604 | 0.586 | 0.623 | 0.731 | 6.50 |
| *– MR 10%* | 0.915 | 0.802 | 0.851 | 0.635 | 0.582 | 0.609 | 0.732 | 5.67 |
| *– MR 50%* | 0.909 | 0.833 | 0.837 | 0.604 | 0.587 | 0.601 | 0.729 | 6.33 |
| MDLM | 0.903 | 0.806 | 0.829 | 0.560 | 0.577 | 0.622 | 0.716 | 8.83 |
| Dynamic MR | **0.919** | 0.829 | 0.850 | **0.651** | 0.571 | 0.597 | 0.736 | 5.00 |
| CL-MAE | 0.907 | 0.825 | 0.843 | 0.635 | 0.589 | 0.648 | 0.741 | 4.50 |
| CL-MAE ($k = 0$) | 0.801 | 0.667 | 0.773 | 0.493 | 0.530 | 0.563 | 0.638 | 13.83 |
| $R^2$**MAE** (Ours) | 0.915 | 0.812 | **0.853** | 0.651 | 0.595 | 0.641 | **0.744** | **2.83** |
| *+ Dynamic MR* | 0.914 | 0.842 | 0.835 | 0.597 | 0.616 | 0.658 | 0.744 | 4.17 |
| *+ CL* | 0.911 | 0.817 | 0.837 | 0.618 | 0.572 | 0.619 | 0.729 | 6.67 |
| *+ CL ($k = 0$)* | 0.911 | 0.805 | 0.840 | 0.630 | 0.590 | 0.646 | 0.737 | 5.00 |

Table 7: Comparison of random masking reconstruction Pearson $r$ across different single-cell gene expression models trained on brain MTG SEA-AD dataset. MR, Masking Ratio.

| Methods | MR 10% | MR 20% | MR 30% | MR 50% | MR 70% |
|---|---|---|---|---|---|
| scVI | 0.834 | 0.834 | 0.829 | 0.815 | 0.781 |
| MAE (MR 25%) | 0.843 | **0.847** | **0.846** | 0.840 | 0.819 |
| MAE (MR 10%) | 0.842 | 0.844 | 0.841 | 0.832 | 0.797 |
| MAE (MR 50%) | 0.841 | 0.845 | 0.843 | **0.842** | **0.830** |
| MDLM | 0.840 | 0.844 | 0.842 | 0.840 | 0.827 |
| Dynamic MR | 0.842 | 0.845 | 0.842 | 0.834 | 0.800 |
| CL-MAE | 0.841 | 0.844 | 0.845 | 0.838 | 0.817 |
| $R^2$**MAE** (Ours) | **0.846** | **0.847** | 0.845 | **0.842** | 0.826 |
| *+ Dynamic MR* | 0.844 | **0.847** | **0.846** | 0.841 | 0.824 |
| *+ CL* | 0.836 | 0.839 | 0.839 | 0.836 | 0.825 |
| *+ CL ($k = 0$)* | 0.844 | 0.846 | 0.845 | 0.840 | 0.823 |

Table 8: Comparison of random masking reconstruction Pearson $r$ across different single-cell gene expression models trained on Human Lung Cell Atlas (HLCA). MR, Masking Ratio.

| Methods | MR 10% | MR 20% | MR 30% | MR 50% | MR 70% |
|---|---|---|---|---|---|
| scVI | 0.712 | 0.744 | 0.746 | 0.733 | 0.698 |
| MAE (MR 25%) | 0.743 | **0.774** | 0.780 | 0.777 | 0.747 |
| MAE (MR 10%) | 0.733 | 0.770 | 0.777 | 0.769 | 0.733 |
| MAE (MR 50%) | 0.744 | 0.770 | 0.779 | **0.781** | **0.760** |
| MDLM | 0.729 | 0.760 | 0.766 | 0.764 | 0.739 |
| Dynamic MR | 0.742 | 0.772 | 0.777 | 0.772 | 0.737 |
| CL-MAE | 0.736 | 0.768 | 0.776 | 0.776 | 0.756 |
| CL-MAE ($k = 0$) | 0.174 | 0.131 | 0.109 | 0.090 | 0.086 |
| **$R^2$MAE** (Ours) | **0.746** | 0.773 | **0.782** | 0.778 | 0.752 |
| *+ Dynamic MR* | 0.743 | **0.774** | 0.781 | 0.779 | 0.753 |
| *+ CL* | 0.731 | 0.763 | 0.772 | 0.770 | 0.746 |
| *+ CL ($k = 0$)* | 0.733 | 0.767 | 0.777 | 0.770 | 0.742 |

Table 9: Normalized test risk of $R^2$MAE (MR range 0.4-0.5) against optimal fixed MR and mean MR settings across different random seeds for Beta covariance and latent space models. The ground truth signal $\beta$ is set to be the 10th quantile eigenvector of covariance $\Sigma$ in all cases. $n = 200$, $\gamma = 5$.

| | Beta Covariance Model | | | | Latent Space Model | | | |
|---|---|---|---|---|---|---|---|---|
| Seed | Best MR | Min Risk | MR 45% | $R^2$MAE | Best MR | Min Risk | MR 45% | $R^2$MAE |
| 2 | 0.43 | 0.859 | 0.865 | **0.855** | 0.45 | 0.826 | 0.826 | **0.823** |
| 12 | 0.43 | 0.863 | 0.865 | **0.859** | 0.42 | 0.832 | 0.837 | **0.830** |
| 22 | 0.54 | **0.862** | 0.898 | 0.890 | 0.37 | 0.848 | 0.851 | **0.842** |
| 32 | 0.43 | 0.817 | 0.822 | **0.814** | 0.33 | **0.852** | 0.865 | 0.859 |
| 42 | 0.36 | **0.817** | 0.830 | 0.819 | 0.43 | 0.814 | 0.818 | **0.806** |

