# OpenReview forum: "Understanding and Enhancing Mask-Based Pretraining towards Universal Representations"
_NeurIPS.cc/2025/Conference — NeurIPS 2025 poster_

### Official Review · Reviewer_T5Vd · 2025-06-24

**Clarity:** 2
**Significance:** 2
**Originality:** 2
**Rating:** 4
**Confidence:** 4

**Summary:**

This paper investigates the theoretical foundation and empirical behavior of mask-based pretraining, which is widely used in self-supervised learning for language, vision, and biological data. The authors propose a unified theoretical framework based on high-dimensional linear regression (including spiked covariance models) to explain why and when mask-based pretraining is effective, and how the optimal mask ratio depends on data and model size. Building on these insights, they introduce a simple pretraining method called R²MAE (Randomly Random Mask AutoEncoding), where the mask ratio is randomly sampled for each training batch. The method is evaluated on DNA sequence and single-cell gene expression tasks, showing modest improvements over traditional approaches. While the theoretical analysis is mathematically solid within linear settings, its extension to deep nonlinear models remains questionable, and the empirical gains are relatively small.

**Questions:**

See in Weaknesses

**Ethical Concerns:**

["NO or VERY MINOR ethics concerns only"]

**Final Justification:**

The consistent cross-domain improvements and notable 16.5% improvement in OMIM regulatory variant prediction demonstrate method robustness. Given these substantial additions and the potential for improved presentation, I am inclined to raise my assessment to borderline accept.

**Limitations:**

See in Weaknesses

**Quality:**

2

**Strengths And Weaknesses:**

## Strengths

- The approach is easy to implement and performs well on some biological datasets, showing minor improvements over standard random masking.
- The paper attempts to provide a theoretical explanation for the mask-based pretraining phenomenon, which is a meaningful direction.

## Weaknesses

1. The main idea is just randomizing the mask ratio, which is very similar to dynamic masking and scheduled masking from earlier works [1,2,3,4,5]. The paper doesn’t fairly compare or cite these, and the novelty is overclaimed.
2. All the theoretical analysis is based on linear models, while real masked autoencoders are highly nonlinear deep networks. There’s a big gap between the theory and practice, so the “universality” claim isn’t convincing [1,6].
3. Experiments are only on a few models and datasets, with no evidence for challenging or large-scale biological tasks. The generality claim is not well supported [1,7].
4. The biological results show only marginal improvements, and there’s almost no discussion on whether these gains are meaningful for real science or medicine [7].
5. The Introduction is poorly constructed—it promises universal advances for language and vision, but the paper only does very limited, niche biological experiments (DNA sequence and single-cell gene expression) or toy datasets like MNIST, with absolutely no evaluation on general language or vision tasks. There’s no bio-specific design either; the method is generic and not tailored at all.
6. The writing is lengthy and dense, with key points buried in technical details and formulas.
7. There are issues in the theoretical proofs. For example, in the spiked covariance analysis, the authors make strong assumptions (like high-dimensional limits and feature alignment) without really checking if they hold in practice. Many derivations just assume ideal Gaussian data and feature structures, but small deviations in real data can break the conclusions. The claim that an optimal mask ratio always exists under spiked covariance only holds under strict conditions, which are not verified in their experiments. Much of the analysis is quite similar to recent works like MaskTwins [8], which also use linear model or spiked covariance for mask theory, and while those papers have their own limitations, they actually discuss model assumptions and limitations more carefully. This paper overstates its theoretical contribution and doesn't clearly position itself with respect to these existing analyses.

### References

1. Ankner et al., Dynamic masking rate schedules for MLM pretraining, arXiv:2305.15096 (2023). [link](https://arxiv.org/abs/2305.15096)
2. Yang et al., Learning better masking for better language model pre-training, arXiv:2208.10806 (2022). [link](https://arxiv.org/abs/2208.10806)
3. Wettig et al., Should you mask 15% in masked language modeling?, arXiv:2202.08005 (2022). [link](https://arxiv.org/abs/2202.08005)
4. Bao et al., BEiT: BERT Pre-Training of Image Transformers, arXiv:2106.08254 (2021). [link](https://arxiv.org/abs/2106.08254)
5. He et al., Masked Autoencoders Are Scalable Vision Learners, CVPR 2022. [link](https://arxiv.org/abs/2111.06377)
6. Chen et al., TokenUnify: Scalable Autoregressive Visual Pre-training with Mixture Token Prediction, arXiv:2405.16847 (2024). [link](https://arxiv.org/abs/2405.16847)
7. Dalla-Torre et al., Nucleotide transformer: building and evaluating robust foundation models for human genomics, Nature Methods, 2025. [link](https://www.nature.com/articles/s41592-024-02523-z)
8. Wang et al., MaskTwins: Dual-form Complementary Masking for Domain-Adaptive Image Segmentation, ICML 2025. [link](https://openreview.net/pdf?id=9CpeZ8BzPO)

---

> ### Author Rebuttal · Authors · 2025-07-31
>
> Thank you very much for your comments. Our rebuttal is organized as follows. We will first present new results that address general feedback from all reviewers. They include new experiments on vision and language models and a strengthened connection between R$^2$MAE and our theoretical framework. We will next provide a point-by-point response to specific concerns.
>
> ### **New Results 1: Experiments of R$^2$MAE on Vision and Language Models**
>
> In general, reviewers appreciated our empirical study but expressed curiosity about whether R$^2$MAE would achieve SOTA performance in more established domains like vision and language modeling. We were fortunate to gain access to additional computational resources during the rebuttal period to perform these experiments.
>
> **Vision:** Our implementations closely follow established practices from the vision MAE work [1]. We implemented different mask ratio settings on the ViT-base MAE model using their official Pytorch codebase. The settings include: 1. Default MAE (constant 0.75 mask ratio, MR); 2. R$^2$MAE (p ~ U(0.6, 0.9)); 3. Dynamic MR [3] (linearly decreasing MR from 0.9 to 0.6); 4. High (0.9) and low (0.5) fixed MR baselines. Given the time limit, all models were trained for 150 epochs. While this is shorter than the 800 epochs in [1], their analysis shows predictable improvements with longer training schedules. Therefore, while our absolute accuracies may be suboptimal, we expect the relative performance across settings to be comparable and the conclusion to hold if trained longer. All other hyperparameters follow the defaults in [1]. Models were then fine-tuned for classification for 100 epochs, following [1].
>
> Our results below show R$^2$MAE achieves the best performance in both top-1 and top-5 accuracy. We found that R$^2$MAE introduces fluctuations in the pretraining loss, likely due to variable sequence lengths since MAE only encodes unmasked tokens. This presents a potential area for even further improvement.
>
> | ViT-base | Acc@1 | Acc@5 |
> |:---|:---:|:---:|
> | MAE default (MR 0.75) | 81.97 | 96.02 |
> | fixed MR 0.9 | 81.20 | 95.68 |
> | fixed MR 0.5 | 81.80 | 95.93 |
> | Dynamic MR | 81.97 | 96.04 |
> | R$^2$MAE | **82.00** | **96.05** |
>
> **Language:** We trained RoBERTa-base and RoBERTa-medium models on the FineWeb dataset for 10B tokens (max seq length 128, comparable to [2]) and fine-tuned them on GLUE benchmarks (MNLI, QQP, SST-2, QNLI). We evaluated: 1. Default MLM (MR 0.15); 2. R$^2$MAE (p ~ U(0.15, 0.4)); 3. Dynamic MR [3] (0.4 to 0.15); 4. MLM with a fixed 0.4 MR. Fine-tuning accuracies are comparable to those in [2]. The results show R$^2$MAE achieves the best overall rank, with a more pronounced advantage in the larger RoBERTa-Base model.
>
> | RoBERTa-Medium (52M) | MNLI | QQP | SST-2 | QNLI | Mean Rank |
> |:---|:---:|:---:|:---:|:---:|:---:|
> | MLM default (MR 0.15) | 80.8 | 89.8 | 89.9 | 86.3 | 3.25 |
> | Fixed MR 0.4 | 80.3 | 89.7 | 90.1 | 86.6 | 3.50 |
> | Dynamic MR | 80.8 | **90.1** | 90.5 | **87.1** | 1.50 |
> | R$^2$MAE | **80.9** | **90.1** | **90.6** | 86.7 | **1.25** |
>
> | RoBERTa-Base (125M) | MNLI | QQP | SST-2 | QNLI | Mean Rank |
> |:---|:---:|:---:|:---:|:---:|:---:|
> | MLM default (MR 0.15) | 81.5 | 90.7 | 91.7 | 87.8 | 3.00 |
> | Fixed MR 0.4 | 81.7 | 90.7 | 91.2 | 88.5 | 3.00 |
> | Dynamic MR | 81.8 | 90.7 | 91.4 | **89.1** | 2.00 |
> | R$^2$MAE | **81.9** | **90.8** | **91.9** | 88.6 | **1.25** |
>
> **Note:** In these fine-tuning evaluations, the advantage of R$^2$MAE appears smaller than in our linear probing or zero-shot evaluations. This is expected, as the identical fine-tuning pipeline can reduce differences from pre-training. Nevertheless, the consistent advantage of R$^2$MAE across tasks and domains strongly supports its effectiveness and generalizability.
>
> ### **New Results 2: Explicit Connection between R$^2$MAE and the High-Dimensional Linear Regression Framework**
>
> Inspired by reviewer comments, we evaluated R$^2$MAE within our linear model framework. Specifically, we simulated the latent space model from Fig. 1E. Intriguingly, we observed that for features of different strengths, R$^2$MAE's risk tends to align with the *optimal* risk, with its "effective masking ratio" decreasing as feature strength decays.
>
> | Normalized test risk | p=0.45 | p=0.5 | p=0.55 | p=0.6 | R$^2$MAE, p~U(0.45,0.6) | Effective p of R$^2$MAE |
> |---|---|---|---|---|---|---|
> | β = 1st Eig | 0.687 | 0.671 | 0.666 | 0.671 | 0.665 | ~0.55 |
> | β = 2nd Eig | 0.751 | 0.748 | 0.752 | 0.772 | 0.750 | 0.5-0.55 |
> | β = 3rd Eig | 0.839 | 0.842 | 0.849 | 0.872 | 0.842 | ~0.5 |
> | β = 4th Eig | 0.880 | 0.889 | 0.898 | 0.929 | 0.888 | 0.45-0.5 |
>
>
> Our new results support the intuition that R$^2$MAE achieves a “feature-adaptive” masking ratio that approximates dataset-specific optimal masking ratios, allowing it to outperform any fixed ratio in settings with multi-scale features. This is validated by real data (Tables 4 and 5), where R$^2$MAE achieves near-optimal reconstruction across its entire masking range (0.1-0.5), while fixed-rate settings only well cover a smaller range. This result aligns with the robust advantage of R$^2$MAE and strengthens the value of our work.
>
> ### **Point-by-point Response to Specific Concerns**
>
> **W1. Relevant works:** Thank you for the comment. We would like to note that we indeed cited all relevant literature mentioned in the review. Furthermore, we have a dedicated section (Appendix A.1) discussing previous works, and our benchmarks included several of these approaches. While R$^2$MAE is simple, it remains uncovered by previous work. Our theoretical framework is also novel in clarifying unexplained phenomena in mask-based pre-training. Given this, we respectfully disagree with the judgment that our work “does not fairly compare or cite these, and the novelty is overclaimed.” We are happy to address more specific comments if provided.
>
> **W2. Gap between the theory and practice:** We discussed this gap in the “Relation with real network optimization” part of Section 3.1. Despite the gap, using linear models to explain behaviors in neural networks is a well-established approach in the literature [4, 5]. To verify our findings generalize, we performed extensive evaluations on MLPs, CNNs, and Transformers (Figs. 1-2). Importantly, these evaluations highlighted several phenomena that are explained by our overparametrized linear model but not by alternative frameworks, suggesting our model serves as a valuable “working theory” of masked pre-training.
>
> **W3-W5. On the limitations of experimental results:**
> 1.  **On "marginal" improvement:** We respectfully disagree. R$^2$MAE provides consistent and, in several cases, substantial performance improvements. For instance, we improved the PRAUC for the OMIM regulatory variant prediction task by 16.5% over the SOTA model reported in [6]. In brain gene expression data, we improved cell-level age prediction Spearman's r by 10.9% over an optimal MAE. While improvements are smaller in a number of cases, they are consistent across metrics, tasks, and data domains, including the new vision/language results. This improvement is a solid advance for a pre-training scheme that does not alter model architecture.
>
> 2.  **On task relevance and scientific discovery:** Our tasks are well-established benchmarks. The DNA variant prediction task is widely used for evaluating DNA models (e.g., Nucleotide Transformer [NT] cited by the reviewer). The OMIM benchmark itself is a large-scale task evaluating over two million DNA sequences. Our experiments span a range of evaluation scenarios from zero-shot (DNA) and linear probing (gene expression) to fine-tuning (ViT, RoBERTa).
>
> 3.  **On the lack of vision/language results:** We agree this is an important point and have now provided these results in the first section of our response.
>
> **W5-W6. On paper writing:** We appreciate the feedback. We are unclear how to address general criticisms like “lengthy and dense writing” but would be happy to modify our text if more specific suggestions are provided. Our introduction aims to motivate a general framework, and we did not find sentences that “promise universal advances for language and vision” as criticized. We believe these comments may stem from a misunderstanding.
>
> **W7. On theory assumptions and the MaskTwins paper:** Our theoretical work does not aim to prove properties of neural networks directly. Instead, we define a tractable linear model, analyze its properties, and then empirically evaluate if these properties hold in real networks. We empirically verified our findings across multiple data models (Fig. 1E) and network architectures (Fig. 2), in order to support our framework's validity.
>
> We were not aware of MaskTwins as it was published after our submission deadline, but we will cite it in the next version. We have reviewed the paper and find the technical approach to be very different. MaskTwins uses compression sensing theory to derive bounds for signal discovery in a signal+noise model. Our work uses random matrix theory to characterize test risk in an overparameterized regression setting. The goal is also distinct. We would be happy to revise any part of our manuscript that the reviewer feels is an overstatement.
>
> Thanks again for your thorough evaluation. Please let us know if there are any additional concerns.
>
> [1] He, Kaiming, et al. "Masked autoencoders are scalable vision learners." CVPR. 2022.
> [2] Wettig, Alexander, et al. "Should You Mask 15% in Masked Language Modeling?." EACL. 2023.
> [3] Ankner, Zachary, et al. "Dynamic Masking Rate Schedules for MLM Pretraining." EACL. 2024.
> [4] Hastie, Trevor, et al. "Surprises in high-dimensional ridgeless least squares interpolation." Annals of statistics. 2022.
> [5] Bahri, Yasaman, et al. "Explaining neural scaling laws." PNAS. 2024.
> [6]  Benegas, Gonzalo, et al. "A DNA language model based on multispecies alignment predicts the effects of genome-wide variants." Nature Biotechnology, 2025.

---

> > ### Comment · Reviewer_T5Vd · 2025-08-05
> > **Updated Review Assessment**
> >
> > Thank you for the comprehensive rebuttal. The new vision and language modeling results significantly strengthen your work by demonstrating R²MAE's effectiveness beyond biological domains. The ViT-Base and RoBERTa experiments show consistent improvements across domains, and the explicit connection between R²MAE and your theoretical framework provides valuable "feature-adaptive" masking intuition.
> >
> > Regarding the writing concerns, I suggest restructuring the main text for better conference readability. The detailed theoretical assumptions (Reduced linear model, Spiked covariance model) should be briefly mentioned in the main text with comprehensive details moved to supplementary materials. The main text should prioritize presenting your method and empirical results clearly. The current structure, while mathematically rigorous, is better suited for journal venues like JMLR rather than conferences where readability and communication efficiency are paramount.
> >
> > The consistent cross-domain improvements and notable 16.5% improvement in OMIM regulatory variant prediction demonstrate method robustness. Given these substantial additions and the potential for improved presentation, I am inclined to raise my assessment to **borderline accept**.
> >
> > **Updated Rating:** 4: Borderline accept

---

> > > ### Author Response · Authors · 2025-08-05
> > >
> > > Thank you very much. We agree that the writing in our current theory section could be improved, especially for a broader audience. We will thoroughly revise this section in the updated manuscript.

---

### Official Review · Reviewer_CpLL · 2025-07-03

**Clarity:** 4
**Significance:** 3
**Originality:** 3
**Rating:** 5
**Confidence:** 5

**Summary:**

This paper presents a theoretical framework for understanding mask-based pre-training through the lens of high-dimensional linear regression. The authors demonstrate that masked autoencoding can be characterized by the test risk of ridgeless regression with masked covariates. Following these insights, they propose R2MAE (Randomly Random Mask AutoEncoding), which samples masking ratios uniformly during training. Their method achieves state-of-the-art results on DNA sequence and single-cell gene expression tasks.

**Questions:**

What is the relationship between this method and autoregressive modeling? Fully causal masks in fact also expose a uniformly sample masking ratio to every token in the sequence (first token has full masking, last token has almost no masking). Connections to this paradigm would be very valuable for further understanding.

How does R2MAE's performance scale with dataset size and diversity? A test on an even more studied domain with clearly highly non-homogenous data such as images would be even more convincing and would make me upgrade my score to a 6. If the method fails to improve performance on image encoding (beat MAE) why is that the case?

How does your framework relate to recent work on spatially-structured masking strategies in vision, such as SiamMAE (Gupta et al., 2023) or CWM (Unifying Machine Vision ... Bear et al., 2023) which utilize specifically structured masking strategies to obtain specific behaviors from the model (such as separation of appearance and motion information) which are not present in uniform, non biased masking approaches?

**Ethical Concerns:**

["NO or VERY MINOR ethics concerns only"]

**Final Justification:**

I maintain my score of accept

**Limitations:**

Yes

**Quality:**

4

**Strengths And Weaknesses:**

Strengths:
- The reduction of masked pretraining to high-dimensional linear regression provides tractable closed-form analysis while reproducing key empirical phenomena like optimal masking ratios and plateau behaviors.
- Theoretical predictions are validated across multiple architectures (MLPs, CNNs, Transformers) and datasets, confirming that benefits only emerge in overparameterized regimes.
- Method achieves consistent improvements over existing methods on DNA and single-cell tasks
- Rigorous mathematical analysis

Weaknesses:
- The proposed method (R^2MAE) is only evaluated on biological data despite claims of universality, with no results on standard vision or language benchmarks.
- The linear model ignores optimization dynamics, architectural inductive biases, and nonlinear feature interactions that may be crucial to masked pretraining

---

> ### Author Rebuttal · Authors · 2025-07-31
>
> Thank you very much for your comments. Our rebuttal is organized as follows. We will first present new results that address general feedback from all reviewers. They include new experiments on vision and language models and a strengthened connection between R$^2$MAE and our theoretical framework. We will next provide a point-by-point response to specific concerns.
>
> ### **New Results 1: Experiments of R$^2$MAE on Vision and Language Models**
>
> In general, reviewers appreciated our empirical study but expressed curiosity about whether R$^2$MAE would achieve SOTA performance in more established domains like vision and language modeling. We were fortunate to gain access to additional computational resources during the rebuttal period to perform these experiments.
>
> **Vision:** Our implementations closely follow established practices from the vision MAE work [1]. We implemented different mask ratio settings on the ViT-base MAE model using their official Pytorch codebase. The settings include: 1. Default MAE (constant 0.75 mask ratio, MR); 2. R$^2$MAE (p ~ U(0.6, 0.9)); 3. Dynamic MR [3] (linearly decreasing MR from 0.9 to 0.6); 4. High (0.9) and low (0.5) fixed MR baselines. Given the time limit, all models were trained for 150 epochs. While this is shorter than the 800 epochs in [1], their analysis shows predictable improvements with longer training schedules. Therefore, while our absolute accuracies may be suboptimal, we expect the relative performance across settings to be comparable and the conclusion to hold if trained longer. All other hyperparameters follow the defaults in [1]. Models were then fine-tuned for classification for 100 epochs, following [1].
>
> Our results below show R$^2$MAE achieves the best performance in both top-1 and top-5 accuracy. We found that R$^2$MAE introduces fluctuations in the pretraining loss, likely due to variable sequence lengths since MAE only encodes unmasked tokens. This presents a potential area for even further improvement.
>
> | ViT-base | Acc@1 | Acc@5 |
> |:---|:---:|:---:|
> | MAE default (MR 0.75) | 81.97 | 96.02 |
> | fixed MR 0.9 | 81.20 | 95.68 |
> | fixed MR 0.5 | 81.80 | 95.93 |
> | Dynamic MR | 81.97 | 96.04 |
> | R$^2$MAE | **82.00** | **96.05** |
>
> **Language:** We trained RoBERTa-base and RoBERTa-medium models on the FineWeb dataset for 10B tokens (max seq length 128, comparable to [2]) and fine-tuned them on GLUE benchmarks (MNLI, QQP, SST-2, QNLI). We evaluated: 1. Default MLM (MR 0.15); 2. R$^2$MAE (p ~ U(0.15, 0.4)); 3. Dynamic MR [3] (0.4 to 0.15); 4. MLM with a fixed 0.4 MR. Fine-tuning accuracies are comparable to those in [2]. The results show R$^2$MAE achieves the best overall rank, with a more pronounced advantage in the larger RoBERTa-Base model.
>
> | RoBERTa-Medium (52M) | MNLI | QQP | SST-2 | QNLI | Mean Rank |
> |:---|:---:|:---:|:---:|:---:|:---:|
> | MLM default (MR 0.15) | 80.8 | 89.8 | 89.9 | 86.3 | 3.25 |
> | Fixed MR 0.4 | 80.3 | 89.7 | 90.1 | 86.6 | 3.50 |
> | Dynamic MR | 80.8 | **90.1** | 90.5 | **87.1** | 1.50 |
> | R$^2$MAE | **80.9** | **90.1** | **90.6** | 86.7 | **1.25** |
>
> | RoBERTa-Base (125M) | MNLI | QQP | SST-2 | QNLI | Mean Rank |
> |:---|:---:|:---:|:---:|:---:|:---:|
> | MLM default (MR 0.15) | 81.5 | 90.7 | 91.7 | 87.8 | 3.00 |
> | Fixed MR 0.4 | 81.7 | 90.7 | 91.2 | 88.5 | 3.00 |
> | Dynamic MR | 81.8 | 90.7 | 91.4 | **89.1** | 2.00 |
> | R$^2$MAE | **81.9** | **90.8** | **91.9** | 88.6 | **1.25** |
>
> **Note:** In these fine-tuning evaluations, the advantage of R$^2$MAE appears smaller than in our linear probing or zero-shot evaluations. This is expected, as the identical fine-tuning pipeline can reduce differences from pre-training. Nevertheless, the consistent advantage of R$^2$MAE across tasks and domains strongly supports its effectiveness and generalizability.
>
> ### **New Results 2: Explicit Connection between R$^2$MAE and the High-Dimensional Linear Regression Framework**
>
> Inspired by reviewer comments, we evaluated R$^2$MAE within our linear model framework. Specifically, we simulated the latent space model from Fig. 1E. Intriguingly, we observed that for features of different strengths, R$^2$MAE's risk tends to align with the *optimal* risk, with its "effective masking ratio" decreasing as feature strength decays.
>
> | Normalized test risk | p=0.45 | p=0.5 | p=0.55 | p=0.6 | R$^2$MAE, p~U(0.45,0.6) | Effective p of R$^2$MAE |
> |---|---|---|---|---|---|---|
> | β = 1st Eig | 0.687 | 0.671 | 0.666 | 0.671 | 0.665 | ~0.55 |
> | β = 2nd Eig | 0.751 | 0.748 | 0.752 | 0.772 | 0.750 | 0.5-0.55 |
> | β = 3rd Eig | 0.839 | 0.842 | 0.849 | 0.872 | 0.842 | ~0.5 |
> | β = 4th Eig | 0.880 | 0.889 | 0.898 | 0.929 | 0.888 | 0.45-0.5 |
>
>
> Our new results support the intuition that R$^2$MAE achieves a “feature-adaptive” masking ratio that approximates dataset-specific optimal masking ratios, allowing it to outperform any fixed ratio in settings with multi-scale features. This is validated by real data (Tables 4 and 5), where R$^2$MAE achieves near-optimal reconstruction across its entire masking range (0.1-0.5), while fixed-rate settings only well cover a smaller range. This result aligns with the robust advantage of R$^2$MAE and strengthens the value of our work.
>
> ### **Point-by-point response to other specific concerns**
>
> In response to the reviewer’s comments, we have performed extensive experiments on vision and language masked pretraining models. Our responses to each question are provided as follows.
>
> **Q1. Relation to next-token prediction.** Thank you for this very insightful question. Despite the apparent relevance, we believe next-token prediction cannot be adequately addressed by our current theoretical framework. This is because next-token prediction is a token-wise, rather than sample-wise, prediction task. These prediction tasks within an input sequence are highly dependent. This constitutes a clear gap with our current setup, and addressing it will require future research. This is an important point, and we will discuss it in the next version of our paper.
>
> **Q2. R²MAE's performance on larger and more diverse datasets.** Our results on vision and language modeling show that R²MAE provides consistent improvements (albeit sometimes marginal), even for larger models trained on diverse datasets across various data domains. Our evaluations on two configurations of RoBERTa models show that R²MAE provides consistent improvements across model configurations, whereas the performance of other settings varies. These findings align with our results on biological data and strengthen our conclusions.
>
> **Q3. Comparison with structured masking strategies.** This is another great point. The cited structured masking strategies are designed to enforce the model to learn specific patterns in the data of interest. Therefore, they may not be directly comparable to ordinary MAE (or R²MAE) pretraining. Nevertheless, the masking procedure in the cited works is an ordinary random mask with a fixed rate, a setting where incorporating R²MAE could potentially improve performance.
>
> Meanwhile, a variety of structured masking strategies have been proposed to directly improve general pretraining models, as discussed in the literature and in Appendix A.1 of our paper. The RoBERTa masking study [2] compared several masking strategies and concluded that these strategies do not outperform simply tuning the masking rate. Our evaluations of learnable masking strategies for biological data, as well as a parallel study on biological data [4], reached the same conclusion.
>
> Thank you again for your thorough evaluation. Please let us know if our responses have fully addressed your concerns or if any additional information is needed.
>
> [1] He, Kaiming, et al. "Masked autoencoders are scalable vision learners." CVPR. 2022.
> [2] Wettig, Alexander, et al. "Should You Mask 15% in Masked Language Modeling?." EACL. 2023.
> [3] Ankner, Zachary, et al. "Dynamic Masking Rate Schedules for MLM Pretraining." EACL. 2024.
> [4] Richter, Till, et al. "Delineating the effective use of self-supervised learning in single-cell genomics." Nature Machine Intelligence. 2025.

---

> > ### Comment · Reviewer_CpLL · 2025-08-05
> >
> > Thank you for the additional results and the discussion. I hope that some of the points of relation to autoregressive modeling and various masking strategies get incorporated into a discussion section (or appendix) of the final paper as I think they provide clarify and connection.

---

> > > ### Author Response · Authors · 2025-08-05
> > >
> > > Thank you very much. We will revise the manuscript as suggested.

---

### Official Review · Reviewer_yjuN · 2025-07-04

**Clarity:** 3
**Significance:** 2
**Originality:** 3
**Rating:** 4
**Confidence:** 3

**Summary:**

This paper presents a novel theoretical framework to understand mask-based pretraining through the lens of high-dimensional ridgeless linear regression. The authors show that both qualitative and quantitative behaviors of masked autoencoding across domains (vision, language, biology) can be recapitulated through this simplified yet analytically tractable model. Based on insights from this theory, they propose a simple yet effective pretraining strategy called R2MAE (Randomly Random Mask AutoEncoding), which applies uniformly sampled masking ratios during pretraining. Empirical results on DNA sequence and single-cell gene expression data demonstrate that R2MAE outperforms existing state-of-the-art approaches across multiple metrics.

**Questions:**

1. Have you considered incorporating R2MAE into more general-purpose pretrained models (e.g., ViT or LLaMA) to evaluate generalization beyond biology?

2. My understanding is that R²MAE randomly samples a different masking ratio for each input from a uniform distribution—is this correct? If so, based on your theoretical analysis, is it possible to determine an optimal masking ratio for each individual sample, rather than relying on random sampling?

**Ethical Concerns:**

["NO or VERY MINOR ethics concerns only"]

**Final Justification:**

During the rebuttal, the authors address many of my concerns and I increased my score.

**Limitations:**

The main limitation lies in the strong assumption of a linear model and the minimal interaction between features, which may oversimplify the complex dependencies present in real-world deep networks.

**Quality:**

3

**Strengths And Weaknesses:**

Strengths:
1. Theoretical Contributions: The use of high-dimensional linear models to explain masking behaviors is theoretically grounded. The analysis reveals key phenomena such as optimal masking ratios, phase transitions, and performance plateaus. Closed-form expressions for test risk under various covariance structures (isotropic, spiked, general) are novel and informative.

2. Clear Empirical Validation: The authors validate theoretical predictions on diverse architectures (MLPs, CNNs, Transformers) and domains (vision, language, biology). Experiments on real biological data (DNA, single-cell) show that theoretical predictions align well with empirical trends.

3. Practical Impact via R2MAE: Despite its simplicity, R2MAE consistently improves performance over standard and dynamic masking schemes in biological settings. The strategy is easy to implement and generalizable.

Weaknesses:
1. Theoretical-Algorithmic Gap: While this work provides novel insights into masked pretraining, some of the underlying technical assumptions are quite strong. In particular, the reliance on a linear model and simplified data distribution assumptions may limit the direct applicability of the theoretical findings to practical deep learning settings involving nonlinear architectures and complex data structures.

2.  Simplified Feature Independence in $\beta$ Estimation:  The assumption that each regression target ($\beta_i$) is estimated independently or with minimal interaction may ignore feature co-adaptation, which is critical in practice.

3. Minimal Comparison on General Datasets: Most empirical validations beyond theory focus on DNA and single-cell data. It would strengthen the paper to include results on standard vision/language benchmarks with R2MAE (e.g., ImageNet, GLUE).

---

> ### Author Rebuttal · Authors · 2025-07-31
>
> Thank you very much for your comments. Our rebuttal is organized as follows. We will first present new results that address general feedback from all reviewers. They include new experiments on vision and language models and a strengthened connection between R$^2$MAE and our theoretical framework. We will next provide a point-by-point response to specific concerns.
>
> ### **New Results 1: Experiments of R$^2$MAE on Vision and Language Models**
>
> In general, reviewers appreciated our empirical study but expressed curiosity about whether R$^2$MAE would achieve SOTA performance in more established domains like vision and language modeling. We were fortunate to gain access to additional computational resources during the rebuttal period to perform these experiments.
>
> **Vision:** Our implementations closely follow established practices from the vision MAE work [1]. We implemented different mask ratio settings on the ViT-base MAE model using their official Pytorch codebase. The settings include: 1. Default MAE (constant 0.75 mask ratio, MR); 2. R$^2$MAE (p ~ U(0.6, 0.9)); 3. Dynamic MR [3] (linearly decreasing MR from 0.9 to 0.6); 4. High (0.9) and low (0.5) fixed MR baselines. Given the time limit, all models were trained for 150 epochs. While this is shorter than the 800 epochs in [1], their analysis shows predictable improvements with longer training schedules. Therefore, while our absolute accuracies may be suboptimal, we expect the relative performance across settings to be comparable and the conclusion to hold if trained longer. All other hyperparameters follow the defaults in [1]. Models were then fine-tuned for classification for 100 epochs, following [1].
>
> Our results below show R$^2$MAE achieves the best performance in both top-1 and top-5 accuracy. We found that R$^2$MAE introduces fluctuations in the pretraining loss, likely due to variable sequence lengths since MAE only encodes unmasked tokens. This presents a potential area for even further improvement.
>
> | ViT-base | Acc@1 | Acc@5 |
> |:---|:---:|:---:|
> | MAE default (MR 0.75) | 81.97 | 96.02 |
> | fixed MR 0.9 | 81.20 | 95.68 |
> | fixed MR 0.5 | 81.80 | 95.93 |
> | Dynamic MR | 81.97 | 96.04 |
> | R$^2$MAE | **82.00** | **96.05** |
>
> **Language:** We trained RoBERTa-base and RoBERTa-medium models on the FineWeb dataset for 10B tokens (max seq length 128, comparable to [2]) and fine-tuned them on GLUE benchmarks (MNLI, QQP, SST-2, QNLI). We evaluated: 1. Default MLM (MR 0.15); 2. R$^2$MAE (p ~ U(0.15, 0.4)); 3. Dynamic MR [3] (0.4 to 0.15); 4. MLM with a fixed 0.4 MR. Fine-tuning accuracies are comparable to those in [2]. The results show R$^2$MAE achieves the best overall rank, with a more pronounced advantage in the larger RoBERTa-Base model.
>
> | RoBERTa-Medium (52M) | MNLI | QQP | SST-2 | QNLI | Mean Rank |
> |:---|:---:|:---:|:---:|:---:|:---:|
> | MLM default (MR 0.15) | 80.8 | 89.8 | 89.9 | 86.3 | 3.25 |
> | Fixed MR 0.4 | 80.3 | 89.7 | 90.1 | 86.6 | 3.50 |
> | Dynamic MR | 80.8 | **90.1** | 90.5 | **87.1** | 1.50 |
> | R$^2$MAE | **80.9** | **90.1** | **90.6** | 86.7 | **1.25** |
>
> | RoBERTa-Base (125M) | MNLI | QQP | SST-2 | QNLI | Mean Rank |
> |:---|:---:|:---:|:---:|:---:|:---:|
> | MLM default (MR 0.15) | 81.5 | 90.7 | 91.7 | 87.8 | 3.00 |
> | Fixed MR 0.4 | 81.7 | 90.7 | 91.2 | 88.5 | 3.00 |
> | Dynamic MR | 81.8 | 90.7 | 91.4 | **89.1** | 2.00 |
> | R$^2$MAE | **81.9** | **90.8** | **91.9** | 88.6 | **1.25** |
>
> **Note:** In these fine-tuning evaluations, the advantage of R$^2$MAE appears smaller than in our linear probing or zero-shot evaluations. This is expected, as the identical fine-tuning pipeline can reduce differences from pre-training. Nevertheless, the consistent advantage of R$^2$MAE across tasks and domains strongly supports its effectiveness and generalizability.
>
> ### **New Results 2: Explicit Connection between R$^2$MAE and the High-Dimensional Linear Regression Framework**
>
> Inspired by reviewer comments, we evaluated R$^2$MAE within our linear model framework. Specifically, we simulated the latent space model from Fig. 1E. Intriguingly, we observed that for features of different strengths, R$^2$MAE's risk tends to align with the *optimal* risk, with its "effective masking ratio" decreasing as feature strength decays.
>
> | Normalized test risk | p=0.45 | p=0.5 | p=0.55 | p=0.6 | R$^2$MAE, p~U(0.45,0.6) | Effective p of R$^2$MAE |
> |---|---|---|---|---|---|---|
> | β = 1st Eig | 0.687 | 0.671 | 0.666 | 0.671 | 0.665 | ~0.55 |
> | β = 2nd Eig | 0.751 | 0.748 | 0.752 | 0.772 | 0.750 | 0.5-0.55 |
> | β = 3rd Eig | 0.839 | 0.842 | 0.849 | 0.872 | 0.842 | ~0.5 |
> | β = 4th Eig | 0.880 | 0.889 | 0.898 | 0.929 | 0.888 | 0.45-0.5 |
>
>
> Our new results support the intuition that R$^2$MAE achieves a “feature-adaptive” masking ratio that approximates dataset-specific optimal masking ratios, allowing it to outperform any fixed ratio in settings with multi-scale features. This is validated by real data (Tables 4 and 5), where R$^2$MAE achieves near-optimal reconstruction across its entire masking range (0.1-0.5), while fixed-rate settings only well cover a smaller range. This result aligns with the robust advantage of R$^2$MAE and strengthens the value of our work.
>
> ### **Point-by-point response to other specific concerns**
>
> In response to the reviewer’s comments, we have performed extensive experiments on vision and language mask pretraining models. Our response to each question is provided as follows.
>
> **W1. Theoretical-Algorithmic Gap**: This is a good point. We acknowledge that our theoretical results are based on linear models that deviate from real neural networks, as mentioned in the subsection “Relation with Real Network Optimization” in Section 3.1. Nevertheless, a rich body of literature has applied ridgeless regression models and random matrix theory to successfully recapitulate neural network behaviors like double descent and the scaling law [4,5]. Despite the deviations, we believe that our extensive evaluations of neural network and linear model behaviors convincingly support the validity of our framework as a “working theory.”
>
> **W2. β Estimation**: Indeed, the β estimation procedure is a simplification that enables feature-wise analysis. Nevertheless, we argue that the “feature” here does not necessarily mean individual pixels or tokens. Intuitively, the MSE loss of a sample can be approximated by the reconstruction loss in a latent feature space, linked by a decoding transformation. This insight suggests that we can investigate mask reconstruction behavior by evaluating different eigenvectors of the design matrix covariance as β. Fully addressing the underlying non-linearity remains an open problem and a promising direction for future research.
>
> **W3, Q1. Comparison on General Datasets**: Thank you for raising this excellent point. We agree that incorporating results on image and language modeling would greatly improve our work. Please refer to the earlier section for these results. Overall, they demonstrate the superior performance of R²MAE, which aligns well with the results on biological data.
>
> **Q2. The possibility of determining an optimal masking ratio per sample**: This is a good question. First, as we demonstrated in Section 3.5 of the paper, the optimal masking ratio varies for different downstream tasks. Therefore, in the pretraining phase, finding a single optimal masking ratio may not be well-defined, as it defeats the purpose of training a universal model. Second, a real-world downstream task (e.g., image classification, GLUE tasks) can be highly complex and require learning features at different scales. In contrast, R²MAE learns a spectrum of features during pretraining through a feature-adaptive masking mechanism, a point we clarified earlier with our new results. We will incorporate relevant discussions on this topic in future versions of our work to make our presentation more self-contained. Please let us know if there are any additional concerns.
>
> [1] He, Kaiming, et al. "Masked autoencoders are scalable vision learners." CVPR. 2022.
> [2] Wettig, Alexander, et al. "Should You Mask 15% in Masked Language Modeling?." EACL. 2023.
> [3] Ankner, Zachary, et al. "Dynamic Masking Rate Schedules for MLM Pretraining." EACL. 2024.
> [4] Hastie, Trevor, et al. "Surprises in high-dimensional ridgeless least squares interpolation." Annals of statistics. 2022.
> [5] Bahri, Yasaman, et al. "Explaining neural scaling laws." Proceedings of the National Academy of Sciences. 2024.

---

> > ### Comment · Reviewer_yjuN · 2025-08-05
> > **Response to the authors**
> >
> > I thank the authors for the detailed rebuttal. Many of my concerns have been addressed. I increased my score.

---

### Official Review · Reviewer_RbST · 2025-07-04

**Clarity:** 3
**Significance:** 4
**Originality:** 3
**Rating:** 5
**Confidence:** 2

**Summary:**

Here, the authors propose a theoretical analysis that leads to a masking strategy for masked language models called "R^2MAE", and demonstrate its effectiveness in self-supervised representation learning for DNA and single-cell.

**Questions:**

(See questions in strengths/weaknesses)

**Ethical Concerns:**

["NO or VERY MINOR ethics concerns only"]

**Final Justification:**

The rebuttal by the reviewers do a good job of demonstrating the generality of their method, as well as show that their proposed method aligns well with the optimal risk. Although the connection between their theory and their proposed masking scheme still remains unclear to me, these are strong empirical results. My original score for this paper was high to begin with, so I will maintain my score.

**Limitations:**

Yes

**Quality:**

4

**Strengths And Weaknesses:**

This paper presents a generally strong theoretical analysis, which essentially argues that the success of the masked-based pretraining objective can be explained as a trade-off between minimizing bias and optimizing risk (i.e. ensuring that for each feature, the model learns which other features can be  used to reconstruct the feature, while minimizing overfitting). They do convincing empirical experiments to validate the key predictions of their theoretical analysis, showing that the masked-based pretraining is only beneficial in the overparameterized regime, and that optimal masking ratio differs by evaluation task.

Based upon these insights, the authors propose a new masking strategy (R^2MAE), which contrary to previous adaptive masking proposals, simply samples masking ratio during pre-training at uniform, and demonstrate the effectiveness of this strategy on two different biological domains, DNA sequences and single-cell gene expression. These empirical results are strong, demonstrating that their proposed method can outperform even SOTA proposals on at least some kinds of tasks in their respective domains, on standardized benchmarks at least, and critically, frequently over fixed/dynamic masking ratio proposals.

Even as the empirical result is strong, I'm struggling to connect their masking strategy proposal to the theory presented - I understand that different masking ratios have different bias-risk trade-offs, but how does sampling masking ratio at uniform necessarily lead to a better final model? By intuition, I'd expect this to lead to a model that's an "expectation" over all of these masking ratios (including ones with high bias/risk), so I'm not fully understanding why this is effective.

---

> ### Author Rebuttal · Authors · 2025-07-31
>
> Thank you very much for your comments. Our rebuttal is organized as follows. We will first present new results that address general feedback from all reviewers. They include new experiments on vision and language models and a strengthened connection between R$^2$MAE and our theoretical framework. We will next provide a point-by-point response to specific concerns.
>
> ### **New Results 1: Experiments of R$^2$MAE on Vision and Language Models**
>
> In general, reviewers appreciated our empirical study but expressed curiosity about whether R$^2$MAE would achieve SOTA performance in more established domains like vision and language modeling. We were fortunate to gain access to additional computational resources during the rebuttal period to perform these experiments.
>
> **Vision:** Our implementations closely follow established practices from the vision MAE work [1]. We implemented different mask ratio settings on the ViT-base MAE model using their official Pytorch codebase. The settings include: 1. Default MAE (constant 0.75 mask ratio, MR); 2. R$^2$MAE (p ~ U(0.6, 0.9)); 3. Dynamic MR [3] (linearly decreasing MR from 0.9 to 0.6); 4. High (0.9) and low (0.5) fixed MR baselines. Given the time limit, all models were trained for 150 epochs. While this is shorter than the 800 epochs in [1], their analysis shows predictable improvements with longer training schedules. Therefore, while our absolute accuracies may be suboptimal, we expect the relative performance across settings to be comparable and the conclusion to hold if trained longer. All other hyperparameters follow the defaults in [1]. Models were then fine-tuned for classification for 100 epochs, following [1].
>
> Our results below show R$^2$MAE achieves the best performance in both top-1 and top-5 accuracy. We found that R$^2$MAE introduces fluctuations in the pretraining loss, likely due to variable sequence lengths since MAE only encodes unmasked tokens. This presents a potential area for even further improvement.
>
> | ViT-base | Acc@1 | Acc@5 |
> |:---|:---:|:---:|
> | MAE default (MR 0.75) | 81.97 | 96.02 |
> | fixed MR 0.9 | 81.20 | 95.68 |
> | fixed MR 0.5 | 81.80 | 95.93 |
> | Dynamic MR | 81.97 | 96.04 |
> | R$^2$MAE | **82.00** | **96.05** |
>
> **Language:** We trained RoBERTa-base and RoBERTa-medium models on the FineWeb dataset for 10B tokens (max seq length 128, comparable to [2]) and fine-tuned them on GLUE benchmarks (MNLI, QQP, SST-2, QNLI). We evaluated: 1. Default MLM (MR 0.15); 2. R$^2$MAE (p ~ U(0.15, 0.4)); 3. Dynamic MR [3] (0.4 to 0.15); 4. MLM with a fixed 0.4 MR. Fine-tuning accuracies are comparable to those in [2]. The results show R$^2$MAE achieves the best overall rank, with a more pronounced advantage in the larger RoBERTa-Base model.
>
> | RoBERTa-Medium (52M) | MNLI | QQP | SST-2 | QNLI | Mean Rank |
> |:---|:---:|:---:|:---:|:---:|:---:|
> | MLM default (MR 0.15) | 80.8 | 89.8 | 89.9 | 86.3 | 3.25 |
> | Fixed MR 0.4 | 80.3 | 89.7 | 90.1 | 86.6 | 3.50 |
> | Dynamic MR | 80.8 | **90.1** | 90.5 | **87.1** | 1.50 |
> | R$^2$MAE | **80.9** | **90.1** | **90.6** | 86.7 | **1.25** |
>
> | RoBERTa-Base (125M) | MNLI | QQP | SST-2 | QNLI | Mean Rank |
> |:---|:---:|:---:|:---:|:---:|:---:|
> | MLM default (MR 0.15) | 81.5 | 90.7 | 91.7 | 87.8 | 3.00 |
> | Fixed MR 0.4 | 81.7 | 90.7 | 91.2 | 88.5 | 3.00 |
> | Dynamic MR | 81.8 | 90.7 | 91.4 | **89.1** | 2.00 |
> | R$^2$MAE | **81.9** | **90.8** | **91.9** | 88.6 | **1.25** |
>
> **Note:** In these fine-tuning evaluations, the advantage of R$^2$MAE appears smaller than in our linear probing or zero-shot evaluations. This is expected, as the identical fine-tuning pipeline can reduce differences from pre-training. Nevertheless, the consistent advantage of R$^2$MAE across tasks and domains strongly supports its effectiveness and generalizability.
>
> ### **New Results 2: Explicit Connection between R$^2$MAE and the High-Dimensional Linear Regression Framework**
>
> Inspired by reviewer comments, we evaluated R$^2$MAE within our linear model framework. Specifically, we simulated the latent space model from Fig. 1E. Intriguingly, we observed that for features of different strengths, R$^2$MAE's risk tends to align with the *optimal* risk, with its "effective masking ratio" decreasing as feature strength decays.
>
> | Normalized test risk | p=0.45 | p=0.5 | p=0.55 | p=0.6 | R$^2$MAE, p~U(0.45,0.6) | Effective p of R$^2$MAE |
> |---|---|---|---|---|---|---|
> | β = 1st Eig | 0.687 | 0.671 | 0.666 | 0.671 | 0.665 | ~0.55 |
> | β = 2nd Eig | 0.751 | 0.748 | 0.752 | 0.772 | 0.750 | 0.5-0.55 |
> | β = 3rd Eig | 0.839 | 0.842 | 0.849 | 0.872 | 0.842 | ~0.5 |
> | β = 4th Eig | 0.880 | 0.889 | 0.898 | 0.929 | 0.888 | 0.45-0.5 |
>
>
> Our new results support the intuition that R$^2$MAE achieves a “feature-adaptive” masking ratio that approximates dataset-specific optimal masking ratios, allowing it to outperform any fixed ratio in settings with multi-scale features. This is validated by real data (Tables 4 and 5), where R$^2$MAE achieves near-optimal reconstruction across its entire masking range (0.1-0.5), while fixed-rate settings only well cover a smaller range. This result aligns with the robust advantage of R$^2$MAE and strengthens the value of our work.
>
> ### **Point-by-point response to other specific concerns**
>
> The only concern from the reviewer is the connection between R²MAE and the linear model. We are grateful for the feedback which inspired the second part of our new results. Here, we highlight the difference between the R²MAE setting and the fixed mask ratio setting in the linear model framework. In the linear model, R²MAE translates to first sampling a row-wise masking ratio $p_i \sim U(p_{\text{min}}, p_{\text{max}})$ and then using it to sample each row of the masking matrix $Z$. This introduces a subtle difference in the effective sample matrix $X_{sub}$, as the sampling probability for each row is no longer a constant ($p$) but depends on the specific value of $p_i$. Due to this difference, deriving the closed-form solution for the test risk becomes very challenging and remains a topic for future research. At the very least, this complexity suggests that the test risk of R²MAE is not identical to that of any fixed masking ratio. We further empirically validated this, as shown in the second part of our new results. We believe these results effectively address the concern. Please also refer to the first part of new results which contains additional validation in vision and language models that improve our work. Please let us know if there are any additional concerns.
>
> [1] He, Kaiming, et al. "Masked autoencoders are scalable vision learners." CVPR. 2022.
> [2] Wettig, Alexander, et al. "Should You Mask 15% in Masked Language Modeling?." EACL. 2023.
> [3] Ankner, Zachary, et al. "Dynamic Masking Rate Schedules for MLM Pretraining." EACL. 2024.

---

> > ### Comment · Reviewer_RbST · 2025-08-04
> >
> > Thank you to the reviewers for this rebuttal - these new results do a good job of demonstrating the generality of their method, as well as show that their proposed method aligns well with the optimal risk. Although the connection between their theory and their proposed masking scheme still remains unclear to me, these are strong empirical results.

---

> > > ### Author Response · Authors · 2025-08-05
> > >
> > > Thank you very much for your response. Indeed, the theoretical connection between the linear model framework and our method should be better clarified, beyond the empirical results and theoretical difficulties discussed in the earlier rebuttal. Here we would like to further elaborate on this point. Please note that the following discussion is not fully theoretically rigorous, but we hope it will be helpful for building intuition.
> > >
> > > Our main theoretical result on the limiting risk in the spiked covariance model suggested that the behavior of the risk curve is largely determined by the quadratic term (Eq. 7), where the alignment between β and the ground truth covariance Σ (reduced to β^T v in this case) plays a key role in determining the optimal p. For more general models, while a closed-form limiting risk remains infeasible, the same conclusion holds empirically (Fig. 1E). Conversely, this implies that a fixed masking ratio would only be optimal for features with a certain level of alignment.
> > >
> > > Intuitively, the goal of mask-pretraining is to learn all “eigenvectors” from data through the masked prediction procedure. Importantly, as these features are eventually learned simultaneously (corresponding to a number of parallel linear models in the theoretical framework), an ideal method should automatically achieve near-optimal risk for a variety of features β. Therefore, we concluded that it is essential to expose the model to a range of masking ratios during pretraining. Our empirical results show that the simplest possible approach that achieves this (R^2 MAE, uniformly covering a range of masking ratios) is sufficient/effective. A full theoretical characterization of its feature-adaptive mechanism would be highly nontrivial and requires future work.
> > >
> > > We realized these intuitions are not fully explicit in the current paper, and we appreciate the reviewer for highlighting this point. We will further polish and incorporate this discussion in the revised paper. We hope this reply helps and would be happy to hear your further thoughts.

---

### Decision · Program_Chairs · 2025-09-17

**Decision:**

Accept (poster)

**Comment:**

This paper develops a theoretical framework for understanding mask-based pretraining through the lens of high-dimensional ridgeless linear regression, offering new insights into when and why masked autoencoding is effective and how optimal mask ratios depend on data and model size. Guided by this analysis, the authors introduce R2MAE (Randomly Random Mask AutoEncoding), a simple scheme that samples masking ratios uniformly during training. Applied to DNA sequence and single-cell gene expression tasks, the new method achieves SOTA or improved performance over standard masking strategies, demonstrating the value of their theoretically derived method.

The reviewers generally acknowledged the importance of the work in contributing a theoretical foundation about why and how masked autoencoding works. This by itself is an important contribution. Furthermore, the reviewers appreciated the presented, wide-range empirical validation of predictions from this theoretical work for a diverse set of network architectures. This full range approach in addressing an open scientific question in ML is impressive. Some concerns were raised over some of the linear and other approximations made in the analysis, as well as about the range of application presented in the original submission. The authors well addressed the latter by showing new empirical validation over a wide range of additional datasets that were all favorable for the new method. Inclusion of these results and some refinements on the clarity of the writing will turn this paper into a valuable contribution for the NeurIPS community.